# In situ X-ray and acoustic observations of deep seismic faulting upon phase transitions in olivine

Tomohiro Ohuchi [1] ✉, Yuji Higo[2], Yoshinori Tange[2,3], Takeshi Sakai[1], Kohei Matsuda[1] & Tetsuo Irifune[1,4]

The activity of deep-focus earthquakes, which increases with depth from ~400 km to a peak at ~600 km, is enigmatic, because conventional brittle failure is unlikely to occur at elevated pressures. It becomes increasingly clear that pressure-induced phase transitions of olivine are responsible for the occurrence of the earthquakes, based on deformation experiments under pressure. However, many such experiments were made using analogue materials and those on mantle olivine are required to verify the hypotheses developed by these studies. Here we report the results of deformation experiments on $(Mg,Fe)_2SiO_4$ olivine at 11–17 GPa and 860–1350 K, equivalent to the conditions of the slabs subducted into the mantle transition zone. We find that throughgoing faulting occurs only at very limited temperatures of 1100–1160 K, accompanied by intense acoustic emissions at the onset of rupture. Fault sliding aided by shear heating occurs along a weak layer, which is formed via linking-up of lenticular packets filled with nanocrystalline olivine and wadsleyite. Our study suggests that transformational faulting occurs on the isothermal surface of the metastable olivine wedge in slabs, leading to deep-focus earthquakes in limited regions and depth range.

$(Mg,Fe)_2SiO_4$ olivine is the major mineral in the Earth's mantle and subducting slabs; it shows successive phase transitions to wadsleyite (at pressures around 13 GPa) and ringwoodite (~20 GPa) with modified spinel and spinel structures, respectively. Olivine decomposes into an assemblage of bridgmanite + ferro-periclase (~24 GPa) with increasing pressure, although the transitions may proceed via an intermediate spinelloid structure[1]. An increase in the frequency of earthquakes at depths from ~400 km to ~600 km[2] seems to be linked to the phase transitions of olivine. An abrupt decrease in the seismicity from a peak at ~600 km to almost zero at 680 km[2], with a very limited number of earthquakes at deeper regions down to 750 km[3], also implies an essential role of the phase transitions of olivine on occurrence of deep-focus earthquakes. The concept of transformational faulting upon the phase transitions of olivine, which is triggered by propagation and

linking-up of lens-shaped 'anticracks', has been widely proposed to explain faulting in the mantle transition zone (410 – 660 km in depth), based on experimental studies on a germanate analogue $(Mg_2GeO_4)$[4,5] and ice[6]. The anticrack is orientated normal to the direction of the maximum compressive stress and is filled with a nanocrystalline aggregate of the high-pressure phases, that behaves as a low-viscosity layer due to grain-size-sensitive (GSS) creep[4,5,7]. However, $Mg_2GeO_4$ olivine transforms directly to the spinel phase without passing through the modified spinel phase[8], implying that this might not be a perfect analogue of mantle olivine[8]. A pioneering work on $(Mg,Fe)_2SiO_4$ olivine[8] pointed out the possibility of faulting at the onset of the olivine-wadsleyite transition at 15 GPa and 1550–1650 K based on the microstructural observations of recovered samples. A recent study on fayalite $(Fe_2SiO_4)$[9] confirmed that transformational faulting can be

[1]Geodynamics Research Center, Ehime University, Matsuyama 790-8577, Japan. [2]Japan Synchrotron Radiation Research Institute, Sayo, Hyogo 679-5198, Japan. [3]Sumitomo Electric Industries Ltd., Itami, Osaka 664-0016, Japan. [4]Earth-Life Science Institute, Tokyo Institute of Technology, Tokyo 152-8550, Japan. ✉e-mail: ohuchi@sci.ehime-u.ac.jp

initiated by the olivine-spinel transition at 4−9 GPa but suggested that the development of substantial fault slip and the propagation of the rupture requires an additional mechanism such as adiabatic instability[10−12]. Both of these studies on silicate olivine (San Carlos or fayalite), however, were performed with the 'stress-relaxation test'[13], where the timing and degree of sample deformation cannot be controlled. Thus, further experimental studies are needed on mantle olivine using advanced techniques with well-controlled strain rates under pressure and temperature conditions of the mantle transition region to address the cause of the deep-focus earthquakes.

In this work, we performed high-pressure, high-temperature deformation experiments using in situ synchrotron X-ray diffraction and radiography combined with acoustic recording on a sintered body of a natural olivine powder with a composition of $Mg_{1.8}Fe_{0.2}SiO_4$ (hereafter, OL100 sample). We also used a sintered aggregate of a 92:8 mixture of this olivine and natural orthoenstatite powders (OL92). In total 15 deformation runs were conducted at 11−17 GPa and 860−1350 K using a deformation-DIA apparatus with controlled strain rates in $\sim2 \times 10^{-5} - 1 \times 10^{-4}\ s^{-1}$ (Fig. 1 and Table 1). X-ray radiographs of the sample under deformation were taken to monitor the occurrence of throughgoing faulting in the sample, in addition to the acoustic emission (AE) measurements during the deformation. At temperatures of 1100−1160 K, throughgoing faulting accompanied intense AEs from both inside and outside of the sample. Linking-up of lenticular packets filled with nanocrystalline olivine and wadsleyite was confirmed to form a weak layer, that induces strain localization followed by throughgoing faulting. Formation of such weak layers around the surface of the metastable olivine wedge (MOW) could induce transformational faulting and its resultant deep seismicity in subducted slabs.

## Results

### Experimental conditions and mechanical behavior

Three pressure-temperature-time (P-T-t) paths were adopted for the deformation runs. In P-T-t path#1, deformation started in the stability region of olivine, followed by a pressure increase to the stability region of wadsleyite. Certain amounts of wadsleyite (>~5 vol.%) were observed at 1250 K by X-ray diffraction measurements. At lower temperatures, trace amounts (~0.1 − 0.2 vol.%) of wadsleyite were found in the recovered samples at 1160 K, but no wadsleyite was confirmed at 960 K and 1060 K. In P-T-t path#2, the sample was pressurized to the wadsleyite region just before the deformation to enhance nucleation of wadsleyite. Wadsleyite was noted at 1160 K by X-ray diffraction measurements, while only trace amounts of wadsleyite were identified in the recovered samples at 1110 K and those with temperature ramping from 860 to 1160 K in P-T-t path#3 (i.e., a modified P-T-t path#2, see Methods).

Differential stress was almost constant at strains of >0.1 and pressures of >11 GPa (i.e., the wadsleyite/ringwoodite stability field)[14] in all of the runs. Yield strengths of our samples are significantly lower than the steady-state creep strengths of dry olivine (i.e., upper limit of the strength of olivine) controlled by the Peierls creep[15,16] (Fig. 2a). While the highest strength at each temperature is close to the value calculated from the Peierls creep of wet olivine[17], neither dislocation creep[18], nor dislocation-accommodated grain boundary sliding of wet olivine[19] account for the observed strengths. Variations in the amount of water in olivine (10−420 wt. ppm: Table 1) would have enhanced the dispersion of strength values. Partial contribution of GSS creep of olivine/wadsleyite induced by the nucleation of wadsleyite[17] to the bulk strength may cause softening of the samples, resulting in further dispersion of strength values. The acoustic activities are quite high with either high AE rates or high cumulative energy release at temperatures up to 1160 K, but greatly decrease and become almost silent at higher temperatures, indicating the dominance of microcracking-free plastic deformation above 1160 K (Fig. 2b). The first motion at each AE sensor shows that most of the AEs were generated by shear cracks (i.e., both positive and negative polarities) or those associated with compaction, while isotropic compaction (i.e., all negative polarities)[20] due to formation of the olivine-wadsleyite anticracks[4] is too small (~7 vol.%[21]) to be detected using our monitoring system.

### Throughgoing faulting and acoustic activity

In the deformation stages of three runs (M2676 and M3100, both at 1160 K; M3425, at 860−1100 K), the semi-brittle flow was terminated by throughgoing faulting at 1100−1160 K followed by a sudden large pressure drop (blow-out), neither throughgoing faulting nor blow-out was observed in the other 11 deformation runs (Table 1). In these three runs, one or two large AEs (maximum amplitude > 3 V) occurred inside the sample, followed by relatively large AEs outside the sample and aligned with the fault plane crossing the sample (Fig. 3 and Supplementary Fig. 1). This suggests that the occurrence of a throughgoing

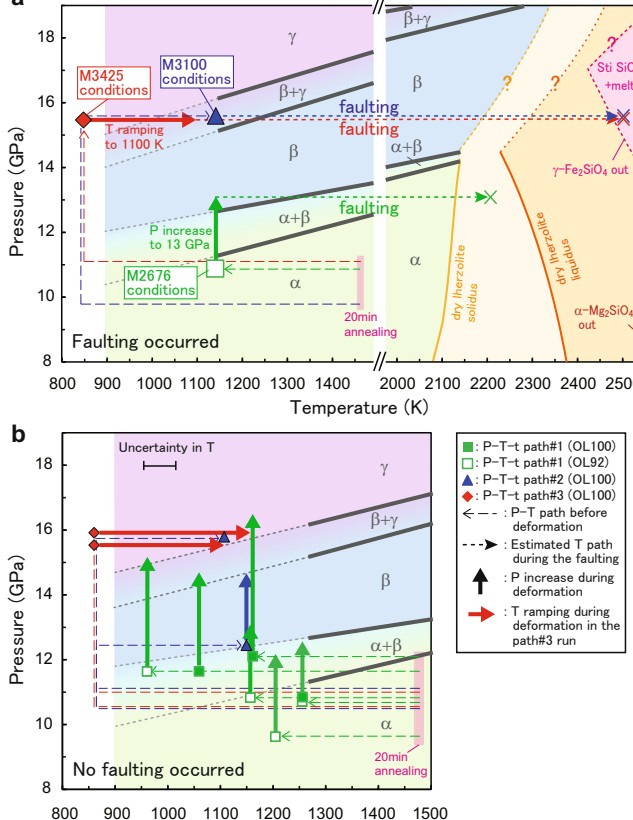

**Fig. 1 | Summary of experimental conditions. a** Throughgoing faulting occurred. **b** No throughgoing faulting occurred. The long-dashed arrows indicate the P-T-t paths for our experiments. Squares, triangles, and diamonds represent the P-T-t path#1 (normal), #2 (overpressurized just before the deformation), and #3 (temperature ramping during the deformation), respectively. Large symbols represent the runs with throughgoing faulting (M2676, M3100, and M3425). Crosses show the lower limit of the peak temperature during the throughgoing faulting (estimated from the microstructures: see text for details). Short dashed lines are the estimated T-paths of shear heating. Red thick arrows show the temperature ranges during each deformation run of path#3. Solid and open symbols represent the runs in which the OL100 and OL92 samples were used, respectively. The equilibrium boundaries of α (olivine), β (wadsleyite), and γ (ringwoodite) for $Mg_{1.8}Fe_{0.2}SiO_4$ are shown by gray solid lines[14,34]. Pale orange curve: solidus for dry lherzolite[26]. Dark-orange curve: liquidus for dry lherzolite[26]. Brown curve: melting of forsterite[25]. Pink curve: incongruent melting of $\gamma$-$Fe_2SiO_4$ to a liquid phase and stishovite (Sti)[24] (i.e., the lower limit of the melting temperature of β/γ-$Mg_{1.8}Fe_{0.2}SiO_4$). The M2472 run, in which a blow-out occurred in the early stage of deformation, is not shown.

**Table 1 | Experimental conditions and results**

| Run No. | Starting sample | P (GPa) [a] | T (K) | Total strain | Strain rate (s⁻¹) | $\sigma_{y, 021}$ (GPa) [b] | $\sigma_{y, 101}$ (GPa) [b] | $\sigma_{y, 130}$ (GPa) [b] | $\sigma_{y, 131}$ (GPa) [b] | Phase assemblage [c] | $C_{OH}$ (wt. ppm) [d] | $N_{in}$ [e] | $N_{out}$ [e] | Formation of throughgoing faults | 2-D detector used |
|---|---|---|---|---|---|---|---|---|---|---|---|---|---|---|---|
| **P-T-t path#1** | | | | | | | | | | | | | | | |
| M2472 [f] | OL100 | 14.3 | 1350 | 0.02 | $1.0 \times 10^{-5}$ | --- | --- | --- | --- | Ol | --- | 10 | 58 | No | MarCCD |
| M2673 | OL92 | 15.1 | 960 | 0.18 | $3.5 \times 10^{-5}$ | 2.57 (±0.10) | 2.15 (±0.24) | 2.64 (±0.14) | --- | Ol + Mj + Cpx | 150 (±30) | 48 | 17 | No | MarCCD |
| M2600 | OL100 | 14.6 | 1060 | 0.15 | $2.3 \times 10^{-5}$ | 2.27 (±0.13) | 1.97 (±0.12) | 1.99 (±0.15) | --- | Ol | 10 (±10) | 9 | 3 | No | MarCCD |
| M2676 [f] | OL92 | 13.1 | 1160 | 0.14 | $5.0 \times 10^{-5}$ | 1.77 (±0.08) | 1.34 (±0.12) | 2.15 (±0.09) | --- | Ol + Mj + Cpx + Wad +Amorphous | 130 (±20) | 26 | 27 | Yes | MarCCD |
| M2880 | OL92 | 12.6 | 1250 | 0.23 | $1.2 \times 10^{-4}$ | 1.74 (±0.08) | 1.56 (±0.13) | 1.69 (±0.09) | 1.28 (±0.06) | Ol + Mj + Cpx + Wad | --- | 0 | 1 | No | MarCCD |
| M2955 | OL92 | 14.8 | 1160 | 0.24 | $5.7 \times 10^{-5}$ | 2.69 (±0.15) | 1.85 (±0.13) | 2.61 (±0.19) | --- | Ol + Mj + Cpx | 170 (±10) | 1 | 0 | No | MarCCD |
| M3101 | OL100 | 16.6 | 1160 | 0.14 | $5.7 \times 10^{-5}$ | 2.39 (±0.15) | 2.00 (±0.21) | 2.18 (±0.19) | --- | Ol + Wad | 50 (±10) | 9 | 1 | No | MarCCD |
| M2957 | OL92 | 12.0 | 1200 | 0.25 | $6.2 \times 10^{-5}$ | 2.72 (±0.11) | 2.06 (±0.15) | 2.36 (±0.10) | 1.67 (±0.05) | Ol + Mj + Cpx | 420 (±10) | 0 | 0 | No | MarCCD |
| M2959 | OL100 | 12.6 | 1250 | 0.25 | $1.3 \times 10^{-4}$ | 1.78 (±0.11) | 1.81 (±0.16) | 1.57 (±0.10) | 1.06 (±0.11) | Ol + Wad | 140 (±10) | 4 | 4 | No | MarCCD |
| **P-T-t path#2** | | | | | | | | | | | | | | | |
| M3078 | OL100 | 14.6 | 1160 | 0.15 | $5.0 \times 10^{-5}$ | 1.57 (±0.16) | 1.11 (±0.17) | 1.43 (±0.17) | 1.23 (±0.13) | Ol + Wad | 50 (±10) | 16 | 7 | No | MarCCD |
| M3100 [f] | OL100 | 15.6 | 1160 | 0.07 | $2.3 \times 10^{-5}$ | 1.54 (±0.13) | 1.50 (±0.15) | 1.58 (±0.24) | --- | Ol + Wad + Rin + Fe-rich + Si-rich | --- | 9 | 25 | Yes | MarCCD |
| M3423 | OL100 | 16.0 | 1110 | 0.13 | $4.2 \times 10^{-5}$ | 3.05 (±0.28) | 2.36 (±0.22) | 2.51 (±0.26) | --- | Ol + Wad | 60 (±10) | 1 | 0 | No | WidePix 5 × 5 |
| **P-T-t path#3** | | | | | | | | | | | | | | | |
| M3424 | OL100 | 16.1 | 860–1160 | 0.14 | $3.5 \times 10^{-5}$ | 3.11 (±0.25) | 2.26 (±0.20) | 3.66 (±0.37) | --- | Ol + Wad | 210 (±30) | 2 | 0 | No | WidePix 5 × 5 |
| M3425 [f] | OL100 | 15.5 | 860–1100 | 0.14 | $4.4 \times 10^{-5}$ | 3.41 (±0.23) | 3.32 (±0.27) | 3.35 (±0.31) | --- | Ol + Wad + Fe-rich | 160 (±30) | 10 | 20 | Yes | WidePix 5 × 5 |
| M3426 | OL100 | 15.8 | 860–1110 | 0.14 | $3.6 \times 10^{-5}$ | 3.26 (±0.22) | 2.89 (±0.28) | 2.89 (±0.23) | --- | Ol + Wad | 180 (±30) | 4 | 0 | No | WidePix 5 × 5 |
| **Test experiment** | | | | | | | | | | | | | | | |
| M0637 | Quartz beads | ~11 [g] | 297 | --- | --- | --- | --- | --- | --- | Quartz | --- | 34 | 32 | No | --- |

aAverage values of pressure around the yielding point. The maximum values are shown for M2472 and M0637 runs. Error is within 0.2 GPa.
bYield strength evaluated from olivine diffraction patterns.
cOl Olivine, Mj Majorite, Cpx Clinopyroxene, Wad Wadsleyite, Rin Ringwoodite, Fe-rich Fe-rich phase, Si-rich Si-rich phase
dWater content in the recovered samples. Initial water contents are 70 and 80 wt. ppm for the OL100 and OL92 samples, respectively.
eN: Total number of AE events from the inside (in) or outside (out) of the sample.
fFinal blow out.
gEstimated from the relationship between the main-ram load and pressure (determined from the calibration runs using pressure references).

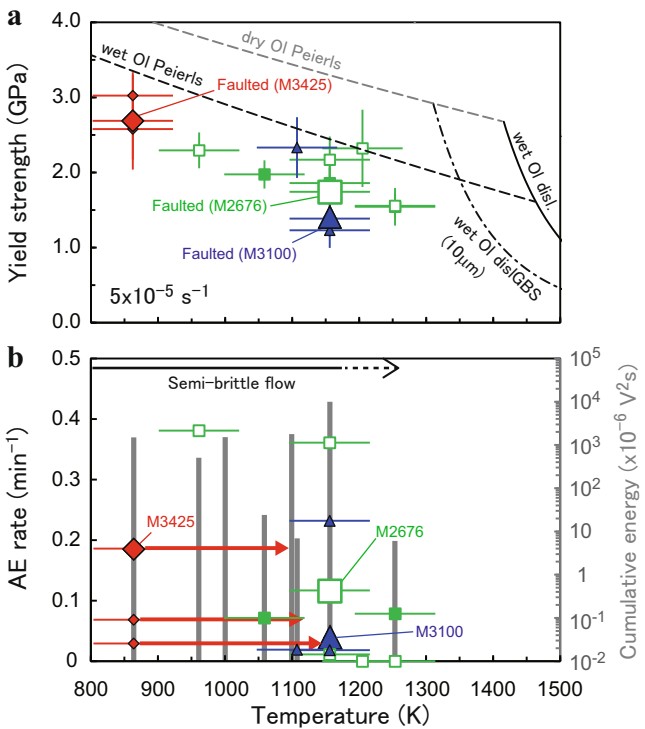

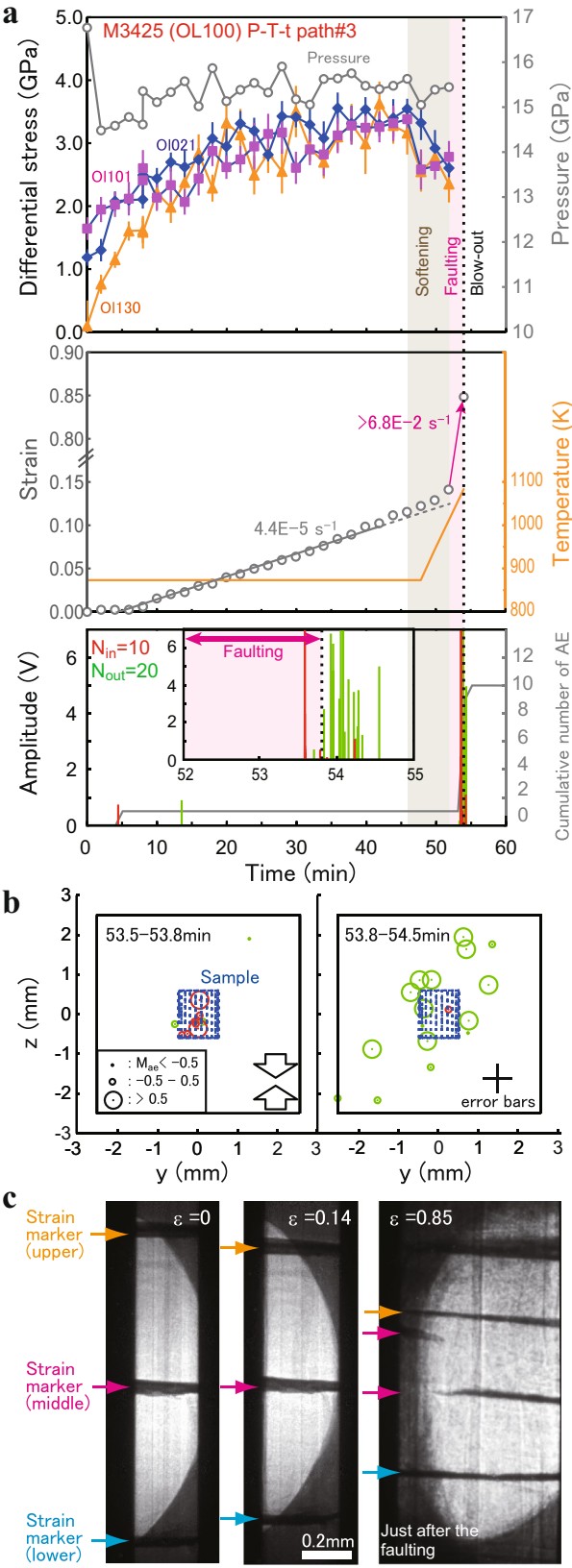

**Fig. 2 | Summary of experimental results as a function of temperature.**
**a** Temperature dependence of the yield strength of the samples. Creep strength of olivine (Ol) is calculated assuming the Peierls creep for sintered dry/wet aggregates[15–17], wet dislocation (disl.) creep[18] and wet dislocation-accommodated grain boundary sliding (dislGBS; for a typical grain size of 10 μm)[19]. Water content of 190 wt. ppm is assumed for the calculations. **b** Temperature dependence of averaged acoustic emission (AE) rate (symbols) and cumulative AE energy release (gray bars). Symbols and red thick-arrows are the same as those in Fig. 1. The error bars represent the uncertainties in temperature or stress.

fault slip caused by the rupture was followed by rupture propagation outside the sample. Among the three runs where faulting and blow-out occurred, one run using the upgraded X-ray detector system with the P-T-t path#3 (M3425) most clearly shows the deformation processes and associated AE activities (Fig. 3a). Radiations of two large AEs (maximum amplitude > 7 V) occurred from the inside of the sample (in 53.5–53.8 min after starting deformation) and subsequent intense AEs from the outside of the sample (53.8–54.5 min) upon throughgoing faulting. The faulting is likely to induce a catastrophic adjustment of the pressure medium/gasket which inevitably leads to a blow-out, as none of other 11 runs without faulting suffered any blow-outs. The blow-out was accompanied by sudden drops in both pressure (ΔP -2 − 3.5 GPa: estimated from the change in the main-ram load) and temperature (to room temperature).

Note that a blow-out is not the cause of faulting in our cell assembly. As demonstrated in M2472, stress abnormally decreased down to −2 GPa (i.e., a strong tensile stress) accompanied by AEs before the occurrence of a blow-out (Supplementary Fig. 4a). Such strong tensile stress, which would be a characteristic phenomenon for blow-outs, was not detected in other runs. Even though sudden sample shortening was observed at the time of a blow-out, development of any faults was not observed in the recovered sample (Supplementary Fig. 4b).

Notable softening of the deforming sample (i.e., combination of a stress drop and an acceleration of strain rate) was observed before the occurrence of throughgoing faulting in M3425 with high time-resolution stress/strain measurements (Fig. 3a). Such softening was also accompanied by the large AEs in other runs with the P-T-t path#3 which experienced no faulting (M3424 and M3426). The softening was

likely to be initiated by temperature ramping ($\dot{T}$ -0.5 K·s⁻¹) during the sample deformation in the P-T-t path#3. Splitting of the middle strain marker in the X-ray radiograph is expected as a consequence of throughgoing faulting. However, we only observed clear split images after the blow-out (Fig. 3c and Supplementary Fig. 1a) in the runs with throughgoing faulting, except for a slight splitting of the marker which

**Fig. 3 | Mechanical (pressure, stress, and strain) and acoustic records in the OL100 sample faulted at 1100 K (M3425). a** Stress values were obtained from three diffraction peaks of olivine (diamonds: 021, squares: 101, triangles: 130). Pressure and strain are shown by gray open circles. Cumulative number of acoustic emissions (AEs) (gray line) and amplitudes (red lines: AEs from the inside of the sample; green: outside of the sample) are plotted against time. Total number of AE events ($N_{in}$: from the inside of the sample: $N_{out}$: outside of the sample) is also shown. The inset shows a magnified view of the AE record during the period of 52–55 min. Timings of softening, throughgoing faulting and blow-out are shown by a brown area, a pink area, and a dotted line, respectively. The pink arrow shows a discontinuous increase in strain due to throughgoing faulting. The error bars represent the uncertainties in stress. **b** Two-dimensional views of AE hypocenters in the deforming sample (blue cylinder) and the pressure medium (black square) during (53.5–53.8 min) and just after the throughgoing faulting (53.8–54.5 min). The black cross shows the uncertainty range for hypocenter locations. Arrows represent the compressional direction. **c** X-ray radiographs of the deforming sample for three values of strain ε representative of the entire experiment. Positions of platinum strain markers are shown by arrows.

occurred 9 min before the blow-out in M2676 (at a strain of 0.14; Supplementary Fig. 1a), indicative of the commencement of throughgoing faulting. Because the X-ray radiographic imaging was interrupted by the acquisition of each diffraction pattern with a long exposure time (70 – 540 s), it was hard to determine the whole faulting process solely by the in situ measurements. Thus, microstructures of the recovered samples were also carefully examined to obtain complementary evidence for the process after the deformation experiments.

## Melting on faults

A throughgoing fault with an angle of 50 – 80° to the compression direction with a displacement of 280 μm occurred in the recovered sample M3425 (Fig. 4). The fault is filled with a gouge layer of thickness 3 – 5 μm. Transmission electron microscope (TEM) observations of a foil fabricated using a focused-ion beam (FIB) revealed that the core region of the gouge mainly consists of round-shaped grains of olivine/wadsleyite with diameters of 0.2–1 μm, while the gouge rim consists of those with smaller grains of 20–200 nm. Although some dislocations are observed in the gouge core, they are absent in the gouge rim, indicating that GSS creep controlled the deformation of the gouge layer. Platinum-iron alloy blobs (10–200 nm) and an interconnected iron-rich phase (FeO = 20–37 wt.%) are also observed in the gouge core (Fig. 4e), showing melting of the platinum strain marker and olivine during the throughgoing faulting. Iron in the platinum-iron alloy blobs was incorporated from the surrounding olivine/wadsleyite grains and the iron-rich phase. The liquidus curve for the platinum-iron alloy constrains the peak temperature during the throughgoing faulting to the range of ~2200–2500 K (at 15.5 GPa)[22,23] or higher. Generation of instantaneous high temperature would be due to adiabatic shear heating upon faulting under the assumption that the observed microstructures were not altered by the blow-outs. The shear heating to the peak temperature would associate grain growth of olivine/wadsleyite, as observed in the gouge core (Fig. 4e), while the smaller grain sizes of olivine/wadsleyite in the nano-regime may be preserved in the gouge rim (Fig. 4c).

Evidence of melting in the gouge is also found in the OL100 sample of M3100 (Fig. 5). Iron-rich nanoparticles (FeO/MgO ~0.4 – 0.9) associated with silicon-rich patches ($SiO_2$ = 62 – 84 wt.%) are distributed in a gouge layer that developed (Figs. 5f–h, 6b). The mixing line along olivine polymorphs and the silicon-rich patches (Fig. 6b) implies that the main constituent of the silicon-rich patches is a $SiO_2$ phase (i.e., stishovite under the experimental conditions). The variation in the chemical compositions of silicon-rich patches (Fig. 6b) would be due to contamination of the energy dispersive X-ray (EDX)

spectra from the neighboring grains of olivine polymorphs (or majorite) in the analyses of the $SiO_2$ phase. The iron-rich phase is also observed in the recovered sample from M3425 (Figs. 4e, 6c). Formation of both the iron-rich and silicon-rich phases may be explained by incongruent melting of olivine polymorphs to an iron-rich liquid and stishovite at a peak temperature of > 2500 K (at 15.5 GPa: Fig. 1a) if we adopt a melting relation of $Fe_2SiO_4$[24]. Taking an incongruent melting of forsterite (i.e., α-phase of $Mg_2SiO_4$) to a liquid and periclase (i.e., MgO) at a temperature above 2800 K (at 15.5 GPa)[25] into account, the possibility that the observed silicon-rich phase is a metastable phase or a quench product cannot be excluded.

Similar to the case of M3425, a throughgoing fault with an angle of ~50° to the compression direction occurred in run M2676 (sample OL92). The fault has a cumulative displacement of ~500 μm and is filled with a gouge layer having a thickness of 5–20 μm (Fig. 5a, b). The gouge consists of round-shaped dislocation-free olivine/wadsleyite grains (20–300 nm in diameter), platinum blobs (10–500 nm) and an interconnected iron-rich phase (Fig. 5c). No wadsleyite grain was observed in off-fault intact domains of the sample. Formation of platinum-iron alloy blobs in the M2676 sample (Fig. 5c) shows that the temperature crossed the solidus line for dry harzburgite (2140 K at 13 GPa)[26] and the liquidus curve for the platinum-iron alloy (~2200–2450 K at 13 GPa)[22,23] during throughgoing faulting. Partial and total melting of the gouge are expected in the cases that the peak temperatures are below and above the liquidus curve for dry harzburgite (~2270 K at 13 GPa)[26], respectively (Fig. 1a). The olivine grains in the gouge in M2676 sample could be the product of back-transformation of wadsleyite or partial crystallization from the melt in the olivine stability field (1900–2100 K at 13 GPa: Fig. 1a). The phase wetting grain boundaries is amorphous (Fig. 5d, e) and enriched in silicon, calcium, and iron ($SiO_2$ = 54–62 wt.%; CaO ~4 wt.%; FeO/MgO ~0.5: Fig. 6a). This chemical composition corresponds to that by low-degree (<40 wt.%) partial melting of dry lherzolite[27] above 2140 K at 13 GPa (Fig. 1a)[26]. As well as case of platinum-iron alloy blobs, nickel-iron alloy blobs (10 – 200 nm), due to melting of the nickel capsule used in M2676 sample (Fig. 5c), also support the generation of instantaneous high temperature exceeding the melting point of pure nickel (~2100 K at 13 GPa)[22].

## Lenticular packets filled with nanocrystalline olivine and wadsleyite

Figure 7 shows the results of the run deformed at 860–1110 K without throughgoing faulting (M3426), using the upgraded X-ray detector system with P-T-t path#3. In this run, the softening is less obvious but two large AEs are observed upon increasing temperature from 860 K to 1110 K during the deformation. Both of these two and other smaller AEs are all confined in the sample region. The recovered sample shows the presence of lenticular packets filled with nanocrystalline olivine and wadsleyite (20 – 300 nm in diameter: Fig. 7c–e), showing grain size reduction of olivine and nucleation of wadsleyite induced by the olivine-wadsleyite transition. Intracrystalline transition via nucleation of wadsleyite on high-density dislocation walls, which have a highly disordered structure like grain boundaries (GBs) in olivine grains, would contribute to the reduction in grain size of olivine[28]. In fact, the growth of numerous nanocrystalline olivine and wadsleyite grains from old olivine grains containing many dislocations is suggested (Fig. 5h). The lenticular packets are aligned with high angles (~30°) to the compression direction and have a width of 2–3 μm. The morphology of the observed microstructures (other than the shape-preferred orientation) is quite similar to that of the 'anticracks' in $Mg_2GeO_4$[4,29] and $Fe_2SiO_4$[9]. Linking-up of the lenticular packets with nanocrystalline olivine and wadsleyite should be the origin of weak fault gouge layers observed in the faulted samples (Figs. 4 and 5), consistent with the hypothesis of transformational faulting based on the anticrack model[4] or the nano-shear band model[5].

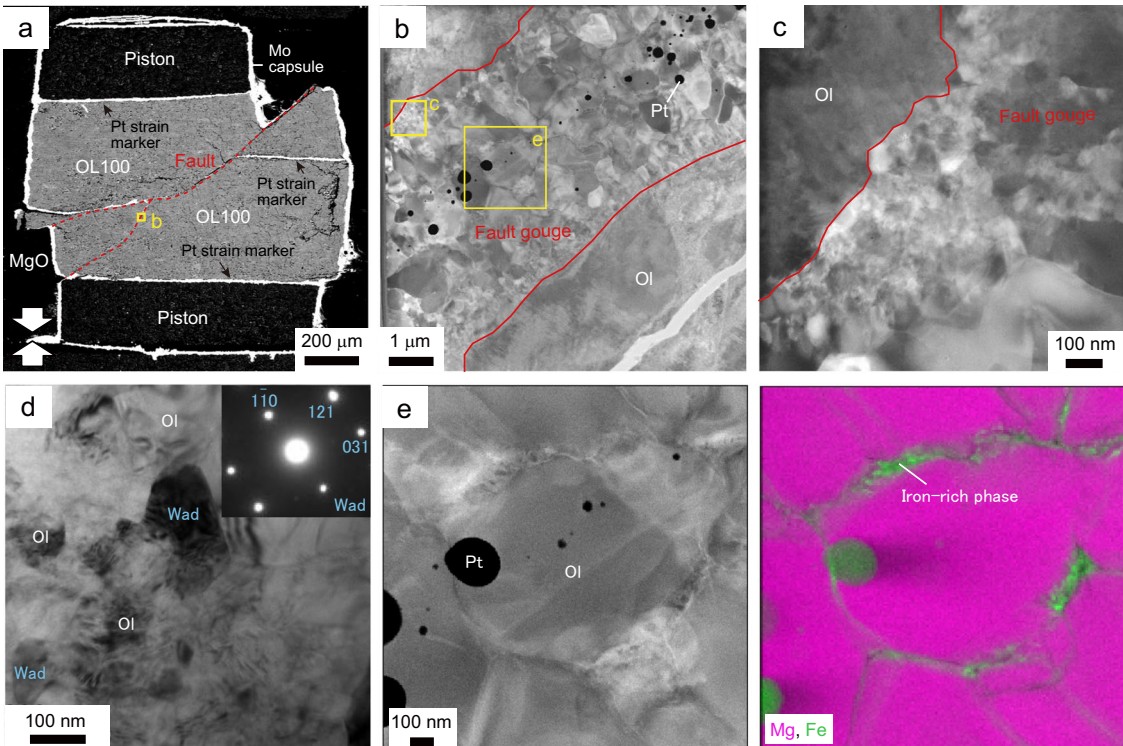

**Fig. 4 | Microstructures of the OL100 sample faulted at 1100 K (M3425).**
**a** Backscattered electron (BSE) image of the recovered sample. The throughgoing faults are highlighted by red-dashed lines. **b** Scanning transmission electron microscope (STEM) image of a fault (located in **a**). Platinum-iron alloy (Pt) blobs are distributed along the gouge core. Ol: olivine. **c** Magnified view of an ultrafine-grained domain in the gouge rim (located in **a**). Grain size is 20–50 nm.

**d** Transmission electron microscope image of a wadsleyite (Wad) nanocrystal in the olivine-wadsleyite gouge aggregate, with associated diffraction pattern. **e** Enlarged STEM image and Mg/Fe element map (located in **b**). In the element map, the intensity of colors accounts for element concentrations (pink: magnesium; green: iron).

## Discussion

The kinetics of the transformed volume fraction are described as the rate constant of the olivine-wadsleyite phase transition ($K$) in the Avrami equation. We estimate $K$ for two typical pressures of 13 and 15.5 GPa in the wadsleyite stability field assuming that preferential nucleation proceeds at GBs (including high-density dislocation walls)[30]. To estimate $K$ under our experimental conditions, we adopt theoretical equations for the kinetics of polymorphic phase transitions on interfaces. The $K$ during the early stages of transition is given as a function of the nucleation rate $\dot{N}$ and growth rate $\dot{x}$[31,32]:

$$K = \frac{\pi}{3} \dot{N} \dot{x}^3 \qquad (1)$$

$$\dot{N} = a_0 T \exp\left(-\frac{\phi \Delta G^*_{hom}}{kT}\right) \exp\left(-\frac{Q}{RT}\right) \qquad (2)$$

$$\dot{x} = b_0 T \exp\left(-\frac{Q}{RT}\right) \left[1 - \exp\left(-\frac{\triangle G_r}{RT}\right)\right] \qquad (3)$$

where $a_0$ (=$5.3 \times 10^{42}$ m$^{-2}$s$^{-1}$K$^{-1}$)[31] and $b_0$ (=$4.5 \times 10^4$ ms$^{-1}$K$^{-1}$)[33] are constants, $T$ is temperature, $\phi$ is a shape factor, $\Delta G^*_{hom}$ is the activation energy for homogeneous nucleation, $Q$ is the activation energy for growth, $k$ is the Boltzmann constant, $R$ is the molar gas constant, and $\Delta G_r$ is the free energy change of the reaction. We adopt $Q = 404$ kJ/mol for the olivine-wadsleyite transition[33]. The values of $\Delta G^*_{hom}$ and $\Delta G_r$ were calculated from the thermodynamic data of the olivine-wadsleyite transition[34] and the olivine-wadsleyite phase boundary[14] (see Rubie et al.[31] for the details). The formation of the lenticular packets filled with nanocrystalline olivine/wadsleyite is effective when

$K$ is high. The temperature dependence of $K$ is shown in Fig. 8. Calculated values of $K$ at 13 GPa using a reported value of $\phi = 6 \times 10^{-4}$ for nickel olivine[31] are about eight orders of magnitude lower than the values of $K$ from the previous experiments on silicate olivine[35,36]. An assumption of $\phi = 1 \times 10^{-4}$ well reproduce the experimental data at temperatures lower than 1100 K. Deviations of experimental data from the calculations at temperatures of ~1150 K might be due to the difficulty of measurements on elevated nucleation rates. Figure 8 shows that GB nucleation of wadsleyite is accelerated with increasing temperatures. At 13 GPa, however, the GB nucleation rate turns to decrease at temperatures above ~1350 K in the case of $\phi = 1 \times 10^{-4}$ because of the lack of overpressure which induces the nucleation of wadsleyite (i.e., the olivine+wadsleyite coexisting field: Fig. 1). As microcracking would be ineffective at temperatures above 1200 K, indicated by the very low AE activities (Fig. 2b), the maximum transition rate on GBs in the semi-brittle regime should be realized at temperatures of 1100–1160 K under these pressures, consistent with the occurrence of throughgoing faulting only in the runs around this temperature.

The elastic energy stored in a viscoelastic material may be spontaneously released at the seismic strain rate by self-localizing thermal runaway[10]. A theoretical study shows the possibility of frictional melting during the rupture of the 1994 Bolivian earthquake (depth = 637 km)[37]. Such deep-focus earthquakes associating ruptures are considered to be nucleated by transformational faulting within an MOW[38]. The critical strain rate $\dot{\varepsilon}_c$, which is required for the adiabatic instability, is expressed as follows[39]:

$$\dot{\varepsilon}_c \sim \frac{h \rho c_p \kappa R T^2}{\sigma L^2 \left(E^* + PV^*\right)} \qquad (4)$$

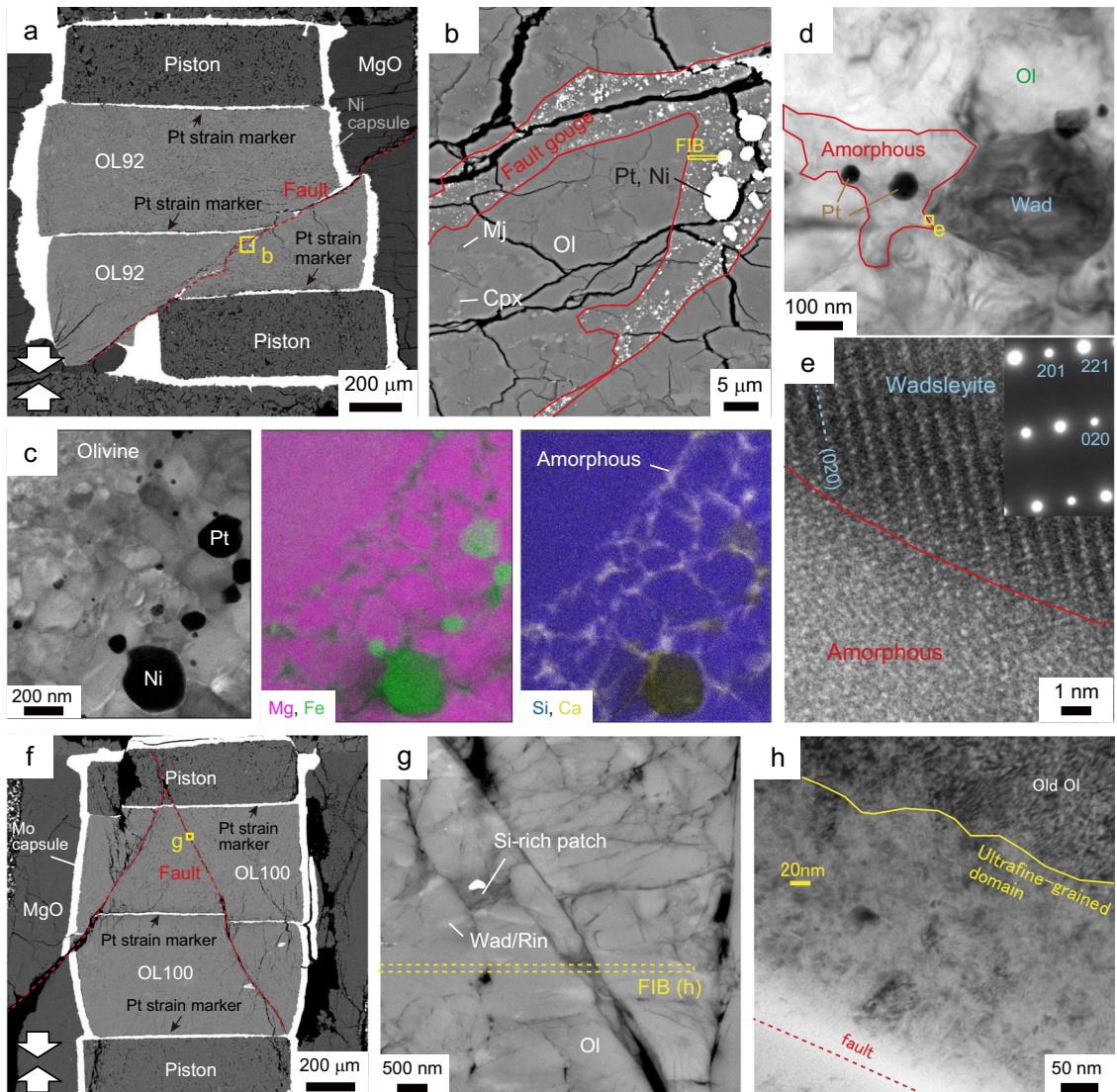

**Fig. 5 | Microstructures in the OL92 and the OL100 samples faulted at 1160 K.**
**a–e**: M2676; **f–h**: M3100. **a** Backscattered electron (BSE) image of the recovered
sample (OL92) faulted at 1160 K. The throughgoing fault is highlighted by a red
dashed line. (**b**) Magnified image of the fault (located in **a**). Ol olivine, Cpx clin-
opyroxene, Mj majorite. **c** Scanning transmission electron microscope image and
element maps showing the gouge layer in the fault (pink: magnesium; green: iron;
blue: silicon; yellow: calcium). Platinum-iron and nickel-iron alloy (Ni) blobs are
distributed along the network of the amorphous phase. The focused-ion beam (FIB)
foil was prepared from the area shown in **b** (yellow square). **d** Transmission electron

microscope (TEM) image of a wadsleyite (Wad) next to an amorphous pocket in the
fault gouge. **e** High-resolution TEM image showing the boundary between the
wadsleyite grain and the amorphous pocket. **f** BSE image of the OL100 sample
faulted at 1160 K. **g** Magnified view of the fault gouge along the fault. Medium gray:
olivine; light gray: wadsleyite or ringwoodite (Rin); dark gray: silicon-rich patch.
**h** TEM image of an ultrafine-grained domain in the gouge zone. Typical grain size is
~20 nm. Many dislocations are observed in the old olivine grain forming the
wall rock.

where $h$ is work hardening coefficient (=1), $\rho$ is density (=3.6 g cm$^{-3}$
at 13 GPa)[40], $c_p$ is specific heat (= 817 J kg$^{-1}$K$^{-1}$)[41], $\kappa$ is thermal diffu-
sivity (= 0.7 mm$^2$ s$^{-1}$)[42], $\sigma$ is stress, $L$ is sample size (=1.2 mm in length
for the present study), $E^*$ is the activation energy for deformation
(=502 kJ mol$^{-1}$ for the Peierls creep)[16], $P$ is pressure, and $V^*$ is the
activation volume for deformation (=30 cm$^3$ mol$^{-1}$ for the Peierls
creep)[15]. Substituting these parameters of olivine for Eq. (4), we
obtain a critical strain rate of $1.2 \times 10^{-2}$ s$^{-1}$ at a differential stress of
1.5 GPa to initiate adiabatic shear heating at the background tem-
perature of 1160 K (Fig. 9a). The dynamic strain rate of the gouge
layer (thickness $L_d$ =3–20 μm) may reach ~0.1–27 s$^{-1}$ during the
throughgoing faulting of ~10 s, as estimated from the stroke sensors
of the anvils in M2676, M3100, and M3425, under the assumption
that strain is localized with a factor of $L/L_d$. Thus, the adiabatic
instability can be initiated by the formation of a nanocrystalline
olivine layer (due to linking-up of the lenticular packets: Fig. 9b)

followed by shear localization accelerated by diffusion creep[43] or
superplasticity of olivine[44] (Fig. 9a: see Methods for the details of
calculations). On the other hand, diffusion creep of nanocrystalline
wadsleyite[45] with a grain size of 20 nm does not account for the
initiation of adiabatic instability due to slower diffusivities of silicon
in high-pressure polymorphs of olivine[46]. Although the flow law for
superplasticity of wadsleyite has not been reported, its effect on
shear localization might be less important than that of olivine
because of the lower diffusivity of silicon (i.e., the rate-limiting
process of superplasticity) in wadsleyite[46].

It has been reported that the latent heat release associated with
phase transition enhances deep-focus earthquake activity[47]. We ther-
modynamically estimated the latent heat release[48] and evaluated its
effect on the relations between strain rate and yield strength for
nanocrystalline olivine and high-pressure phases. A net temperature
increment $\Delta T$ by the latent heat release across the phase transition is

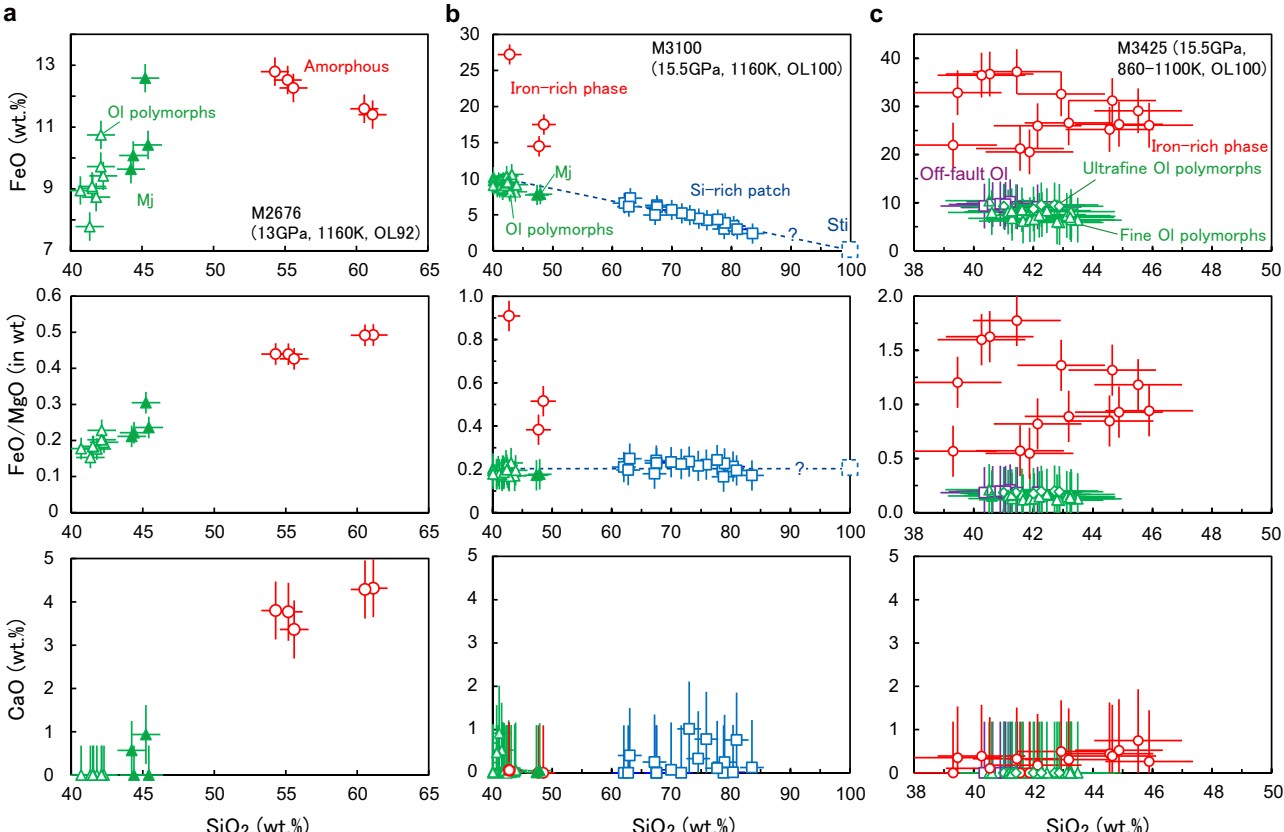

**Fig. 6 | Chemical compositions of the phases obtained with a scanning transmission electron microscope equipped with an energy dispersive X-ray detector (in wt.%).** **a** Chemical compositions of olivine polymorphs (open triangles), majorite (Mj: solid triangles), and amorphous phase (circles) in the fault gouge observed in M2676 sample. **b** Chemical compositions of olivine polymorphs (open triangles), majorite (solid triangles), silicon-rich patches (Si-rich: squares), and the iron-rich phase (circles) in the gouge layer developed in the M3100 sample (e.g., Fig. 5 g). The dashed line is the mixing line between olivine polymorphs and stishovoite (Sti). **c** Chemical compositions of the iron-rich phase (circles), fine-grained (triangles) and ultrafine-grained olivine polymorphs (diamonds) in the fault gouge observed in the M3425 sample, respectively. Squares represent the chemical composition of off-fault olivine grains. The error bars correspond to the errors in the counts of characteristic X-rays.

given as[48]:

$$\Delta T \sim \frac{-\left(T \Delta S_r + \triangle G_v\right)}{C_p} \qquad (5)$$

where $\Delta S_r$ is the entropy change of reaction, $\Delta G_v$ is the free energy change of reaction, and $C_p$ is isobaric heat capacity. $\Delta G_v$ is approximated as $\Delta P \cdot \Delta V / V_O$, where $\Delta P$ is the overpressure from the equilibrium and $\Delta V / V_O$ is the fractional volume change[21]. Equations (4) and (5) show that phase transition of metastable olivine at higher pressures is effective in initiating adiabatic instability due to the positive temperature dependence of strain rate. For the olivine-wadsleyite transition at 1160 K, $\Delta T$ is ~60 and ~110 K for the confining pressure of 13 and 15.5 GPa, respectively. Assuming that nucleation of wadsleyite proceeds much faster than the conductive heat transfer[48], strain rate can reach the critical value for the adiabatic instability at a stress level of ~0.5 GPa, which is lower than that without latent heat release (~1 GPa: Fig. 9a), suggesting that the heat release would enhance the instability. Nevertheless, considering that theoretical studies predict high stress magnitudes up to ~2 GPa[49] due to the accumulation of strain in the deep seismic zone[50], deep-focus earthquakes triggered by the adiabatic instability may occur even with small degrees of latent heat release upon the partial olivine-wadsleyite transition. The effect of latent heat release on the initiation of adiabatic instability is important only when the grain size of olivine in the gouge is larger and/or superheating by the direct transition of metastable olivine to

ringwoodite ($\Delta T$ ~200 K at 20 GPa)[48] is expected. An increase in $\Delta T$ to ~200 K would induce further weakening of olivine/ringwoodite aggregates (i.e., two orders of magnitude increase in strain rate) and adiabatic instability afterwards, whereas strain rate increase by an order of magnitude is expected with $\Delta T$ ~60 K (Fig. 9a).

Seismic imaging of the deep mantle demonstrates the presence of the MOW, accompanied by deep-focus earthquakes, as a low-velocity zone with a thickness of tens of kilometers inside the cold slab[51,52]. The present study suggests that the deep-focus earthquakes nucleate when the temperature of the MOW reaches those close to 1100–1160 K at pressures of the mantle transition region. Here the nanocrystallization of olivine occurs upon the phase transition (i.e., formation and linking-up of the lenticular packets filled with nanocrystalline olivine/wadsleyite) followed by throughgoing faulting due to shear instability (Fig. 9b), although this critical temperature may be lower on the geological time scale (Fig. 8). In fact, the observed deep-focus earthquakes are reported to be located along an isotherm of ~1000 K in deep slabs[53] and can propagate outside the MOW[38]. Numerical studies based on experimental kinetic data show that metastable olivine in cold slabs probably persists to depths of ~630 km, below which ringwoodite is the major phase down to ~700 km[36,54], where it breaks down to bridgmanite and ferropericlase. The shear zones filled with nanocrystalline olivine required for throughgoing faulting would not be preserved in subducted slabs at depths beyond this kinetic boundary of ~630 km, consistent with the abrupt decrease in seismicity at depths below ~600 km[2].

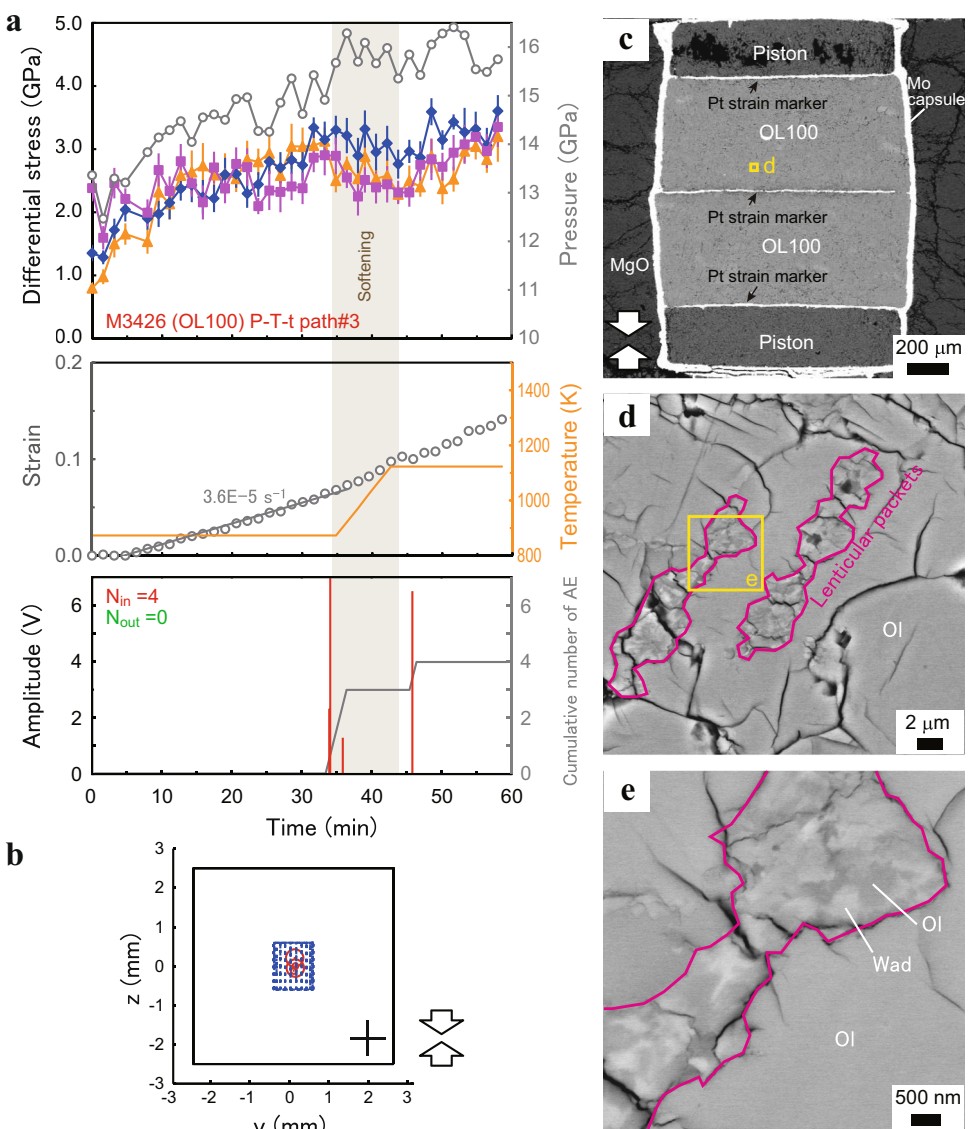

**Fig. 7 | Mechanical/acoustic records and microstructures in the OL100 sample deformed at 860–1110 K without occurrence of throughgoing faulting (M3426). a** Mechanical data (pressure, stress, and strain). The error bars represent the uncertainties in stress. **b** Two-dimensional views of acoustic emission (AE) hypocenters. Symbols and lines are the same as those in Fig. 2. **c** Backscattered electron (BSE) image of the recovered sample. **d**, **e** Magnified views of lenticular packets filled with nanocrystalline olivine (dark gray) and wadsleyite (light gray). Ol olivine, Wad wadsleyite.

## Methods

### Starting material and deformation experiments

The starting material and the procedure for synthesis of OL100 and OL92 samples are the same as those in our previous study[55]. Gem-quality crystals of olivine (from San Carlos, USA) and orthoenstatite (from Kilosa, Tanzania) having no inclusions were finely ground for 2 h in an agate mortar. The fine-grained powders of olivine and orthoenstatite were weighed in the desired proportion (i.e., 92 wt.% olivine and 8 wt.% orthoenstatite) and then mechanically mixed/ground for 3 h in an agate mortar. The mechanical mixing process is essential to obtain OL92 samples with homogeneously dispersed olivine and orthoenstatite grains. The fine-grained powder of olivine or olivine-orthoenstatite was put into a nickel capsule (inner diameter: 8 mm; length: 11 mm). The powders were sintered at 4 GPa and 1073 K for 1.5 hour using a Kawai-type multi-anvil apparatus at Ehime University. The average grain size of olivine in the sintered sample is ~15 μm. The sintered sample was core-drilled to a rod with a diameter of 1 mm and a length of 0.6 mm. Melting of the OL100 and OL92 samples is used for estimation of a peak temperature during faulting (e.g., OL100: >

2500 K at 15.5 GPa; OL92: ~2170 K at 13 GPa)[24,26]. The mechanical behavior of the OL92 sample was found to be almost the same as that of the OL100 sample because 92 wt.% of olivine forms the load-bearing framework[56], suggesting that the present discussion and conclusions are independent of this compositional difference.

We conducted deformation experiments on the OL100 and OL92 samples using a deformation-DIA apparatus[57] combined with the large-volume MA-6-6 system at the BL04B1 beamline of SPring-8. A semi-sintered cube of cobalt-doped magnesia with an edge length of 5 mm was used as the pressure medium, which was surrounded by five tungsten carbide anvils (with a truncated edge length of 3 mm) and an X-ray transparent cubic boron nitride (cBN) (or sintered diamond) anvil placed on the down-stream side. Two of the four sliding blocks on the down-stream side have a conical X-ray path with a maximum 2θ angle of ~10°. Two core-drilled samples were placed along the compression direction, and the samples were separated by a 10 μm-thick platinum strain marker. The cored samples, which were sandwiched in between two hard-alumina pistons with a diameter of 1 mm and a length of 0.3 mm, were placed in a nickel (or molybdenum) capsule

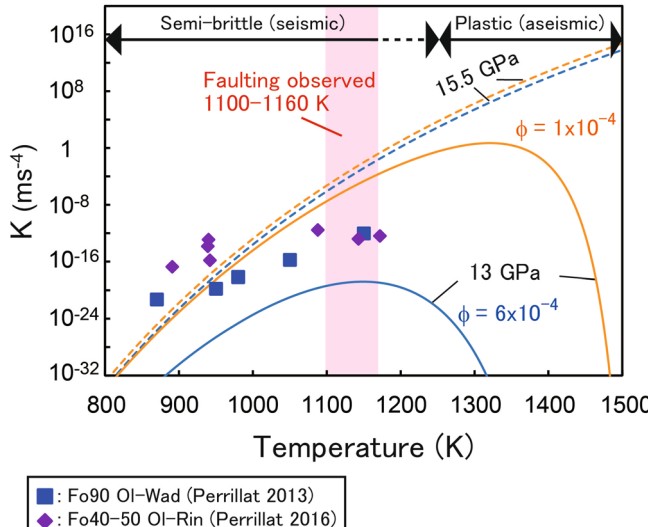

**Fig. 8 | Numerical model results for grain boundary nucleation of wadsleyite.** Temperature dependence of the rate constant $K$ at 13 (solid curves) and 15.5 GPa (dashed). Two shape factor values ($\phi = 6 \times 10^{-4}$: Rubie et al.[31]; $1 \times 10^{-4}$: this study) are considered for blue and orange curves, respectively. Symbols represent the values calculated from the reported experimental data (squares: olivine-wadsleyite transition in $Fo_{90}$ olivine[36]; diamonds: olivine-ringwoodite transition in $Fo_{40-50}$ olivine[35]). Semi-brittle flow associating acoustic emissions dominates the bulk deformation at temperatures below 1160 K. On the other hand, aseismic plastic deformation is effective at higher temperatures. The pale-red area represents the range of temperatures at which throughgoing faulting occurred.

(outer diameter: 1.1 mm; wall thickness: 0.05 mm; length: 1.8 mm) (Supplementary Fig. 5a). The hard-alumina pistons were coated with platinum (thickness of a few hundred nanometers) to avoid chemical reaction between the pistons and the samples. Each piece of sample/piston was separated by a platinum strain-marker having a thickness of 10 μm. The nickel/molybdenum capsule surrounded by a magnesia (MgO) sleeve was inserted in the inner bore of a boron nitride composite heater ($TiB_2 + BN + AlN$), which was placed in the $LaCrO_3$ thermal insulator. Small portions of the pressure medium and the $LaCrO_3$ thermal insulator along the X-ray path (a diameter of 1.4 mm) were replaced by an amorphous boron powder cemented with epoxy resin (at a ratio of 4:1 by wt.) to maximize the intensity of the diffracted X-rays.

The cell assembly was first pressurized hydrostatically up to 0.6 MN in the main-ram load and temperature was raised to 1470 K at a rate of ~50 K/min. Typically, ~20 small events (AE maximum amplitude <1 V) were detected from the samples during the cold-compression stage. A thermocouple was not used in the deformation experiments because the thermocouple damaged the truncated surface of cBN (or sintered diamond) anvils at high temperatures. Temperature was estimated from the relationship between temperature and power established for a $W_{97}Re_3$-$W_{75}Re_{25}$ thermocouple placed along one of the diagonal directions of the cell assembly (Supplementary Fig. 5b). Temperature around the outer wall of the nickel/molybdenum capsule samples was monitored in the series of calibration experiments. The uncertainties in temperature estimate in this study is ±60 K. The vertical temperature difference between the central part and the edge of the sample is less than 50 K.

The P-T-t paths of all deformation experiments (excepting for run M2472) and their relations to the olivine-wadsleyite-ringwoodite equilibria phase boundaries are shown in Fig. 1. We adopted two kinds of P-T-t paths before the deformation stage: i) path#1; the sample was annealed at 1470 K and pressures below 12 GPa in the olivine stability field for 20 minutes to decrease the density of defects and

microcracks created during the cold-compression stage (M2600-2959 and M3101). ii) Path#2 (and #3); the annealing procedure of path#1 was followed by hydrostatic pressurization (up to 0.85−1 MN) at 860 K to avoid the pressure-induced transitions of olivine before the deformation stage (Path#2: M3078, M3100 and M3423; Path#3: M3424-3426). In all of these runs, the sample was subsequently deformed in the uniaxial compression geometry at a constant stroke rate (2−4 μm/min) by advancing the upper and lower first-stage anvils. Temperature was kept constant (960−1250 K) during the deformation stage in the path#1 and #2 runs. In the path#3 runs, temperature was increased from 860 to 1160 K with a constant rate of ~0.5 K·$s^{-1}$ during the later deformation stage (strains > 0.05). The ratios of temperature-ramping rate and strain rate in the path#3 runs ($\dot{T}/\dot{\varepsilon}$ =1.0−1.4 × $10^4$ K) are comparable to those expected values for natural subduction zones[58]. Pressure was gradually increased above the phase boundary between the olivine stability field and the olivine+wadsleyite coexisting field[14] during the early deformation stage in the path#1 run. The main-ram load (+0.1 to +0.3 MN/hour) was increased during the deformation stage to keep the pressure within the wadsleyite stability field at higher strains, as some pressure drop with increasing strain was observed in four deformation runs (M2880, M2955, M2957 and M2959). A blow-out occurred in the M2472 run.

### X-ray data analysis

Two-dimensional radial diffraction patterns of monochromatic X-rays (energy 60 keV) were taken by using a MAR-CCD camera with 8−9 min of exposure time. To increase the time resolution of measurements, a WidePix 5×5 imaging detector with CdTe sensor (Advacam Co.) was adopted in the last four runs (M3423-3426), instead of the MAR-CCD camera. The exposure time for a 2-D diffraction pattern was shortened down to 70 s with the WidePix detector. The half-circle radial diffraction patterns were subdivided into 18 sectors having a uniform azimuth angle of 10°. Peak positions of four diffraction peaks of olivine ($hkl$ = 021, 101, 130, and 131: Supplementary Fig. 5c) in each subdivided sector were semi-automatically determined in pressure and stress measurements using software (IPAnalyzer and PDIndexer)[59]. Pressure was determined from the unit-cell volume, which can be calculated from the d-spacing under hydrostatic conditions, based on an equation of state of olivine[60]. Differential stress was estimated from the azimuth angle dependency of the d-spacing[61] using olivine single-crystal elastic constants[40,62]. The uncertainty on the stress resulting from the accuracy of the measurements is defined as the 1-sigma in the least square fit of Singh's theoretical curve of the azimuth angle dependency of d-spacing. A diffraction peak of wadsleyite ($hkl$ =240: Supplementary Fig. 5c) was used for in situ identification of wadsleyite. A comparison of the diffraction dataset and microstructures of the recovered samples showed that the peak was detectable when the volume fraction of wadsleyite was higher than 5 vol.%.

The strain $\varepsilon$ of a deforming sample was evaluated from the distance between two platinum strain-markers placed between the sample and an alumina piston, which was monitored by in situ monochromatic X-ray radiography. Each radiograph image (30 and 2 s of exposure time when a MAR-CCD camera and a WidePix 5 × 5 detector were used, respectively) was taken just before the acquisition of the 2-D X-ray diffraction pattern. Natural strain (i.e., $\varepsilon = -ln\ (l/l_0)$, where $l_0$ is the initial length of the sample; $l$: the length of the sample during deformation) was adopted to evaluate the sample strain. The uncertainty in the strain, mainly due to the shape of strain marker, is within 10%. A sudden splitting of a platinum strain marker monitored by in situ observations was interpreted as the occurrence of throughgoing faulting[55], this was confirmed in the recovered sample.

### AE monitoring and data processing

We monitored AEs as a proxy of fracture propagation at high pressure and temperature. AE monitoring was combined with in situ X-ray

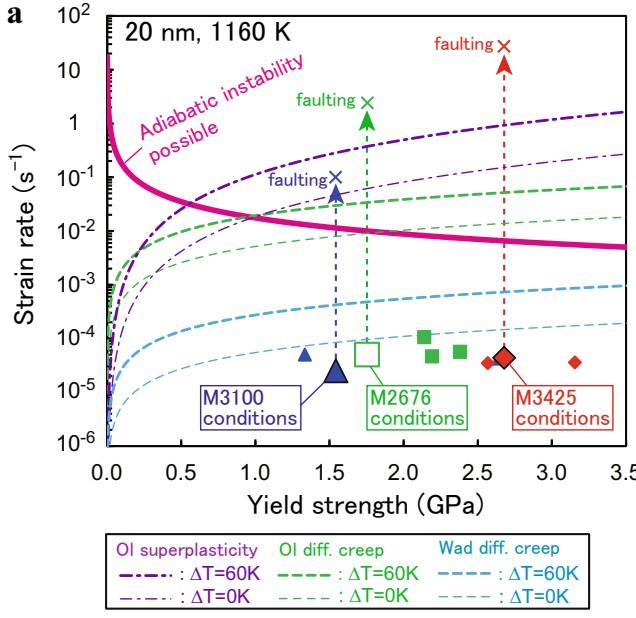

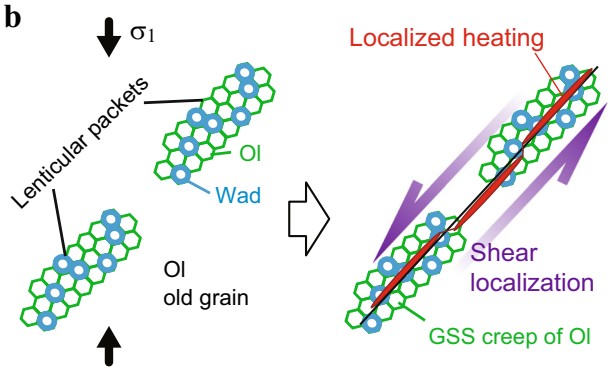

**Fig. 9 | Occurrence of transformational faulting under conditions of the mantle transition zone. a** Conditions for localized heating in metastable olivine at pressures of 13–15.5 GPa for the background temperature of 1160 K and grain size of 20 nm. The thick-solid curve (pink) shows the critical strain rate for adiabatic instability. Long-dashed curves represent the creep strength of ultrafine-grained olivine (green) and wadsleyite (blue) without ($\Delta T = 0$ K: thin) and with (=60 K: thick) the effect of superheating (Eq. 5). Dot-dashed curves are the estimates for superplasticity of olivine. Our experimental results at 1100–1160 K are shown by symbols (the same as those in Fig. 2). Crosses show the estimated strain rate at the timing of throughgoing faulting (see text). **b** Schematic illustration of the present transformational faulting model. Formation and linking-up of lenticular packets filled with nanocrystalline olivine (Ol)/wadsleyite (Wad) induce shear localization associating grain-size-sensitive (GSS) creep and localized heating.

diffraction/imaging analysis during the stage of deformation. The setup and procedure for AE monitoring are based on those in previous studies[20,55]. AE events were collected with six piezoceramic lead-zirconate titanate (PZT) transducers that were attached to the sides of the second-stage tungsten carbide anvils (Supplementary Fig. 5d, e). The transducers were not placed on the rear side of the second-stage anvils so as not to decrease the bottom area of the second-stage anvils, as this linearly correlates with the upper limit of the main-ram load allowed for in experiments. Each transducer, having a size of 4 mm in diameter and a thickness of ~0.5 mm (resonant frequency of ~4 MHz) was electrically isolated from the second-stage anvils by a mirror-polished alumina wear plate (0.5 mm thick). The raw acoustic signals were amplified by ultra-small pre-amplifiers (20 dB gain), which were located near the transducers (distance of ~140 mm from a transducer)

and the amplified signals were further amplified by low-noise 30 dB pre-amplifiers. Use of the ultra-small pre-amplifiers significantly reduced the electrical noise in the synchrotron laboratory environment, resulting in improvement of the signal-to-noise ratio (i.e., S/N) of waveforms (Supplementary Fig. 5f). The doubly-amplified signals having a maximum amplitude of >80 mV were recorded using a high-speed multi-channel waveform recording system at a sampling rate of 100 MHz (16 bit, 8192 samples). The P-wave arrival time was automatically picked based on the Akaike Information Criterion approach[20,63] for each waveform. To minimize the possible errors coming from the automatic picking, the P-wave arrival time was also manually checked and corrected. The P-wave arrival time was then corrected by substituting the travel time of the P-wave through a WC anvil (= $D/v_{pa}$, where $D$ is the distance between the 2nd-stage anvil top and the transducer; $v_{pa}$, P-wave velocity in the anvil), prior to calculation of the hypocenter location. P-wave velocity of the WC anvil (6.6 km/s) is assumed to be constant throughout the experiments. Following the definition of the body-wave earthquake magnitude, the relative magnitude ($M_{ae}$) of an AE event is determined according to the logarithm of the maximum amplitude of the AE signal ($V_{max}$) detected on a chosen channel[64], that is, $M_{ae} = log_{10}V_{max}$. Note that the values of $M_{ae}$ obtained in this way ought to be considered only as first-order approximations of the relative values[64]. To evaluate cumulative energy release via AE radiations in each run, AE energy was obtained by integrating the absolute amplitude value of a given waveform.

Hypocenter locations were calculated from the difference in the arrival time between the 1st and $i$-th transducers ($t_i − t_1$) as a function of the true hypocenter location $X'$ ($x+dx$, $y+dy$, $z+dz$), the tentative hypocenter location $X$ ($x, y, z$) and the location of the $i$-th transducer $P_i$ ($a_i, b_i, c_i$) ($i = 1, 2\cdots, 6$):

$$\left(\frac{x - a_i}{R_i} - \frac{x - a_1}{R_1}\right)dx + \left(\frac{y - b_i}{R_i} - \frac{y - b_1}{R_1}\right)dy$$
$$+ \left(\frac{z - c_i}{R_i} - \frac{z - c_1}{R_1}\right)dz = v_p\left(t_i - t_1\right) - R_i + R_1 \quad (6)$$

where $v_p$ is the averaged P-wave velocity in the sample, and $R_i$ is the distance between $X$ and $P_i$. In the calculation, we obtained the best location of $X'$ that gives the minimum residual of Eq. (6) under the assumption that the velocity structure of the cell assembly was homogeneous and the averaged P-wave velocity was equal to that for olivine (e.g., 9.1 km·s⁻¹ at 14 GPa and 1160 K)[60] in the cell assembly. In this study, the location uncertainty is defined as the root mean square of the right side of Eq. (6), that is:

$$error = \sqrt{\frac{1}{5}\sum_{i=2}^{6}\left\{v_p\left(t_i - t_1\right) - R_i + R_1\right\}^2} \quad (7)$$

To demonstrate that AE hypocenters radiated from the samples were properly determined in our experiments, we conducted an offline test on cold compression of quartz beads[65] using a deformation-DIA apparatus at Ehime University. Quartz is known to display a brittle behavior associating a large number of AEs due to grain crushing and porosity loss during compression[66]. The cell assembly used for the cold-compression experiment is the same as that used for the in situ deformation experiments. Quartz beads with a diameter of 200 μm or less were packed into a nickel capsule (Supplementary Fig. 5a). The quartz sample was sandwiched in between two hard-alumina pistons with a length of 0.3 mm in the nickel capsule. The cell assembly was pressurized hydrostatically to 0.6 MN in the main-ram load. AE events during cold compression were collected with the six PZT transducers that were attached to the sides of the second-stage tungsten carbide anvils (Supplementary Fig. 5d, e). Hypocenters of the AEs are shown in Supplementary Fig. 6. Most of the high amplitude AE events were recorded during the early stage of cold compression, suggesting that

the AEs are related to crushing of quartz grains accompanied by associated pore collapse at low pressures[65]. Hypocenters of intense AEs (the maximum amplitude >1 V) are located within the quartz sample, showing that the determined locations of AE hypocenters are plausible.

## Micro- and nanostructural observations and water content measurement

The recovered cell assembly was cut with a low-speed saw to obtain the sectioned plane parallel to both the directions of axial compression and the incident X-ray. The sectioned samples were impregnated with epoxy under a vacuum and then polished using alumina powder followed by colloidal silica suspension. Backscattered electron (BSE) images of the polished surface of the recovered samples were taken using a JEOL JSM-7000F field-emission scanning electron microscope (FE-SEM) equipped with an EDX detector. The chemical compositions of minerals were measured under the operating conditions of 15 kV accelerating voltage and 1 nA probe current. Submicron-scale microstructures around the faults in three samples (M2676, M3100 and M3425) were also examined by using a field-emission scanning TEM (STEM) JEOL-2100F equipped with a JEOL EDX detector system at Ehime University operated at 200 kV accelerating voltage. The absence of electron damage was carefully ensured during the operation. Thin foils (thickness of ~100 nm) were prepared using a focused ion beam system (FEI Scios) with an accelerating voltage of 30 kV. Characteristic X-rays detected by the STEM/EDX detector were used to analyze the compositions, where the EDX spectra were acquired for 30 seconds for each analysis. The cores of each crystal having a diameter larger than 100 nm (except for cases of ultrafine grains/patches) were selected for quantitative analysis with a probe size of ~2 nm to avoid the contamination of spectra from the neighboring grains. The chemical compositions were determined from the STEM/EDX data using the Cliff-Lorimer equation[67]. We used the composition of olivine grains obtained with the SEM/EDX detector as an internal standard for determining Cliff-Lorimer k-factors. Chemical compositions are recalculated as 100 wt.% anhydrous. The uncertainty of the calculated chemical composition reflects the error in the counts of characteristic X-rays.

Unpolarized infrared spectra of the recovered samples were measured using a JASCO IRT-5200EUO Fourier-transform infrared spectrometer (FTIR) with a mid-infrared light source, a KBr beam splitter and a mercury cadmium telluride detector. The doubly-polished thin sections with thickness of 40–90 μm were placed on a $BaF_2$ plate in dried air. An aperture size of $50 \times 50$ μm$^2$ was used for all of the measurements. At least five spectra were obtained from each section with 128 integrated scans with 4 cm$^{-1}$ resolution. The spectra were normalized by the thickness of the sections. The water content in olivine was determined by integrating infrared absorption spectra from 3100 to 3640 cm$^{-1}$ on the basis of the extinction coefficient calibration by Paterson[68] using an orientation factor of 1/3 and the extinction coefficients for olivine under the assumption that the water contents dissolved in minor phases are ignorable.

## Calculations of creep strength of the gouge

At low temperatures and high differential stresses corresponding to the conditions inside deep slabs, dislocation glide is operative but dislocation climb is inactive (i.e., the Peierls mechanism) and thus the creep of olivine is known to be expressed by the following exponential flow law:

$$\dot{\varepsilon} = A_{pei} \, exp\left[ -\frac{E^* + PV^*}{RT}\left(1 - \frac{\sigma}{\sigma_p}\right)^2 \right] \qquad (8)$$

where $A_{pei}$ is the pre-exponential constant, $\dot{\varepsilon}$ is strain rate, and $\sigma_p$ is the Peierls stress[15,16]. The critical strain rate for the adiabatic instability

(thick-solid curve in Fig. 9a) was calculated from Eq. (4) under the assumption that deformation of the wall rock is controlled by the Peierls creep.

The steady-state creep strengths of ultrafine-grained rocks are more or less controlled by a diffusive process (i.e., GSS creep) even at lower temperatures and thus the plastic deformation is described as the power-law equation:

$$\dot{\varepsilon} = A_{pow}\frac{\sigma^n}{G^p}f_w^r \, exp\left(-\frac{E^* + PV^*}{RT}\right) \qquad (9)$$

where $A_{pow}$ is a constant, $n$ is the stress exponent, $G$ is grain size, $p$ is the grain size exponent, $f_w$ is water fugacity, and $r$ is the water fugacity exponent. In the case of diffusion creep with $n = 1$, Eq. (9) is rewritten as follows:

$$\dot{\varepsilon} = 13.3\frac{\sigma}{G^2}\left[D_{0,l}\,exp\left(-\frac{H_l^*}{RT}\right) + \frac{\pi\delta}{G}D_{0,gb}\,exp\left(-\frac{H_{gb}^*}{RT}\right)\right]\frac{\Omega}{RT} \qquad (10)$$

where $D_O$ is the pre-exponential factor ($l$: lattice diffusion; $gb$: GB diffusion), $H^*$ is the activation enthalpy for diffusion; $\delta$ is width of the GB, the $\Omega$ is the molar volume. In Fig. 9a, three cases are considered for the deformation of the gouge layer with grain size of 20 nm: (i) superplasticity of olivine[44], (ii) diffusion creep of olivine[43], and (iii) diffusion creep of wadsleyite[45]. Parameters used for the calculations are summarized in Supplementary Table 1. The activation volume of 4 cm$^3$/mol (GB diffusion of silicon in olivine)[43] was adopted to the superplasticity of olivine.

## Data availability

All data supporting the findings of this study are available within the paper and its Supplementary Information file. The raw X-ray data files are freely available from our webpage (http://earth.sci.ehime-u.ac.jp/~ohuchi/download/m2600.zip). See Source data for the links for other runs. Other raw data are available from the corresponding author upon request. Source data are provided with this paper.

## Code availability

We used IPAnalyzer/PDindexer code (provided by Dr. Seto), which is available at https://github.com/seto77.

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

## Acknowledgements

We thank X. Lei and M. Nishi for their technical support for AE monitoring and high-speed X-ray imaging, and M.C. Wood for making corrections in the language in the manuscript. This research was conducted under the approval of SPring-8 (Nos. 2018B1052, 2019A1731, 2019B1115, and 2021B1240) and supported by JSPS KAKENHI (Nos. 16H04077, 18K18788, and 19H00722).

## Author contributions

T.O. conceived the idea, conducted experiments, and wrote the manuscript. T.S. contributed to TEM foil fabrication. Y.H., Y.T., K.M., and T.I. contributed to in situ experiments and discussion.

## Competing interests

The authors declare no competing interests.
