## [Peer Review File · Nature Communications]

In situ X-ray and acoustic observations of deep seismic faulting upon phase transitions in olivineREVIEWER COMMENTS

Reviewer #1 (Remarks to the Author):

Review of In situ X-ray and acoustic observations of deep seismic faulting upon phase transitions in mantle olivine by T. Ohuchi et al.

The paper by Ohushi et al. presents new experimental results on transformational faulting in silicate olivine and its relevance for the occurrence of Deep Focus Earthquakes (DFEs). This topic has been extensively debated in the last 30y since the pioneering work of Burnley et al. 1991 on the germanium olivine analogue. As emphasized by the authors, due to experimental limitations, only limited work has been performed until now on the silicate olivine (Green et al. 1990; Officer and Secco 2020) both performed under 'uncontrolled differential stress' in multi-anvil apparatus. Nevertheless, some experiments have already been performed by the authors on the silicate olivine, albeit at lower pressures (Ohushi et al. 2017, Nat. Geosc.), a study in which they argued that intermediate depth EQs could be triggered by localized heating. There is doubt in my mind that the paper presents cutting-edge results, which should be published in a high impact paper such as Nature Communications. However, in its current version, I found the manuscript hard to follow, and filled with a number of flaws/inconsistencies that need to be thoroughly addressed before publication. The paper also seems to miss a proper conclusion.

Major comments:

1. AE location/Magnitudes/Polarities

My first comment is on AE location. How was the error bar on location inferred? Due to the geometry of the AE sensor array, a calibration procedure must have been performed in order to test the location procedure. Indeed because of their set-up where sensors are glued on the anvil sides, there could be important effects of reflection on the side and back walls of the anvils. A simple calibration, for instance by performing cold compression of fine quartz aggregates as in Gasc et al. 2011, seems to be needed. In the absence of calibration, the actual location given in Figure 3c & extended figures 1-2 have little meaning. It might be that they are not needed in the final manuscript, as they do not reveal much anyways.

In a similar manner, the authors argue for a double couple source for these AEs (I98-102). First, one should point that a DC source does not correspond to a simple mix of positive and negative polarities. Tensile or compressive cracking can also present, in the present of a limited shear component, a mix of first motion polarities. In such a way, the authors would have to demonstrate the actual DC nature of AE sources by performing a moment tensor inversion, such as was performed by Schubnel al. 2013 (Fig. 3). With their set-up, I believe it would be close to impossible, again due to the possible reflection on the side of the anvils, which may lead to polarity inversion upon reflection.

Finally, there is seem to be a problem when calculating AE magnitudes, which should be proportional to V^2 (energy) and not V .

2. Microstructure of faulted specimen

The microstructure of three specimens has been investigated because they faulted and present fault planes. However, these microstructures are particularly difficult to interpret, since in the three experiments, faulting was followed by a blow out. So the questions are: 1. How much pressure / temperature was lost during these blow-outs? 2. How much of the microstructure reported in Fig. 4-6 is related to faulting and/or to blow outs? For instance, the displacement reported of several hundreds of microns (1140&1158) are hardly compatible with that being solely related to the HP-HT faulting. Indeed, a simple relationship exists in seismology between displacement and stress drop, where $\Delta\sigma \approx \mu D/L$, where $\Delta\sigma$ is the stress drop, μ the shear modulus, D the co-seismic displacement and L the fault length. Taking number such as 100GPa for μ , 1mm for L and 250 microns for D , leads to stress drops of >25 GPa, i.e. a value higher than the pressure itself. In consequence, whatever has happened on these experimental faults is certainly mostly related to blowouts and not shear-rupture (maximum co-seismic displacement of approx. 30microns for a complete stress drop at 3GPa differential stress). Accordingly, all the section on the occurrence of partial melting seems dubious and far-fetched, as it relies on improper attribution of a microstructure to the fracture propagation. This section has to be greatly revised, tampered and to the best, put in conditional form.

3. Figures and figure captions

Most of the figures of the paper are very small and hard to read, with too much information, which leads to confusion:

- Figure 1 is hard to read. What is the difference between PTt paths 2 & 3. Some of the PTt path 2 specimens seem to have undergone pressurization at high T, while some not. Please clarify the figure.
- In Figure 2, is it yield (if yes how as it determined) or rather flow strength that you represent and also refer to in Table 1? Please use clearer symbols for faulted samples. What do the red arrows correspond to?
- The vertical grey bars (Max Amplitude V) in Figure 2b represent single experiment, several, single AEs, cumulative - please clarify -, Maybe a plot of cumulative AE energy detected for each experiment at each T would be easier to read?
- Fig3 - dashed box – no Wad240 – is this really useful?
- Following the above comment#2, Fig. 6 is meaningless.

4. Model of the instability

This part of the paper is very interesting, but remains unclear to me.

- First, how does the kinetics the authors infer from their experiments compare to the already published ones for the silicate olivine (see for instance Perrillat et al. 2015 for a compilation)?

- Second, Figure 8 shows the temperature/pressure dependence of the kinetics, in terms of normalized values. But what are the actual values? And again how do they compare to experimental ones and the ones inferred in these experiments.

- L202 – I believe a minus sign is missing in the first exponential term of growth rate

- L209 - I can see how ΔG_r was calculated, but how was ΔG^*_{hom} calculated. Please clarify.

- From eqn 1-3, I am sorry to say I do not understand why/how come there is such a huge difference in the shape of K/K_{max} between 13 and 16GPa. Is this effect of the metastability term ($1-\exp(\Delta G/RT)$) in eqn(2) solely?

- Figure 9a has too much information. What are the colors (blue red & green) and the different symbols (triangles, diamond & stars) referring to? Does each two set of colored curves correspond to the difference of 60°C due to 'superheating' by the reaction itself. How was the superplastic flow law inferred? Please reference in the methods/sup.mat the actual equations for each curve.

- L269 – superheating due to direct transition to ringwoodite – this is very interesting and should be emphasized. How does the critical adiabatic shear heating strain rate compares to your flow laws with a 200°C superheating?

5. Inadequate referencing

In a number of places, the reference numbers are wrong. Please check carefully your reference list:

- L52 – note that Schubnel et al. 2013, Wang et al. 2017 also performed their experiments using a DDIA - Is this considered a conventional apparatus?

- L60 – the reference 8 (Aben et al.2015) seems to be misplaced / irrelevant to the point made here

- L183 – reference 23 does not correspond to pulverization processes.

- I stopped checking after this, sorry...

6. Conclusion

Maybe the major flaw of the paper is its lack of actual conclusion, and one falls short of understanding what the authors want to say. The conclusion is replaced by a last paragraph of discussion (l270-289), which attempts to bridge with the larger scale and reports on b-value variability. How is this really related to the present study, since the authors reported no experimental b-value?

7- Minor comments:

L24 – the occurrence of deep EQs

L93 – ambiguous triple negation (neither, nor, not) – please reword

L124 – Inadequate wording – velocity weakening refers to a rate and state friction concept, where the strength decreases with increasing sliding velocity. Maybe you meant that that the softening due to an increase in temperature also leads to an increase in strain rate/decrease in flow strength?

L126 – comparison of $(T/\epsilon)_{\dot{}}$ with nature – please place this sentence either in the discussion, or in the methods. It seems odd here.

L136 – “and also to quit the run to examine the sample” – I don't understand what you mean? Please reword

Reviewer #2 (Remarks to the Author):

Dear authors,

Thank you for the challenging experimental work, which brings interesting new insights for the understanding of deep ruptures.

There are various issues that prevent publication at this stage, but I would support publication once these are properly addressed. I explain in details in the attached review file. You might consider this review file as long and intense, and I apologize in advance for the large amount of work that I request.

I hope you will understand my comments and consider it useful to improve the paper.

I sign my review, as usual, for the sake of transparency.

Best regards,

Thomas P. Ferrand

Major comments:

NB: *These major comments are complemented by detailed line-by-line comments (from page 9), fully referenced, to help the authors strengthen their manuscript. I apologize if some comments are redundant between the major comments and the line-by-line comments.*

- **Major comment #1: Imprecisions, misconceptions and misleading statements**

Throughout the manuscript, several inaccurate statements can be listed, as well as imprecisions on either observations or interpretations. Furthermore, I report some confusion between observations and interpretations. I describe these in details in the *line-by-line comments*.

Numerous mistakes can be listed throughout the paper, some of which directly induced by a lack of consideration for previous studies (**major comment #2**). First, the depth limit of seismicity within the solid Earth is not 680 km, as deep-focus earthquakes were reported deeper, up to ≈ 750 km depth (Kuge, 2017; Kiser, 2021). Not only olivine, but also other minerals, e.g. pyroxenes (Shi et al., 2017; Xu et al., 2018), have been proposed to trigger seismicity upon destabilization in the MTZ. In their reference to the Peierls creep, the authors favor a mechanism and do not recall the alternative mechanisms, which does not give a good impression (**major comment #3**). Similarly, the authors favour the partial melting interpretation, although contrary to what they claim their observations (and previous works) are mostly consistent with partial crystallization of the melt (**major comment #5**).

Several terms are problematic, such as “melt-like phase” (lines 160-161). I point out that “low-viscosity fluid” (lines 49-50) does not make sense, for several reasons. By definition, “fluid” as an adjective is the opposite of “viscous”; as a noun, it is the opposite of “solid”.

I request that the authors clarify as much as possible everything, especially the methods and the figures. For example, it should be specified that the “mechanical data” consists of the stress state as seen via X-ray diffraction of olivine crystals. There is a critical problem with the electron diffraction patterns in **Fig.4d** and **Fig.5e**, which should not be truncated and should be accompanied with a scale (in nm⁻¹). Also, the “no Wad” purple boxes should be moved below subfigures (just below the time axis) for sake of clarity. Equations and associated descriptions are not presented in the right order. Finally, the authors could shorten several sentences by removing redundant information.

The authors state that brittle failure is “unlikely” at depth, which results from general misconceptions regarding deep earthquakes and their experimental analogs. Contrary to what the authors claim, the actual seismicity is the proof that brittle failure exists at pressures up to ≈ 25 GPa (deep-focus earthquakes) in Earth’s mantle. The fact that a large part of the community is unable to understand how it works is not a reason to state that these ruptures would be “unlikely”. They are not unlikely, otherwise they would not occur so frequently. Nonetheless, it is true that high pressures and/or temperatures do not favor the brittle failure of rocks, which explains the interrogations regarding both deep triggering mechanisms (critical stress distortion generating the mechanical instability) and deep rupture mechanisms (nucleation and dynamic propagation). → See **major comment #3**.

The paper does not look like a revised manuscript. Vocabulary issues should be fixed to avoid confusion, especially if a broad readership is targeted. Grammatical issues should also be verified throughout the text (e.g. missing “the”). At several instances (detailed in the *line-by-line comments*), the paper is hard to follow due to grammar or syntax issues. In some paragraphs, the mixture of conceptual exaggerations and language issues tend to prevent the reader to remain focused.

- **Major comment #2: Lack of consideration for previous works and illegitimate attacks**

There is a serious lack of reference for key studies associated with the development of the techniques used in the present work (e.g. Wang et al., 2004; Gasc et al., 2011). The authors should better acknowledge the studies that inspired the present work. They should recall that the stress calculation technique from the diffraction of high-energy X-rays in the synchrotron comes from the key development reported in several studies (e.g. Uchida et al., 1996; Hilairet et al., 2012; Merkel & Hilairet, 2015), not only Singh et al. (1998). They should properly describe the analysis tools they use: the Multifit/Polydefix framework (Merkel & Hilairet, 2015) or any other protocol. Finally, note that the

provided definition of the AE magnitude is used by previous studies on similar lithologies (e.g. Schubnelet al., 2013; Ferrand et al., 2017; Kita & Ferrand, 2018).

Regarding the research topic, some key studies of the recent literature are ignored. Thus, the discussion on mechanisms is incomplete (**major comment #3**). The overinterpretation favoring partial melting also ignore studies reporting partial crystallization (**major comment #5**). [In addition, this study looks like a high-pressure equivalent of the *TROPICO* project on $Mg_2GeO_4 + 8\% MgGeO_3$, which I proposed to the JSPS in 2017 and 2018 and was rejected 3 times in a row.]

On top of that, I have noticed inappropriate sentences, which artificially reduce the importance of previous works, although the present study is the continuation of these works. The authors state that the analogs used by previous studies may not be adequate but minimize the legitimate reasons why these analogs have been used. In spite of differences, mentioned by numerous studies (e.g. Green & Burnley, 1989; Green et al., 1990; Schubnel et al., 2013; Ferrand & Deldicque, 2021), both Mg_2GeO_4 and $(Mg,Fe)_2SiO_4$ transform to a spinel or spineloid structure with increasing pressure. I highly recommend that the authors read the paper of Riggs & Green (2005) reporting the exact same microstructures in Mg_2GeO_4 as Officer & Secco (2020) in Fe_2SiO_4 , i.e. coeval development of spinel-filled anticracks and dislocation stacks within olivine. In addition, the authors cite Green et al. (1990) at lines 52-53 in a way that could trigger doubt, and here I recall that this seminal study reported anticracks in natural minerals and not only in analogs. For the sake of fairness and to promote a mutual respect atmosphere, I request that the authors rephrase these sentences (see *line-by-line comments*).

- Major comment #3: Mix between triggering mechanisms and rupture mechanisms

I note that the authors and most reviewers continue to mix the triggering, nucleation and dynamic propagation mechanisms altogether, which is one of the major causes of the current (apparent) mess on the topic. Therefore, hereinafter I explain it all again, with the hope that everyone understand.

Recent studies (and my own PhD thesis) have clearly explained that the triggering mechanisms of initial mechanical instabilities and the rupture (nucleation and propagation) mechanisms should be distinguished (e.g. Ferrand, 2017; Ferrand et al., 2021; Mao et al., 2022). A rupture is a rupture, i.e. a dynamic process with more or less efficient lubrication that depends on the overall conditions of the tearing material (Ferrand et al., 2021). “Transformational faulting” is a vague concept that we need to consider as a transitional tool towards a better understanding of deep earthquakes.

Numerical modeling of thermal runaway helps understand the dynamics of rupture propagation (Thielmann et al., 2015; Thielmann, 2018a) but it is not sufficient to explain the entire rupture process, especially the nucleation stage (Thielmann, 2018b; Ferrand et al., 2021). We have published a review of evidences of rupture-induced melting in various lithologies from the lab and from the field, with numerical modeling of the energy balance at the advancing melting front (Ferrand et al., 2021).

Ferrand et al. (2021) study the dynamic rupture mechanism (which should be distinguished from the nucleation mechanism). Rupture-induced melting (due to shear heating) is also proposed as the most likely mechanism for deep seismic ruptures by recent seismological studies both around and within the metastable olivine wedge by Zhan (2017) and Mao et al. (2022), respectively. Actually, our recent study (Mao et al., 2022) highlights very high b-values of deep seismicity for $M_w < 3.8$ (JMA catalog, i.e. with the smallest completeness magnitude), most likely related to the hydration state (OH content) of cold subducting slab segments. So not only temperature, although the arguments about temperature are important as well. Alternative/complementary mechanisms, such as thermal pressurization (e.g. Wibberley & Shimamoto, 2005; Viesca & Garagash, 2015) and pulverization (e.g. Reches & Dewers, 2005; Aben et al., 2016; Incel et al. 2019) could also be mentioned. In addition, the triggering mechanism (local transformation) and rupture mechanisms (e.g. transient melting due to shearheating of either olivine peridotite or wadsleyite peridotite) are discussed by Mao et al. (2022).

The triggering mechanism of deep earthquakes describes how the initial critical distortion of the stress field is generated. They would mostly be related to olivine phase transitions (e.g. Green & Burnley, 1989; Green et al., 1990; Schubnel et al., 2013) but also possibly to the destabilization of other minerals such as pyroxene (Shi et al., 2017; Xu et al., 2018). The triggering mechanism describes neither

the rupture nucleation nor the rupture propagation. It describes only the origin of the critical stress distortion upon local transformation, as detailed for intermediate depths (Ferrand, 2019a).

Regarding the initiation of the shear crack instability (**nucleation mechanism**), it would be fair to mention the reaction-induced grain size reduction (Incel et al., 2017; Thielmann, 2018b) and the grain-boundary disordering mechanism (Ferrand & Deldicque, 2021). The **reviewer #1** also asked for a better consideration of alternative mechanisms, including GB processes. Please note the grain-boundary instability reported in Mg_2GeO_4 in a narrow temperature window, which can constitute an efficient viscosity reduction mechanism before reaching the melting conditions. I think that the authors should recall that transformations (e.g. phase transitions, grain-boundary disordering, amorphization, melting...) can be important deformation mechanisms (Wheeler, 2020; Green et al., 2015).

The mentioned Peierls creep is a low-temperature high-stress mechanism related to the Peierls stress, i.e. stress required to trigger dislocation glide in crystals. Several other mechanisms could be discussed in the paper. See a review of mechanisms by Ferrand & Deldicque (2021). I think that the authors should recall the different kinds of grain-boundary sliding/rotation mechanisms, including the transformation-assisted grain-boundary sliding proposed to explain the viscosity reduction in Mg_2GeO_4 in a narrow temperature window (1000-1150°C). The “transformation” of the transformation-assisted GBS can be any kind of grain-boundary destabilization, including phase transition (Green et al., 2015), grain-boundary premelting/disordering (Takei et al., 2019; Yamauchi & Takei, 2020; Ferrand et al., 2021), melting (present study; Kloe et al., 2000) or amorphization (Samae et al., 2021).

As explained in **major comment #6**, the present study sheds new light on the nucleation mechanism of deep ruptures in the MTZ, and confirms the melt-assisted propagation mechanism.

Summary of mechanisms (in my opinion):

- **Trigger:** transformation, i.e. anticrack formation, and associated mechanical instability due to stress distortion at the tip of anticracks.
- **Rupture nucleation:** local formation of oblique lenticular aggregates and localizing strain accommodated by grain size reduction and transformation-assisted grain-boundary instability.
- **Dynamic rupture propagation:** shear melting due to accelerating strain rate and melt-assisted fault slip.

- Major comment #4: Confusion between seismic “faulting” and coalescence “faulting”

Throughout the paper, there is a confusion between single-event “faulting” due to seismic or aseismic rupture processes (e.g. Aubry et al., 2018; Ferrand et al., 2017;2018) and sample-scale “faulting” due to the (final) coalescence of individual faults (e.g. Shen et al., 1995; French & Zhu, 2017; Gasc et al., 2017). In such experiments, the coalescence is controlled by the single-event faults that form first (associated with the reported acoustic activity), but also by the sample dimensions and rheological contrasts between the sample and the other materials within the deformation assembly. The word “faulting” is problematic; thus, a proper definition is required.

As I have explained in details (Ferrand et al., 2021), micropseudotachylytes (microfaults) that are confined inside the sample (i.e. fault length smaller than the sample size) are more representative of natural seismic faults than sample-scale faults (affecting the deformation assembly). I recall that the largest rheology contrasts are not within the sample but between the different parts of the assembly. It would have been interesting to see the single-event faults corresponding to the reported labquakes (before coalescence), because the coalescence usually reworks/connects only some of them during the formation of the sample-scale fault zone, and such confined faults (before coalescence) are not contaminated by the other part of the experimental assembly (see **major comment #5**) and can give important insights on rupture mechanics (e.g. Ferrand et al., 2017).

- Major comment #5: Overinterpretation regarding the transient melting stage

The authors claim that partial melting occurs during their experiments, but they do not provide convincing evidence to conclude that the melting would actually be “partial”. The interpretation in terms of incongruent melting is not reasonable and seems contradictory with the reported results.

The composition of the melt could be explained by either partial melting or partial crystallization. Partial melting is a well-known process affecting the mantle in fixed P-T conditions, responsible for the significant chemical difference between mantle and crustal rocks. In contrast, pseudotachylytes (either natural or experimental), report local but total melting (see Ferrand et al., 2021 for a review). The fossilization of a multi-phase system after the seismic event would indicate partial crystallization of the melt during the quenching stage (deceleration), i.e. late coseismic creep (Ferrand et al., 2018). Rupture-induced total melting is observed in both experimental (e.g. Nielsen et al., 2010; Di Toro et al., 2011; Hayward & Cox, 2017; Lockner et al., 2017; Aubry et al., 2018) and in natural rocks (e.g. Andersen et al., 2014; Ferrand et al., 2018), with subsequent crystallization documented during cooling soon after mantle ruptures (Ferrand et al., 2017; 2018). In the HP pseudotachylyte of the Balmuccia peridotite, fractionation is seen along “injection” veins (i.e. tension cracks; Ferrand et al., 2018).

I do not see in the present paper sufficient evidence to conclude in favor of partial melting. If the authors want to stick to the partial melting interpretation, they should clearly detail the alternative possibilities and properly explain why their hypothesis is favored. Importantly, considering competent rocks under pressure, the temperature reached right at the tip of the shear crack is high enough to melt anything on the rupture path (especially for fresh peridotites, for which melting points of all minerals are relatively close to each other). In illustrations, the authors should show not only the solidus but also the liquidus (partial vs total melting). If the authors are interested in understanding the recent achievements about seismic rupture dynamics, I would be happy to explain it in details. They can read the recent comparative study by Ferrand et al. (2021), considering various works from either field geology or laboratory experiments and proposing modeling of the advancing melting front as a Stefan problem. → See **major comment #3**.

The authors claim that they observe “iron-rich nanoparticles (FeO/MgO ~0.4–0.9)” associated with “silicon rich patches (SiO₂ = 62-84 wt.%)” that would be “distributed in a gouge layer developed in the OL100 sample of M3100 (Figs. 5f-h and 6b).” They further claim that they observe SiO₂ particles, interpret it as stishovite and propose that it would originate from incongruent melting of wadsleyite. All of this is overinterpretation. First, the Fe-rich phase is documented in samples M2676 and M3425, whereas the “Si-rich path” is only reported in sample M3100, so these two phases are not “associated”. Second, on **Fig.5** (sample M3100), no SiO₂ is documented. Third, on **Fig.6b** the reported compositional trend could well be explained by the resolution limit of EDS measurements, because the local patches are small and close to minerals of other compositions. The authors should show where the data are collected and check whether or not the observed trend could be due to contamination of the EDS signal by neighboring phases. The EDS method is semi-quantitative and one should be very careful with absolute values. The authors should provide a justification for the uncertainty range, especially for the ultra-fine grain materials and the Fe-rich grain-boundary films.

One could hypothesize that the documented Si-rich patch would consist of nanograins of stishovite lost in the middle of some amorphous material, but this interpretation seems really exaggerated. The presence of stishovite is not really compatible with the rheology. Sample M3100 is dunite, and stishovite is not expected in dunite. In such a rock, the Si-rich phase can only come from the crystallization of the rupture-induced melt or be a pre-existing phase (which is likely the case on Fig.5g). Stishovite can only exist in pyroxene-rich compositions at P > 16 GPa (**Fig.R1**). In addition, the only “Si-rich patch” showed by the authors (**Fig.5g**) does not seem to originate from melting. It is clearly a pre-existing phase, cut and shifted by a branch of the fault zone that I highlight in the re-interpretation of **Fig.R2**. This phase could be minor enstatite, frequent with San Carlos olivine, but the authors could propose some other mineral richer in Si. In any case, I request that the authors show the acquisition location of the data presented in **Fig.6b**.

Fig.R1 – P-T diagrams showing the stability field of olivine (left) and enstatite (right) and their high-pressure polymorphs or decomposition parageneses (after Ferrand & Deldicque, 2021).

I would suggest to summarize the experimental series using a P-T diagram, to help the reader (and maybe the authors) better understand the significance of this work. See an example for sample M3100 in **Fig.R3**. At 15.6 GPa (M3100), a temperature of 1160 K ($\approx 890^\circ\text{C}$) is the condition for the transition from wadsleyite to ringwoodite. This transition may influence the relationship between deformation and transformation during the experiment (**major comment #3**). In addition, shear heating would stabilize wadsleyite and even favor a transformation back to olivine around 2170 K ($\approx 1900^\circ\text{C}$; **Fig.R3**). This means that, in case of transient rupture-induced melting, olivine will crystallize first, followed by wadsleyite. This could well explain the olivine+wadsleyite aggregates in fault veins (but not in the oblique lenticular aggregates (see **major comment #6**).

Fig.R2 – Re-interpretation of **Fig.5g**, highlighting the fault zone, with branches that clearly cut both the wadsleyite (or ringwoodite) zones and the Si-rich patch (most likely HP clino-enstatite). The white stuff seems to be contamination.

Fig.R3 – P-T diagram summarizing experiment M3100 on San Carlos olivine composition, according to the information provided in the manuscript and SI (modified after Ferrand & Deldicque, 2021).

NB: the red shadow accounts for minimum hydration (< 50 ppm). Higher hydrous fraction would reduce the melting temperature (Hirschmann et al., 2009).

On **Fig.4e** and **Fig.5c**, the “Pt” and “Ni” patches give the same elemental signal as what the authors refer to as “Fe-rich phase”. The melting point of “Pt” is 1768°C (i.e. 2041 K), which is significantly less than the estimated peak temperature of 2500 K. Consequently, both the metal spherulites and grain-boundary phase could contain metals from the assembly, and a clarification is requested.

- Major comment #6: Olivine transitions: anticracks vs oblique lenticular aggregates

The authors report oblique lenticular aggregates of mixed olivine polymorphs, which is, in my opinion, the key finding of the study. But there is a serious problem in the interpretation (e.g. lines 188-191). On one hand, indeed, the oblique lenticular aggregates seem to provide evidence for the process of rupture nucleation at the onset of transformational faulting (extremely important!!) **BUT** on the other hand, the microstructures reported by the authors indicate that these features are fundamentally different from anticracks (also visible on **Fig. 5g**), contrary to what the authors write at line 184. Consequently, the paragraph of lines 174-191 should be cut into two paragraphs: one paragraph for the oblique lenticular features (possibly the rupture nucleation stage) and one paragraph for anticracks (with no clear evidence for a link with local instability).

In **Fig.5g**, we can see the anticracks (consisting of wadsleyite and/or ringwoodite) described by Green & Burnley (1989) and Green et al. (1990), and reproduced by further studies (Riggs & Green, 2005; Officer & Secco, 2020). The authors annotate the anticrack-like features with “Wad/Rin” but do not describe or discuss these features. It is unfortunate that the FIB section does not cross the anticracks (**Fig. 5g**). Anticracks form normal to the maximum compressive stress (+ some variability in case of rupture due to the stress distortion at the rupture front), and won’t be aligned with the fault. In addition, anticracks seem cut by faults (see **Fig.R2** or **Fig.5g**), suggesting that the transformation and the rupture do not occur within one single process (main hypothesis of the unclear “transformational faulting” model).

The authors could explain that, while the oblique lenticular aggregates seem indeed related to rupture nucleation, the observed anticracks do not seem related to the fault initiation. Note, however, that the left side of **Fig.5g** shows many anticracks, while the right side does not show much of it. These anticracks might be a consequence of previous seismic ruptures (several events recorded). Considering a rupture nucleating around the center of the sample and propagating to the area shown in **Fig.5g**, the left side, full of anticracks (mostly normal to the main compressive stress or to the fault) would correspond to the compressive quadrant. In my opinion, here the authors have a good opportunity to discuss the rupture mechanisms thanks to these key microstructural imaging (**Fig.7d-e**).

Finally, in this paper, there is no mention of poirierite (the ω-olivine), recently recognized as the fourth olivine polymorph (Tomioka & Okuchi, 2017; Tomioka et al., 2021). Although this is relatively new and we do not know if poirierite has a proper P-T stability field (Ferrand & Deldicque, 2021), it

should at least be mentioned in the introduction and an explanation for its absence (or absence of observation) in the reported microstructures should be provided in the discussion.

Line-by-line comments:

Title:

The present study documents significant achievements + replications of features already documented in the literature. In my opinion, the title does not highlight the new insights of the paper, i.e. the evidence of the nucleation stage of the mysterious “transformational faulting” “mechanism”.

→ I would suggest the following rephrasing and clarification:

“In-situ X-ray and acoustic monitoring during olivine phase transitions reveal the rupture nucleation stage of transformational faulting”

or **“Simultaneous in-situ X-ray and acoustic monitoring during olivine high-pressure transitions reveal the rupture nucleation stage of deep earthquakes”**

Abstract:

Lines 21: “abruptly decreases to zero at 680 km” → Deep earthquakes can occur deeper (e.g. Kuge, 2017; Kiser, 2021). Some large events were recorded down to ≈ 750 km depth, due to cold slabs portions reaching below the transition zone (Kiser et al., 2021). → **clarification needed.**

Lines 21-22: “because brittle failure is unlikely to occur under the corresponding pressures of 13–24 GPa.” → This is a classic statement that all the community has been writing for decades because of the misunderstandings we have had regarding deep rupture processes. They are NOT unlikely, otherwise they would not occur so frequently. So, this sentence is not really informative and sounds outdated and misleading. → see **major comment #1.**

Line 23: “suggested” → At this stage, this is not a “suggestion” anymore. → **rephrasing needed.**

Line 24: “deep earthquakes” → This term is generally used to describe both intermediate-depth earthquakes and deep-focus earthquakes. Writing “deep-focus earthquakes” would help avoid confusion. In any case, it should be properly defined in the introduction.

Lines 25-27: → unnecessary attack against previous seminal works. Science progresses every day, and this is not a reason to minimize the importance of previous works. The present experimental work exists thanks to previous works and can be presented as a necessary continuity of previous achievements, instead of illegitimate criticism → **could be easily rephrased.**

Line 30: “colder” → “cold”

Line 30: “mantle transition region” → “mantle transition zone” **NB:** Even if both terminologies are found in the literature, it is better to keep using the dedicated vocabulary.

Lines 32-32: “immediately before the rupture” → **WARNING: vocabulary issue + misconception.** The acoustic activity located inside the sample is necessarily the consequence of dynamic ruptures (e.g. double-couple cracks). It cannot be before! Thus, the term “rupture” at line 33 is incorrect. Instead, the authors probably refer to the “sample-scale faulting”, due to coalescence (**major comment #4**).

Lines 36-37 → This is an important finding, which definitely legitimates publication in *Nature Communications*!

Lines 37-39 → This is correct, and consistent with most recent studies on deep-focus earthquakes.

Introduction

Line 40: “(Mg,Fe)₂SiO₄ olivine ...” → “Polymorphs of (Mg,Fe)₂SiO₄ ...” or “Natural olivine polymorphs, (Mg,Fe)₂SiO₄, ...”

Lines 45-46 → **WARNING:** anticracks do not form along the maximum compressive stress but normal to it... → see **major comment #6.** → **clarification needed.**

Lines 49-50: “which behaves as a low-viscosity fluid due to...” → **Incorrect** → suggested rephrasing: “which would be characterized by low viscosity due to...” (see **major comment #1**).

Lines 52-53: “which may not be an adequate analogue” → **Inappropriate**. In spite of differences, always mentioned by the cited studies, both Mg_2GeO_4 and $(Mg,Fe)_2SiO_4$ transform to a spinel or spineloid structure when cold slabs travel in the mantle transition zone. In addition, citing Green et al. (1990) to trigger doubt regarding their own work does not seem honest. Indeed, Green et al. (1990) have shown that anticracks form in natural minerals and not only analog materials. → **attack not needed** → **could be easily written in a less “aggressive” way (see major comment #2)**.

Lines 58-60 → The reference to Officer & Secco (2020) is relevant, and I want to draw your attention about additional publications that could help understand/arguing. Recent studies clearly explain that the triggering mechanisms of initial mechanical instabilities and the rupture mechanisms should be distinguished (e.g. Ferrand et al., 2021; Mao et al., 2022). → **(see major comment #3)**.

Line 59: “suggested that development of the fault slip requires ...” → “suggested that the development of substantial fault slip requires ...”

Line 60 → The reference to Aben et al. (2015) is not appropriate here.

Line 60: “olivine/fayalite” → “olivine (forsterite or fayalite)” or “olivine (San Carlos or fayalite)”

Remark: Fayalite is olivine too, and Mg_2GeO_4 is the analogue of forsterite. We need to remain very careful using the terms, especially considering that the targeted journal can bring readers from various fields, who may get lost at some point.

Lines 60-62 → This sentence appears incorrect or excessive. And the formulation is not clear. The experiments of Officer & Secco (2020) were performed using another type of multi-anvil apparatus, but several studies (Wang et al., 2004; Gasc et al., 2011; Schubnel et al., 2013; Incel et al., 2017; Ferrand et al. (2017) have already used the D-DIA apparatus with both high-energy X rays and acoustic recording, notably leading to substantial improvements of the understanding of intermediate-depth earthquakes. I have contributed to several of these studies and I can confirm that the evolution of stress and strain is precisely monitored and that strain at the scale of the deformation assembly can be linear using reasonable conditions. → see **major comment #2**.

Results

Line 69: “synchrotron in situ deformation experiments on sintered body ...” → several problems: first, what is “in-situ” is the X-ray diffraction and radiography, not the experiment; second, the high-energy X-ray produced by the synchrotron technology is important to mention, because they are a key of the study, which requires clarification in the formulation; third, other key aspects of the used technology, such a high pressure and acoustic recording, should be mentioned at the same time. → I recommend for example the following rephrasing: “high-pressure, high-temperature deformation experiments with in-situ synchrotron radiation and acoustic recording on sintered samples of ...”

Line 77 → change “:” for “.”

Lines 92-93: “does not account” → “accounts” (double negation with “Neither... nor...”).

Line 94: “Peierls creep” → This is fully contradictory with the previous sentence. → **clarification needed. + Why highlighting the Peierls creep and not other mechanisms? Especially, I think the authors recall that transformations (e.g. phase transitions, grain-boundary disordering, melting, amorphization...) are among the deformation mechanisms (Wheeler, 2020; Green et al., 2015; Yamauchi & Takei, 2020; Ferrand & Deldicque, 2021; Kloe et al., 2000; Samae et al., 2021). The authors could consider the transformation-assisted grain-boundary sliding proposed by Ferrand & Deldicque (2021).** → see **major comment #3**.

Line 96: “AE” should be defined into parentheses earlier, because some readers may not know that this refers to “acoustic emissions”.

Line 96: “which” → “but” or “and”

Line 97: “suggesting” → “indicating”

Line 98 → Wouldn't "ductile" be the right word here?

Line 106: "the semi-brittle flow was terminated by faulting" → If I understand well what the authors report in the previous subsections, this semi-brittle flow is the source of the acoustic emission, and "faulting" refers to the final sample-scale faulting, due to the coalescence of the seismic and/or aseismic faults forming during the semi-brittle flow. → **Substantial rephrasing and details required (see major comment #4).**

Line 106: "followed by a sudden large pressure drop (blow-out)" → In the high-pressure community, a "blow-out" refers to an unexpected pressure drop and end of the experiment due to a mechanical instability at the scale of the pressure chamber. In all likelihood, the very large deformation associated with the coalescence is responsible for the stress transfer at the assembly scale triggering blow-out (see major comment #4).

Line 108: "the former three runs" → "these three runs"

Line 110: "outside the sample along the fault plane crossing the sample" → "outside the sample and aligned with the fault plane" **NB:** this most likely correspond to ruptures in MgO and/or Al₂O₃, as indicated by fractures in **Fig. 5a** and **Fig.5f**.

Line 111: "suggesting the occurrence of a fault-slip associated rupture" → Extremely confusing for the reader. I recall that, considering a single event, the fault slip is the consequence of the rupture (see major comment #4).

Line 115: "samples" → "sample"

Line 116: "subsequent active AEs" → **confusing:** "active acoustic emissions" does not make sense, so I propose the following rephrasing: "subsequent substantial/intense acoustic activity".

Line 117: "upon faulting" → **confusing** → "upon coalescence" or "upon sample-scale faulting" (see major comment #4).

Line 117: "catastrophic adjustment of the pressure" → This is the stress transfer I refer to in my comment to line 106. → It is not a pressure adjustment but a stress adjustment, and more precisely, a "catastrophic perturbation of the stress field around the sample". Then, the blow-out is associated with a pressure drop, which is a consequence of the latter.

Line 115: "before faulting" → **WARNING: very problematic confusion (see major comment #4).**

Line 124 → The reference to Okazaki & Hirth (2016), i.e. study of dehydration embrittlement of lawsonite at intermediate-depth conditions, is totally out of topic. → **find adequate reference or remove sentence.**

Line 128 → This argument comes from Chernak & Hirth (2011), and has been referred to in studies on mantle rheologies (e.g. Ferrand et al., 2017). → **change reference for Chernak & Hirth (2011).**

Lines 128-129: No... First, as noted above, "faulting" should be properly defined (rupture vs coalescence). Second, the use of the term "precursor" is totally incorrect, because splitting is the consequence of coalescence. → see major comment #4.

Lines 132-133: "commencement of faulting" → seismic and/or co-seismic brittle events (depending on acoustic activity) → see major comment #4.

Lines 133-136 → Long sentence that could be shortened, and the end is unclear.

Line 138 + throughout the manuscript: "partial melting" → Are the authors sure about the evidence of partial melting? To what extent are they sure that it could not be fractionation of the melt due to crystallization during fault slip? → see major comment #5.

Line 140: "sample of M3425" → "sample of M3425"

Line 140: "..., where the displacement of the fault is 280 μm." → "..., with a cumulative displacement of 280 μm." → see major comment #4.

Line 147: "... of gouge layer." → "... of the gouge layer."

Line 148: QUESTION: Is there any other source of Fe in the assembly?

Line 149: "partial" → see **major comment #5.**

Line 150: “Evidence of partial melting” → I cannot see that. It could be either partial melting (unlikely) or partial crystallization! → see **major comment #5**.

Lines 150-152 → If the authors are interested in understanding the recent achievements about seismic rupture dynamics, I would be happy to explain it in details. They can read the recent comparative study by Ferrand et al. (2021), considering various studies from either field geology or laboratory experiments and proposing simple modeling of the advancing melt front. → see **major comment #5**.

Lines 152-153: “instantaneous temperature increase” → I would advise that the authors support their work with some adequate references about adiabatic/diabatic dynamic melting during dynamic rupture (e.g. Ferrand et al., 2021).

Line 156: “a throughgoing faults” → “a throughgoing fault” or “a throughgoing fault zone”

Line 156: “in the OL92 sample of M2676.” → “in run M2676 (sample OL92).”

Line 156: “displacement” → “cumulative displacement”

Lines 160-161: “melt-like phase” → Term not adequate → Suggested rephrasing: “amorphous material” or “grain-boundary amorphous phase” or “grain-boundary glass” → **correction needed**.

Line 161: “No micrometric wadsleyite” → Why do the authors specify “micrometric” here? Could there be nanometric wadsleyite? → **clarification needed**.

Line 162: “The melt-like phase is amorphous (Figs. 5d-e), which is enriched...” → “The phase wetting grain boundaries (Figs. 5d-e) is amorphous and enriched ...”

Lines 163-171: **WARNING: unreasonable interpretation, which seems contradictory with the reported results.** → see **major comments #5 and #6**.

Line 174: “without faulting” → **WARNING: incorrect.** The authors have not demonstrated that there would be no faulting in the sample. In addition, this would be contradictory with the recording of acoustic activity from inside the sample. → **rephrasing required (see major comment #4)**.

Lines 177-178: “Both of these two and other smaller AEs are all confined in the sample region” → It would have been interesting to see the single-event faults corresponding to these labquakes, because such confined faults are not contaminated by the other part of the experimental assembly and can give important insights on rupture mechanics (e.g. Ferrand et al., 2017).

Lines 181-183 → Speculative and useless.

Line 183: “pulverization” → **WARNING:** pulverization is a fully different process, associated with rupture mechanics and related to loss of cohesion and further grain size reduction during rupture propagation (e.g. Reches & Dewers, 2005; Aben et al., 2016; Incel et al. 2019). + Note that there is NO mention of pulverization in the cited paper (Liu et al., 1998)!!

Line 184: “growth of” → “the growth of”

Line 184: “is observed” is doubled → **remove**.

Lines 184-185 → This is not a convincing argument. **Coincidence is not proof of causality.**

Line 187: “The morphology of the observed microstructures is quite similar to that of ‘anticracks’” → **WARNING:** No, it is not. The previous sentence (lines 185-186) deals with the oblique lenticular packets (Fig.7d-e), **very different from anticracks** and with different orientation! → see **major comment #6**.

Lines 188-191 → **Yes and no:** on one hand, indeed, the lenticular packets (key observation of the study) seem to provide evidence for the process of rupture nucleation at the onset of transformational faulting (extremely important!!) **BUT** on the other hand, the microstructures reported by the authors indicate that these features are fundamentally different from anticracks, also visible on **Fig. 5g**.

Lines 174-191 → **This paragraph should be cut in two paragraphs: one paragraph for the oblique lenticular features (possibly the rupture nucleation stage) and one paragraphs for anticracks (with no clear evidence for a link with local instability).**

Discussion

Line 196: “proceeds on GBs” → “proceeds at GBs”

Lines 196-197: “To estimate the rate of olivine-wadsleyite phase transition on” → “To estimate K at”

Line 200: “follows” → “follow”

Line 203 → the units of a_0 and b_0 should be specified.

Line 204: “of reaction” → “of the reaction”

Line 211: “the volume fraction transformed” → “the transformed volume fraction”

Line 212: “the rate constant during the early stages of transition K is given as” → “the rate constant K during the early stages of transition is given as follows”

Line 213 → Choice required between “ K ” and “ K ”. Usually, italic font style is used for variables and regular font style for constants. **Remark:** the definition of K (eq.3) should be given first, followed by the detailed description of all variables in the equation, including additional equations (eq.1 and eq.2).

Line 214: “Formation of” → “The formation of”

Line 215 → For sake of clarity, it should be defined in the first sentence of the *Discussion* section, that this parameter is a constant in fixed P-T conditions but may significantly vary during the experiment.

Line 215: “of the K ” → “of K ”

Line 219: “very low” is not nothing...

Line 221: “faulting” → clarification needed. → see **major comment #4**.

Line 223: “Elastic energy stored in ...” → “The elastic energy stored in ...”

Line 224: “high-temperature” is obvious and redundant with “thermal runaway”

Lines 224-226 → **WARNING: serious lack of consideration for the literature (major comment #2)**.

Line 235: “Instantaneous strain rate” → “The dynamic strain rate”

Line 238: “by formation” → “by the formation”

Lines 238-239 → Regarding the initiation of the shear crack instability, it would be fair to mention the reaction-induced grain size reduction (Incel et al., 2017; Thielmann, 2018b) and the grain-boundary disordering mechanism (Ferrand & Deldicque, 2021). Please note that the grain-boundary instability is reported in Mg_2GeO_4 in a narrow temperature window, which can constitute an efficient viscosity reduction mechanism before reaching the melting conditions. The authors might find useful to mention premelting (e.g. Takei et al., 2019; Yamauchi & Takei, 2020) → see **major comment #3**.

Line 240: “On the other hand” → should come after “On one hand” in order to highlight two things, before concluding about the level of consistency or discrepancy between the two things.

Lines 240-242 → **WARNING: Such a strong statement would require a proper demonstration!!! It does not sound reasonable at all.**

Lines 242-245 → **WARNING: This is incorrect.** The authors completely ignore grain-boundary processes and the increased importance of grain boundaries in case of grain-boundary instability. This is not acceptable. In addition, I want to stress that there is not only one rate-limiting process for superplasticity. I recommend that the authors read more papers of material sciences, especially ceramics and metallurgy. They could have a look at the trial of interdisciplinary integration by Ferrand & Deldicque (2021). → see **major comment #3**.

Line 264: “due to accumulation” → “due to the accumulation”

Line 265 → Again, to what extent are we sure that deep earthquakes are triggered by an adiabatic instability? → Are they “triggered” by this process or do they “nucleate” via this process? This is a key distinction that we all need to understand and/or discuss. → **clarification needed** (I would suggest to replace “triggered” by “nucleated”). → see **major comment #3**.

Line 268: “superheating” → What does that mean?! + grammatical issue... → **clarification needed**.

Lines 272-277 → This is very important and it should be discussed in light of recent machine-learning-based seismological findings, which suggest that deep seismicity within the mantle transition zone

would occur in the unstable rim around the metastable wedge (Mao et al., 2022). The unstable rim is most likely corresponding to a thermal limit at the balance between displacive and diffusive processes (Ferrand & Deldicque, 2021).

Line 280 → Please pay attention to the most-recent b-value analysis supporting this sentence (Mao et al., 2022).

Lines 280-284 → Actually, our recent study (Mao et al., 2022) suggests that it is controlled by temperature and OH content. → see **major comment #3**.

Lines 285-287 → If this is correct and fully applicable to all subducting slabs, then deep earthquakes between 630 and 750 km depth would be due to some other transformation. **But what?**

Lines 287-288: “shear zones filled with nanocrystalline olivine required for faulting would not be preserved” → But their souvenir would remain! (i.e. nanocrystalline wadsleyite and/or ringwoodite).

Line 289 → See **major comment #1**.

Figures and captions

Lines 292-293: “a, Faulting occurred. b, No faulting occurred” → **INCORRECT (major comment #4)**.

Lines 295-304 → the crosses should be described in the caption. A clear distinction should be written between the latter (corresponding to rupture-induced melting, i.e. local and transient) and the other symbols (corresponding to the average bulk conditions). → **clarification needed**.

Caption of Fig.2 → For colors and symbols, it should be written that the legend is in **Fig.1**.

Fig.3a → I suggest to adjust the colors of the strain and amplitude axes to be consistent with the rest and to help the reader not to get lost.

Fig.3c → For $\epsilon = 0.85$, an explanation for the top pink arrow (throughgoing faulting) could be useful for the reader.

Fig.3 → I suggest to move **b** and **c** next to **a**., which I believe would better match the journal format.

Lines 323-324: “No diffraction peak of wadsleyite 240 was detected throughout the run, as shown in a dotted area.” → It looks a bit odd. It is good that it is written in the caption, but the location of this dotted area in the plot seems misleading. → **remove**. In addition, I would rephrase for a more understandable sentence as follows: “The diffraction peak 240 of wadsleyite was not detected in the run.”

Line 327: “shows typical errors for the location of hypocenter.” → “shows the uncertainty range for hypocenter locations.”

Line 328: “at each strain ϵ ” → “for three values of strain ϵ representative of the entire experiment.”

Lines 331-332: “A backscattered electron image” → “Backscattered electron image”

Line 333: “A STEM image of a fault associating the gouge layer (the region “b” in a).” → “Scanning transmission electron microscope (STEM) image of a 4- μm -thick fault (located in a).”

Remark: It is important to define every acronym when it occurs for the first time in the manuscript.
+ most readers know SEM and TEM, but not necessarily STEM.

Line 334: “A magnified view” → “Magnified view”

Line 335: “Typical grain size” → “The grain size”

Line 336: “A TEM image of wadsleyite grains in the gouge layer. A diffraction pattern is also shown.” → “TEM image of a wadsleyite nanocrystal in the olivine-wadsleyite gouge aggregate, with associated diffraction pattern.”

Line 337: “An enlarged STEM image and an element map” → “Enlarged STEM image and Mg/Fe element map”

Line 338 → “In element maps, grayscale corresponds to concentration of elements” → “In the element map, the intensity of colors accounts for element concentrations”

Caption of Fig.4: all the occurrences “(the region “y” in x)” can be shorter → “(located in x)”.

Fig.4d + Fig.5e → **WARNING:** such diffraction patterns should not be truncated and they should be accompanied with a scale (in nm^{-1})!!! → See **major comment #1**.

Fig.5g → WARNING: overinterpretation. → See major comment #5

Caption of Fig.5 → Same remarks as for the caption of Fig.4. → corrections needed.

+ “Mj” definition is missing.

Fig.6 → The composition estimates from EDS measurements do not seem convincing. → see **major comment #5**.

Caption of Fig.6 → again, majorite is not defined.

Caption of Fig.7: “without occurrence of faulting” (line 371) → **WARNING** → see **major comment #4**.

Caption of Fig.8: “The temperature dependence of the olivine-wadsleyite transition K is calculated ...”

→ “Temperature dependence of the rate constant K for the olivine-wadsleyite transition calculated ...”

Caption of Fig.9: “mantle transition region” → “mantle transition zone”

Extended Data Figure 4 → The large yellow dashed circle is hiding the arrow showing the tungsten ring. → clarification possible.

Methods:

Line 419: “8wt.%” → add space

Lines 417-429: → The authors do not describe the mixture procedure of olivine and enstatite, and they do not tell about the homogeneity of the enstatite distribution.

Lines 424-426: → Why not. However, again, the authors should discuss why they consider the solidus and not the liquidus, while field observations (e.g. Ferrand et al., 2018) on mantle rocks show local but total melting on the rupture path.

Line 426: “Mechanical behavior” → “The mechanical behavior” + several other similar grammatical issues in throughout the section (many “The” are missing). → **language verification needed**.

Line 429: “independent on these two samples” → “independent of this compositional different.”

Line 453: “events of small AEs (the maximum amplitude <1V)” → “small events (AE maximum amplitude <1V)”

Line 464: “equilibrium” → “equilibria”

Line 488: “later” → “the last”

Line 497: “The uncertainty of” → “The uncertainty on”

Line 532 → remove “on the signals”

Line 540: “thorough” → “through”

Line 543 → remove “high-pressure” **NB:** It would be useful to remove all redundant information.

Line 546 → Note that this definition of the acoustic emissions magnitude is used by previous studies on similar lithologies (e.g. Schubnel et al., 2013; Ferrand et al., 2017; Kita & Ferrand, 2018).

References cited in this review:

NB: The studies cited by the authors are not listed hereinafter.

- Andersen, T. B., Austrheim, H., Deseta, N., Silkoset, P. & Ashwal, L. D. 2014. Large subduction earthquakes along the fossil Moho in Alpine Corsica. *Geology* **42**(5), 395-398.
- Chernak, L. & Hirth, G. 2011. Syndeformational antigorite dehydration produces stable fault slip. *Geology* **39**, 847-850.
- De Kloe, R., Drury, M. R. & Van Roermund, H. L. M. 2000. Evidence for stable grain boundary melt films in experimentally deformed olivine-orthopyroxene rocks. *Physics & Chemistry of Minerals* **27**(7), 480-494.
- Di Toro, G., Han, R., Hirose, T., De Paola, N., Nielsen, S., Mizoguchi, K. & Shimamoto, T. 2011. Fault lubrication during earthquakes. *Nature* **471**(7339), 494-498.

- Ferrand, T. P., Nielsen, S., Labrousse, L. & Schubnel, A. 2021. Scaling seismic fault thickness from the laboratory to the field. *Journal of Geophysical Research: Solid Earth* **126**(3), e2020JB020694.
- Ferrand, T. P. & Deldicque, D. 2021. Reduced Viscosity of Mg_2GeO_4 with Minor MgGeO_3 between 1000 and 1150° C Suggests Solid-State Lubrication at the Lithosphere-Asthenosphere Boundary. *Minerals* **11**(6), 600.

- Ferrand, T. P., Labrousse, L., Eloy, G., Fabbri, O., Hilaiet, N. & Schubnel, A. 2018. Energy balance from a mantle pseudotachylite, Balmuccia, Italy. *Journal of Geophysical Research: Solid Earth* **123**(5), 3943-3967.
- Ferrand, T. P. Seismicity and mineral destabilizations in the subducting mantle up to 6 GPa, 200 km depth. 2019. *Lithos* **334**, 205-230.
- Ferrand, T. P., Hilaiet, N., Incel, S., Deldicque, D., Labrousse, L., Gasc, J., ... & Schubnel, A. 2017. Dehydration-driven stress transfer triggers intermediate-depth earthquakes. *Nature communications* **8**(1), 1-11.
- Ferrand, T. P. 2017. Reproduction expérimentale d'analogues de séismes mantelliques par déshydratation de l'antigorite & Comparaison à des pseudotachylites naturelles (Doctoral dissertation, Université Paris sciences et lettres).
- French, M. E. & Zhu, W. 2017. Slow fault propagation in serpentinite under conditions of high pore fluid pressure. *Earth & Planetary Science Letters* **473**, 131-140.
- Gasc, J., Schubnel, A., Brunet, F., Guillon, S., Mueller, H. J. & Lathe, C. 2011. Simultaneous acoustic emissions monitoring and synchrotron X-ray diffraction at high pressure and temperature: Calibration and application to serpentinite dehydration. *Physics of the Earth & Planetary Interiors* **189**(3-4), 121-133.
- Gasc, J., Hilaiet, N., Yu, T., Ferrand, T. P., Schubnel, A. & Wang, Y. 2017. Faulting of natural serpentinite: Implications for intermediate-depth seismicity. *Earth & Planetary Science Letters* **474**, 138-147.
- Hayward, K. S. & Cox, S. F. 2017. Melt welding and its role in fault reactivation and localization of fracture damage in seismically active faults. *Journal of Geophysical Research: Solid Earth* **122**(12), 9689-9713.
- Hilaiet, N., Wang, Y., Sanehira, T., Merkel, S. & Mei, S. 2012. Deformation of olivine under mantle conditions: An in situ high-pressure, high-temperature study using monochromatic synchrotron radiation. *Journal of Geophysical Research: Solid Earth* **117**(B1).
- Hirschmann, M. M., Tenner, T., Aubaud, C. & Withers, A. C. 2009. Dehydration melting of nominally anhydrous mantle: The primacy of partitioning. *Physics of Earth & Planetary Interiors* **176**, 54-68.
- Incel, S., Hilaiet, N., Labrousse, L., John, T., Deldicque, D., Ferrand, T. P. & Schubnel, A. 2017. Laboratory earthquakes triggered during eclogitization of lawsonite-bearing blueschist. *Earth & Planetary Science Letters* **459**, 320-331.
- Incel, S., Schubnel, A., Renner, J., John, T., Labrousse, L., Hilaiet, N., ... & Jamtveit, B. 2019. Experimental evidence for wall-rock pulverization during dynamic rupture at ultra-high pressure conditions. *Earth & Planetary Science Letters* **528**, 115832.
- Kiser, E., Kehoe, H., Chen, M. & Hughes, A. 2021. Lower Mantle Seismicity Following the 2015 Mw 7.9 Bonin Islands Deep-Focus Earthquake. *Geophysical Research Letters* **48**(13), e2021GL093111.
- Kita, S. & Ferrand, T. P. 2018. Physical mechanisms of oceanic mantle earthquakes: Comparison of natural and experimental events. *Scientific Reports* **8**(1), 1-11.
- Kuge, K. 2017. Seismic observations indicating that the 2015 Ogasawara (Bonin) earthquake ruptured beneath the 660 km discontinuity. *Geophysical Research Letters* **44**(21), 10-855.
- Lockner, D. A., Kilgore, B. D., Beeler, N. M. & Moore, D. E. 2017. The transition from frictional sliding to shear melting in laboratory stick-slip experiments. *Fault zone dynamic processes: Evolution of fault properties during seismic rupture* **227**, 105-131.
- Mao, G. L., Ferrand, T. P., Zhu, B., Xi, Z. & Chen, M. **In press**. Unsupervised machine learning reveals slab hydration variations from deep earthquake distributions beneath the northwest Pacific. *Communications Earth & Env.*
- Merkel, S. & Hilaiet, N. 2015. Multifit/Polydefix: a framework for the analysis of polycrystal deformation using X-rays. *Journal of Applied Crystallography* **48**(4), 1307-1313.
- Nielsen, S., Mosca, P., Giberti, G., Di Toro, G., Hirose, T. & Shimamoto, T. 2010. On the transient behavior of frictional melt during seismic slip. *Journal of Geophysical Research: Solid Earth* **115**(B10).
- Reches, Z. E. & Dewers, T. A. 2005. Gouge formation by dynamic pulverization during earthquake rupture. *Earth & Planetary Science Letters* **235**(1-2), 361-374.
- Riggs, E. M. & Green, H. W. 2005. A new class of microstructures which lead to transformation-induced faulting in magnesium germanate. *Journal of Geophysical Research: Solid Earth* **110**(B3).
- Samae, V., Cordier, P., Demouchy, S., Bollinger, C., Gasc, J., Koizumi, S., ... & Idrissi, H. 2021. Stress-induced amorphization triggers deformation in the lithospheric mantle. *Nature* **591**(7848), 82-86.
- Shen, B., Stephansson, O., Einstein, H. H. & Ghahreman, B. 1995. Coalescence of fractures under shear stresses

in experiments. *Journal of Geophysical Research: Solid Earth* **100**(B4), 5975-5990.

- Shi, F., Wang, Y., Zhang, J., Yu, T. & Zhu, L. 2017. Instability induced by orthopyroxene phase transformation and implications for deep earthquakes below 300 km depth. *AGU Fall Meeting Abstracts* **2017**, S43B-0841.

- Suzuki, K. 1987. Grain-boundary enrichment of incompatible elements in some mantle peridotites. *Chemical Geology* **63**(3-4), 319-334.
- Takei, Y. 2019. Experimental and theoretical approaches to grain boundary premelting: a possible origin of asthenosphere. *Geophysical Research Abstracts* (Vol. 21).
- Thielmann, M., Rozel, A., Kaus, B.J.P., Ricard, Y., 2015. Intermediate-depth earthquake generation and shear zone formation caused by grain size reduction and shear heating. *Geology* **43** (9), 791-794.
- Thielmann, M., 2018a. Grain size assisted thermal runaway as a nucleation mechanism for continental mantle earthquakes: Impact of complex rheologies. *Tectonophysics* **746**, 611-623.
- Thielmann, M., 2018b. Intermediate depth earthquakes due to grain size assisted thermal runaway: what are the odds? JpGU Meeting.
- Tomioka, N., Bindi, L., Okuchi, T., Miyahara, M., Iitaka, T., Li, Z., ... & Kodama, Y. 2021. Poirierite, a dense metastable polymorph of magnesium iron silicate in shocked meteorites. *Communications Earth & Environment* **2**(1), 1-8.
- Tomioka, N. & Okuchi, T. 2017. A new high-pressure form of Mg₂SiO₄ highlighting diffusionless phase transitions of olivine. *Scientific Reports* **7**(1), 1-9.
- Uchida, T., Funamori, N. & Yagi, T. 1996. Lattice strains in crystals under uniaxial stress field. *Journal of Applied Physics* **80**(2), 739-746.
- Viesca, R. C. & Garagash, D. I. 2015. Ubiquitous weakening of faults due to thermal pressurization. *Nature Geoscience* **8**(11), 875-879.
- Wang, Y., Uchida, T., Von Dreele, R., Rivers, M. L., Nishiyama, N., Funakoshi, K. I., ... & Kaneko, H. 2004. A new technique for angle-dispersive powder diffraction using an energy-dispersive setup and synchrotron radiation. *Journal of Applied Crystallography* **37**(6), 947-956.
- Wang, Y., Zhu, L., Shi, F., Schubnel, A., Hilairet, N., Yu, T., ... & Brunet, F. 2017. A laboratory nanoseismological study on deep-focus earthquake micromechanics. *Science Advances* **3**(7), e1601896.
- Wheeler, J. 2020. A unifying basis for the interplay of stress and chemical processes in the Earth: support from diverse experiments. *Contributions to Mineralogy & Petrology* **175**(12), 1-27.
- Wibberley, C. A. & Shimamoto, T. 2005. Earthquake slip weakening and asperities explained by thermal pressurization. *Nature* **436**(7051), 689-692.
- Xu, J., Zhang, D., Fan, D., Zhang, J. S., Hu, Y., Guo, X. & Zhou, W. 2018. Phase transitions in orthoenstatite and subduction zone dynamics: Effects of water and transition metal ions. *Journal of Geophysical Research: Solid Earth* **123**(4), 2723-2737.
- Yamauchi, H. & Takei, Y. 2020. Application of a Premelting Model to the Lithosphere-Asthenosphere Boundary. *Geochemistry, Geophysics, Geosystems* **21**(11), e2020GC009338.
- Zhan, Z. 2017. Gutenberg-Richter law for deep earthquakes revisited: A dual-mechanism hypothesis. *Earth & Planetary Science Letters* **461**, 1-7.

Sentences with Italic letter in black: reviewers' comments

Sentences in blue: our responses

Response to comments by the reviewer#1

#1-1. AE location/Magnitudes/Polarities

My first comment is on AE location. How was the error bar on location inferred?

As already mentioned in Methods, the error bar on AE location is defined as the root mean square of the right side of Eq. (6). However, this sentence may not be straightforward to the reviewer and other readers. To clarify the definition of the error bar, following sentence with Eq. (7) were added (lines 612-615):

“In this study, the location uncertainty is defined as the root mean square of the right side of Eq. (6), that is:

$$\text{error} = \sqrt{\frac{1}{5} \sum_{i=2}^6 |v_p(t_i - t_1) - R_i + R_1|} \quad (7)''$$

Due to the geometry of the AE sensor array, a calibration procedure must have been performed in order to test the location procedure. Indeed because of their set-up where sensors are glued on the anvil sides, there could be important effects of reflection on the side and back walls of the anvils. A simple calibration, for instance by performing cold compression of fine quartz aggregates as in Gasc et al. 2011, seems to be needed. In the absence of calibration, the actual location given in Figure 3c & extended figures 1-2 have little meaning. It might be that they are not needed in the final manuscript, as they do not reveal much anyways.

We agree that a calibration run using a fine-grained quartz aggregate (Gasc et al., 2011 PEPI) is needed. We actually conducted a calibration run (M0637: Table 1 and Fig. S6) according to the suggestion, showing that the locations of strong AE hypocenters calculated from Eq. (6) are mostly consistent with the actual position of the quartz-beads sample (i.e., AE source). Thus, we believe that AE hypocenters were properly determined in our experiments. Please see Fig. S6. Also, following sentences were added to the revised manuscript (lines 616-632):

“To demonstrate that AE hypocenters radiated from the samples were properly determined in our experiments, we conducted an offline test on cold compression of quartz beads⁶⁷ using a deformation-DIA apparatus at Ehime University. Quartz is known to display a brittle behavior associating a large number of AEs due to grain crushing and porosity loss during compression⁶⁸. The cell assembly used for the cold-compression experiment is the same as that used for the in situ deformation experiments. Quartz beads with a diameter of 200 μm or less were packed into a nickel capsule (Supplementary Fig. 5a). The quartz sample was sandwiched in between two hard-alumina pistons with a length of 0.3 mm in the nickel capsule. The cell assembly was pressurized hydrostatically to 0.6 MN in the main-ram load. AE events during cold compression were collected with the six PZT transducers that were attached to the sides of the second-stage tungsten carbide anvils (Supplementary Figs. 5d-e). Hypocenters of the AEs are shown in Supplementary Fig. 6. Most of the high amplitude AE events were recorded during the early stage of cold

compression, suggesting that the AEs are related to crushing of quartz grains accompanied by associated pore collapse at low pressures⁶⁷. Hypocenters of intense AEs (the maximum amplitude >1 V) are located within the quartz sample, showing that the determined locations of AE hypocenters are plausible.”

In a similar manner, the authors argue for a double couple source for these AEs (198-102). First, one should point that a DC source does not correspond to a simple mix of positive and negative polarities. Tensile or compactive cracking can also present, in the present of a limited shear component, a mix of first motion polarities. In such a way, the authors would have to demonstrate the actual DC nature of AE sources by performing a moment tensor inversion, such as was performed by Schubnel al. 2013 (Fig. 3). With their set-up, I believe it would be close to impossible, again due to the possible reflection on the side of the anvils, which may lead to polarity inversion upon reflection.

Yes, the mixture of positive and negative polarities of AE waveforms means both shear cracking and that associating isotropic compaction (note that tensile cracking is not considered because it is inhibited at pressures of 11-17 GPa in this study). As pointed out by the reviewer, moment tensor inversion is technically impossible in our experiments. Therefore, we do not exclude the contribution of isotropic compaction to AE radiation in the revised manuscript (please see lines 111-113):

“The first motion at each AE sensor shows that most of the AEs were generated by shear cracks (i.e., both positive and negative polarities) or those associated with compaction,…”

Finally, there is seem to be a problem when calculating AE magnitudes, which should be proportional to V^2 (energy) and not V .

Yes, moment magnitude should be proportional to V^2 . However, we deal relative magnitude (M_{ae}) in the manuscript, not moment magnitude. The relative magnitude is linearly proportional to the logarithm of the maximum amplitude of the AE signal (lines 594-597) (Lei and Sato, 2007 Tectonophys.).

“Following the definition of the body-wave earthquake magnitude, the relative magnitude (M_{ae}) of an AE event is determined according to the logarithm of the maximum amplitude of the AE signal (V_{max}) detected on a chosen channel⁶⁶, that is, $M_{ae} = \log_{10}V_{max}$.”

We admit that an additional explanation for the definition of the relative magnitude is needed. Thus, following sentence was added (lines 597-599):

“Note that the values of M_{ae} obtained in this way ought to be considered only as first-order approximations of the relative values⁶⁶.”

#1-2. Microstructure of faulted specimen

The microstructure of three specimens has been investigated because they faulted and

present fault planes. However, these microstructures are particularly difficult to interpret, since in the three experiments, faulting was followed by a blow out. So the questions are:

1. How much pressure / temperature was lost during these blow-outs?

Decrease in the pressure is estimated to be ~2-3.5 GPa from the records of main-ram load. Temperature was suddenly decreased to room temperature (i.e., quench). To clarify these points, following sentence was added (lines 134-136):

“The blow-out was accompanied by sudden drops in both pressure ($\Delta P \sim 2\text{--}3.5$ GPa: estimated from the change in the main-ram load) and temperature (to room temperature).”

2. How much of the microstructure reported in Fig. 4-6 is related to faulting and/or to blow outs? For instance, the displacement reported of several hundreds of microns (1140&1158) are hardly compatible with that being solely related to the HP-HT faulting. Indeed, a simple relationship exists in seismology between displacement and stress drop, where $\Delta\sigma \approx \mu D/L$, where $\Delta\sigma$ is the stress drop, μ the shear modulus, D the co-seismic displacement and L the fault length. Taking number such as 100GPa for μ , 1mm for L and 250 microns for D , leads to stress drops of >25 GPa, i.e. a value higher than the pressure itself. In consequence, whatever has happened on these experimental faults is certainly mostly related to blowouts and not shear-rupture (maximum co-seismic displacement of approx. 30microns for a complete stress drop at 3GPa differential stress). Accordingly, all the section on the occurrence of partial melting seems dubious and far-fetched, as it relies on improper attribution of a microstructure to the fracture propagation. This section has to be greatly revised, tampered and to the best, put in conditional form.

First, we should say that the equation ($\Delta\sigma \approx \mu D/L$) used by the reviewer to estimate the stress drop is applicable only when the rock is a complete elastic body (e.g., cold and shallow crust where plastic deformation never occurs). For instance, according to Kanamori et al. (1998 Science), it is mentioned that “*Because of the high pressure and temperature in the source region, the ordinary brittle failure is not likely to occur, and other mechanisms need to be invoked.*”. In the case of deep-focus earthquakes in the mantle transition zone, the relationship between $\Delta\sigma$ and L is more complex (please see Kanamori et al., 1998 Science for details), and the stress drop would be significantly overestimated using this equation.

To demonstrate blow-outs do not necessarily induce faulting, we added the result of an earlier experiment in the revised manuscript (M2472: Fig. S4). In this run, sample deformation was terminated by a sudden blow-out. However, formation of any faults was not observed in the recovered sample (Fig. S4b), suggesting that blow-outs are not the cause of faulting. This fact also supports our conclusion that a catastrophic adjustment of the pressure medium/gasket induced by the faulting results in a blow-out (lines 132-134).

“The faulting is likely to induce a catastrophic adjustment of the pressure medium/gasket which inevitably leads to a

blow-out, as none of other 11 runs without faulting suffered any blow-outs.”

We conclude that blow-outs do not induce faulting in our sample setting, and thus following sentences were added (lines 137-143):

“Note that a blow-out is not the cause of faulting in our cell assembly. As demonstrated in M2472, stress abnormally decreased down to -2 GPa (i.e., a strong tensile stress) accompanied by AEs before the occurrence of a blow-out (Supplementary Fig. 4a). Such strong tensile stress, which would be a characteristic phenomenon for blow-outs, was not detected in other runs. Even though sudden sample shortening was observed at the time of a blow-out, development of any faults was not observed in the recovered sample (Supplementary Fig. 4b).”

Nevertheless, we cannot completely exclude the possibility that blow-outs alter the observed microstructures. To clarify this point, following sentence was added (lines 176-178):

“Generation of instantaneous high temperature would be due to adiabatic shear heating upon faulting under the assumption that the observed microstructures were not altered by the blow-outs.”

#1-3. Figures and figure captions

Most of the figures of the paper are very small and hard to read, with too much information, which leads to confusion:

- Figure 1 is hard to read. What is the difference between PTt paths 2 & 3. Some of the PTt path 2 specimens seem to have undergone pressurization at high T, while some not. Please clarify the figure.

We admit that the previous version of Fig. 1 is hard to read due to too much information. We modified Fig. 1 with a reader-friendly layout, based on the reviewer#2’s advice (please see our response#2-5). Legends for the arrows were added to clarify the meanings of the P-T-t paths.

- In Figure 2, is it yield (if yes how as it determined) or rather flow strength that you represent and also refer to in Table 1? Please use clearer symbols for faulted samples. What do the red arrows correspond to?

Yes, yield strength is used in Fig. 2. To clarify this point, the label for the Y-axis was changed to “Yield strength (GPa)”. The symbols for faulted samples were highlighted. From the viewpoint of readability, the red arrows in the previous version of Fig. 2 were removed.

- The vertical grey bars (Max Amplitude V) in Figure 2b represent single experiment, several, single AEs, cumulative - please clarify -, Maybe a plot of cumulative AE energy detected for each experiment at each T would be easier to read?

We calculated the cumulative AE energy for each run. According to the reviewer's suggestion, cumulative AE energy is plotted against temperature in Fig. 2b. Also, the following sentence was added to explain the definition of AE energy (lines 599-601).

"To evaluate cumulative energy release via AE radiations in each run, AE energy was obtained by integrating the absolute amplitude value of a given waveform."

- Fig3 - dashed box – no Wad240 – is this really useful?

We agree that the dashed box with "no Wad240" is not useful information. The boxes were removed from Fig. 3 (also from Figs. 7 and S1-3).

- Following the above comment#2, Fig. 6 is meaningless.

As mentioned above (please see our response#1-2), an additional experiment (M2472) demonstrated that a blow-out did not necessarily induce faulting (Fig. S4). This suggests that blow-out would not significantly affect microstructures and chemical compositions of the gouge-forming phases. Thus, we believe Fig. 6 gives us important information about the faulting and should be useful.

#1-4. Model of the instability

This part of the paper is very interesting, but remains unclear to me.

- First, how does the kinetics the authors infer from their experiments compare to the already published ones for the silicate olivine (see for instance Perrillat et al. 2015 for a compilation)?

Following the reviewer's suggestion, the data reported by Perrillat et al. (2013 PEPI, 2015 EPSL) were added to Fig. 8. Please note that previous studies other than Perrillat et al. could not be used for Fig. 8 because the nucleation rate, which is required for the calculation of the rate constant K (equation 3), was not determined in the other studies.

- Second, Figure 8 shows the temperature/pressure dependence of the kinetics, in terms of normalized values. But what are the actual values? And again how do they compare to experimental ones and the ones inferred in these experiments.

To compare our calculations with the reported values by Perrillat et al. (2013 PEPI, 2015 EPSL), the vertical axis of Fig. 8 was replaced by the absolute value of the rate constant K . Please see Fig. 8.

- L202 – I believe a minus sign is missing in the first exponential term of growth rate
The typo was fixed.

- L209 - I can see how ΔG_r was calculated, but how was ΔG^*_{hom} calculated.

Please clarify.

We adopted the procedures for the calculation of ‘DeltaG*_hom’ shown in Rubie et al. (1990 JGR). Unfortunately, the procedures are too long to be shown in this manuscript. Therefore, the sentence was modified as follows (please see lines 256-258):

“The values of ΔG^*_{hom} and ΔG_r were calculated from the thermodynamic data of the olivine-wadsleyite transition³⁴ and the olivine-wadsleyite phase boundary¹⁴ (see Rubie et al.³¹ for the details).”

- From eqn 1-3, I am sorry to say I do not understand why/how come there is such a huge difference in the shape of K/Kmax between 13 and 16GPa. Is this effect of the metastability term (1-exp(DeltaG/RT)) in eqn(2) solely?

The cause of the difference in the shape of K - T curves between 13 and 15.5 GPa is the effect of overpressure. At 13 GPa and above 1350 K, wadsleyite coexists with olivine (Fig. 1) and thus overpressure for the nucleation of wadsleyite is almost zero. This is the reason why the K - T curve has a convex shape at 13 GPa. To clarify this point, following sentences were added (lines 266-270):

“Fig. 8 shows that GB nucleation of wadsleyite is accelerated with increasing temperatures. At 13 GPa, however, the GB nucleation rate turns to decrease at temperatures above ~1350 K in the case of $\phi = 1 \times 10^{-4}$ because of the lack of overpressure which induces the nucleation of wadsleyite (i.e., the olivine+wadsleyite coexisting field: Fig. 1).”

- Figure 9a has too much information. What are the colors (blue red & green) and the different symbols (triangles, diamond & stars) referring to? Does each two set of colored curves correspond to the difference of 60°C due to ‘superheating’ by the reaction itself. How was the superplastic flow law inferred? Please reference in the methods/sup.mat the actual equations for each curve.

We agree that the previous version of Fig. 9 has too much information. From the viewpoint of readability, references for each curve were summarized in a box below the figure. Also, detailed procedures for the calculations are shown in a subsection of Methods (lines 672-695):

“At low temperatures and high differential stresses corresponding to the conditions inside deep slabs, dislocation glide is operative but dislocation climb is inactive (i.e., the Peierls mechanism) and thus the creep of olivine is known to be expressed by the following exponential flow law:

$$\dot{\epsilon} = A_{\text{pei}} \cdot \exp \left[-\frac{E^* + PV^*}{RT} \left(1 - \frac{\sigma}{\sigma_p} \right)^2 \right] \quad (8)$$

where A_{pei} is the pre-exponential constant, $\dot{\epsilon}$ is strain rate, and σ_p is the Peierls stress^{15,16}. The critical strain rate for the adiabatic instability (thick-solid curve in Fig. 9a) was calculated from Eq. (4) under the assumption that deformation of the wall rock is controlled by the Peierls creep.

The steady-state creep strengths of ultrafine-grained rocks are more or less controlled by diffusive process (i.e., GSS creep) even at lower temperatures and thus the plastic deformation is described as the power-law equation:

$$\dot{\varepsilon} = A_{pow} \frac{\sigma^n}{G^p} f_w^r \cdot \exp\left(-\frac{E^* + PV^*}{RT}\right) \quad (9)$$

where A_{pow} is a constant, n is the stress exponent, G is grain size, p is the grain size exponent, f_w is water fugacity, and r is the water fugacity exponent. In the case of diffusion creep with $n=1$, Eq. (9) is rewritten as follows:

$$\dot{\varepsilon} = 13.3 \frac{\sigma}{G^2} \left[D_{0,l} \cdot \exp\left(-\frac{H_l^*}{RT}\right) + \frac{\pi\delta}{G} D_{0,gb} \cdot \exp\left(-\frac{H_{gb}^*}{RT}\right) \right] \frac{\Omega}{RT} \quad (10)$$

where D_0 is the pre-exponential factor (l: lattice diffusion; gb: GB diffusion), H^* is the activation enthalpy for diffusion; δ is width of the GB, the Ω is the molar volume. In Fig. 9a, three cases are considered for the deformation of the gouge layer with grain size of 20 nm: i) superplasticity of olivine⁴⁴, ii) diffusion creep of olivine⁴³, and iii) diffusion creep of wadsleyite⁴⁵. Parameters used for the calculations are summarized in Supplementary Table 1. The activation volume of 4 cm³/mol (GB diffusion of silicon in olivine)⁴³ was adopted to the superplasticity of olivine.”

- L269 – *superheating due to direct transition to ringwoodite – this is very interesting and should be emphasized. How does the critical adiabatic shear heating strain rate compares to your flow laws with a 200°C superheating?*

We agree that superheating due to direct transition to ringwoodite is very important. In previous manuscript, we considered the case of superheating with 60 K. The case of 200 K superheating is also considered in the revised manuscript. We found that adiabatic shear heating is enhanced in the latter case. To clarify this point, following sentences were added (lines 325-328):

“An increase in ΔT to ~200 K would induce further weakening of olivine/ringwoodite aggregates (i.e., two orders of magnitude increase in strain rate) and adiabatic instability afterwards, whereas strain rate increase by an order of magnitude is expected with ΔT ~60 K (Fig. 9a).”

#1-5. Inadequate referencing

In a number of places, the reference numbers are wrong. Please check carefully your reference list:

- L52 – *note that Schubnel et al. 2013, Wang et al. 2017 also performed their experiments using a DDIA - Is this considered a conventional apparatus?*

D-DIA was not meant to be a conventional apparatus. To avoid confusion, the phrase “a conventional apparatus” was not used in the revised manuscript.

- L60 – *the reference 8 (Aben et al.2015) seems to be misplaced / irrelevant to the point made here*

We agree, and Aben et al. (2015 JGR) was removed from the sentence.

- L183 – *reference 23 does not correspond to pulverization processes.*

As pointed out by the reviewer, this reference (i.e., Liu et al. 1998 JGR) deals grain size reduction induced by the olivine-wadsleyite transition. Thus, “pulverization of” is

inadequate expression and was replaced by “reduction in” (lines 228-231).

“Intracrystalline transition via nucleation of wadsleyite on high-density dislocation walls, which have a highly disordered structure like grain boundaries (GBs) in olivine grains, would contribute to the reduction in grain size of olivine²⁸.”

- *I stopped checking after this, sorry...*

We carefully checked the all references cited in the manuscript. Some references were replaced by appropriate ones.

#1-6. Conclusion

Maybe the major flaw of the paper is its lack of actual conclusion, and one falls short of understanding what the authors want to say. The conclusion is replaced by a last paragraph of discussion (l270-289), which attempts to bridge with the larger scale and reports on b-value variability. How is this really related to the present study, since the authors reported no experimental b-value?

As pointed out by the reviewer, the last paragraph should correspond to the conclusion of this study. We admit that the topics about the b-value and the thickness of metastable olivine wedge (MOW) are not directly related with this study and rather overstated. Thus, such sentences were removed from the manuscript, and we modified the last paragraph to show our conclusions more clearly (please see lines 329-345).

“Seismic imaging of the deep mantle demonstrates the presence of the MOW, accompanied by deep-focus earthquakes, as a low-velocity zone with a thickness of tens of kilometers inside the cold slab^{53,54}. The present study suggests that the deep-focus earthquakes nucleate when the temperature of the MOW reaches those close to 1100–1160 K at pressures of the mantle transition region. Here the nanocrystallization of olivine occurs upon the phase transition (i.e., formation and linking-up of the lenticular packets filled with nanocrystalline olivine/wadsleyite) followed by throughgoing faulting due to shear instability (Fig. 9b), although this critical temperature may be lower on the geological time scale (Fig. 8). In fact, the observed deep-focus earthquakes are reported to be located along an isotherm of ~1000 K in deep slabs⁵⁵ and can propagate outside the MOW³⁸. Numerical studies based on experimental kinetic data show that metastable olivine in cold slabs probably persists to depths of ~630 km, below which ringwoodite is the major phase down to ~700 km (ref.⁵⁶), where it breaks down to bridgmanite and ferropericlase. The shear zones filled with nanocrystalline olivine required for throughgoing faulting would not be preserved in subducted slabs at depths beyond this kinetic boundary of ~630 km, consistent with the abrupt decrease in seismicity at depths below ~600 km (ref.²).”

#1-7. Minor comments

L24 – the occurrence of deep EQs

We corrected the sentence according to the reviewer’s comment.

L93 – *ambiguous triple negation (neither, nor, not) – please reword*

The sentence was modified as follows (lines 101-104):

“While the highest strength at each temperature is close to the value calculated from the Peierls creep of wet olivine¹⁷, neither dislocation creep¹⁸, nor dislocation-accommodated grain boundary sliding of wet olivine¹⁹ account for the observed strengths.”

L124 – *Inadequate wording – velocity weakening refers to a rate and state friction concept, where the strength decreases with increasing sliding velocity. Maybe you meant that that the softening due to an increase in temperature also leads to an increase in strain rate/decrease in flow strength?*

We agree, and the term “velocity weakening” was removed from the sentence. The sentence was modified as follows (lines 148-149):

“The softening was likely to be initiated by temperature ramping...”

L126 – *comparison of (T/epsilon)_dot with nature – please place this sentence either in the discussion, or in the methods. It seems odd here.*

Following the reviewer’s advice, the sentence was moved to Methods (please see lines 524-526).

“The ratios of temperature-ramping rate and strain rate in the path#3 runs ($T/\dot{\epsilon} = 1.0\text{--}1.4 \times 10^4$ K) are comparable to those expected values for natural subduction zones⁶⁰.”

L136 – *“and also to quit the run to examine the sample” – I don't understand what you mean? Please reword*

The sentence was modified as follows (lines 155-159):

“Because the X-ray radiographic imaging was interrupted by the acquisition of each diffraction pattern with a long exposure time (70–540 seconds), it was hard to determine the whole faulting process solely by the in-situ measurements. Thus, microstructures of the recovered samples were also carefully examined to obtain complementary evidence for the process after the deformation experiments.”

Response to comments by the reviewer#2

#2-1. Imprecisions, misconceptions and misleading statements

Throughout the manuscript, several inaccurate statements can be listed, as well as imprecisions on either observations or interpretations. Furthermore, I report some confusion between observations and interpretations. I describe these in details in the line-by-line comments.

We really appreciate the reviewer’s detailed invaluable comments and thoughtful suggestions. Our point-by-point responses are shown below.

Numerous mistakes can be listed throughout the paper, some of which directly induced by a lack of consideration for previous studies (major comment #2). First, the depth limit of seismicity within the solid Earth is not 680 km, as deep-focus earthquakes were reported deeper, up to ≈ 750 km depth (Kuge, 2017; Kiser, 2021).

We agree, strictly speaking, there are few earthquakes even below 680 km, though the frequency of the occurrence of such earthquakes is extremely limited. The sentences about the depth limit of seismicity were modified as follows (lines 43-46)

“An abrupt decrease in the seismicity from a peak at ~ 600 km to almost zero at 680 km (ref.²), with a very limited number of earthquakes at deeper regions down to 750 km (ref.³), also implies an essential role of the phase transitions of olivine on occurrence of deep-focus earthquakes.”

Not only olivine, but also other minerals, e.g. pyroxenes (Shi et al., 2017; Xu et al., 2018), have been proposed to trigger seismicity upon destabilization in the MTZ.

The main body of a subducting slab, where the deep earthquakes take place, is believed to have a harzburgitic composition. Olivine is the most dominant mineral, occupying ~ 80 vol.% of the bulk rock (Irifune and Ringwood, 1987 EPSL), in such a depleted mantle composition. Thus, we believe the effect of (metastable) phase transitions in other minerals, such as pyroxene and garnet, on the occurrence of the deep earthquake is relatively minor or even negligible.

In their reference to the Peierls creep, the authors favor a mechanism and do not recall the alternative mechanisms, which does not give a good impression (major comment #3).

We revised the manuscript according to the comment. Please see our response#2-3 for details.

Similarly, the authors favour the partial melting interpretation, although contrary to what they claim their observations (and previous works) are mostly consistent with partial crystallization of the melt (major comment #5).

We do not exclude the possibility of partial crystallization (i.e., crystallization after total melting). The term “partial melting” was replaced by “melting” in the revised manuscript. Please see our response#2-5 for details.

Several terms are problematic, such as “melt-like phase” (lines 160-161).

The term “melt-like phase” was replaced by “iron-rich phase” in this manuscript. Please note that the iron-rich phase is amorphous, which is most likely the product of partial/total melting of M2676 sample.

I point out that “low-viscosity fluid” (lines 49-50) does not make sense, for several reasons. By definition, “fluid” as an adjective is the opposite of “viscous”; as a noun, it is the opposite of “solid”.

We agree, and the term “fluid” was replaced by “layer”. Please see lines 52-53.

“..., that behaves as a low-viscosity layer due to grain-size-sensitive (GSS) creep^{4,5,7}.”

I request that the authors clarify as much as possible everything, especially the methods and the figures. For example, it should be specified that the “mechanical data” consists of the stress state as seen via X-ray diffraction of olivine crystals.

We agree, and “mechanical” was replaced by “mechanical (pressure, stress, and strain)” in the 1st sentence of the caption of Fig. 3. Other technical terms were also clarified in the manuscript.

“Figure 3 | Mechanical (pressure, stress, and strain) and acoustic records in the OL100 sample faulted at 1100 K (M3425).”

There is a critical problem with the electron diffraction patterns in Fig.4d and Fig.5e, which should not be truncated and should be accompanied with a scale (in nm⁻¹).

The spots of the electron diffraction patterns in Figs. 4d and 5e are shown with Miller’s index (i.e., hkl), meaning that phase identification is succeeded. In such a case, it is a common practice not to show a scale bar in the indexed diffraction pattern (please see other papers by TEM experts, for example *Miyajima et al., 2006 GRL*), because indexed patterns already contain information on the scale. On the other hand, we agree that the electron diffraction patterns should not be truncated. The pattern shown in Fig. 5e was replaced by the original one (please see the revised Fig. 5e).

Also, the “no Wad” purple boxes should be moved below subfigures (just below the time axis) for sake of clarity.

Because the reviewer#1 pointed out that the “no Wad” purple boxes are not informative, the boxes were removed from the figures (please see our response#1-3).

Equations and associated descriptions are not presented in the right order.

Equations and the associated descriptions were carefully checked, and some of them were modified according to the comment.

Finally, the authors could shorten several sentences by removing redundant information. The authors state that brittle failure is “unlikely” at depth, which results from general misconceptions regarding deep earthquakes and their experimental analogs. Contrary to what the authors claim, the actual seismicity is the proof that brittle failure exists at

pressures up to ≈ 25 GPa (deep-focus earthquakes) in Earth's mantle. The fact that a large part of the community is unable to understand how it works is not a reason to state that these ruptures would be "unlikely". They are not unlikely, otherwise they would not occur so frequently. Nonetheless, it is true that high pressures and/or temperatures do not favor the brittle failure of rocks, which explains the interrogations regarding both deep triggering mechanisms (critical stress distortion generating the mechanical instability) and deep rupture mechanisms (nucleation and dynamic propagation). → See major comment #3.

Here we meant "brittle failure" to be "conventional brittle failure" for shallow earthquakes. We replaced the sentence with a more commonly used phrase such as used, for example, in the 1st paragraph of Gleason and Green (2009 PEPI) is "Conventional brittle failure is impossible in materials lacking a fluid phase at pressures exceeding approximately 2.5 GPa, ...". (lines 21-22):

"..., because conventional brittle failure is unlikely to occur at elevated pressures."

The paper does not look like a revised manuscript. Vocabulary issues should be fixed to avoid confusion, especially if a broad readership is targeted. Grammatical issues should also be verified throughout the text (e.g. missing "the"). At several instances (detailed in the line-by-line comments), the paper is hard to follow due to grammar or syntax issues. In some paragraphs, the mixture of conceptual exaggerations and language issues tend to prevent the reader to remain focused.

We admit it is rather difficult for us from a non-native English-speaking country to use the term like "the" correctly, as pointed out by the reviewer. We carefully checked grammar and syntax issues, and the revised manuscript was also checked by two scientists who are native English speakers. We believe the vocabulary issues are now fixed.

#2-2. Lack of consideration for previous works and illegitimate attacks

There is a serious lack of reference for key studies associated with the development of the techniques used in the present work (e.g. Wang et al., 2004; Gasc et al., 2011). The authors should better acknowledge the studies that inspired the present work.

We admit that we were forced to remove some references mainly because of the tight limitation on the number of references in *Nat. Commun.* We additionally cited a pioneering work (about the development of the deformation-DIA apparatus) by Wang et al. (2003 Rev. Sci. Instrum.), according to the suggestion. Please see lines 477-478.

"We conducted deformation experiments on the OL100 and OL92 samples using a deformation-DIA apparatus⁵⁹ combined..."

They should recall that the stress calculation technique from the diffraction of high-energy X-rays in the synchrotron comes from the key development reported in several studies (e.g. Uchida et al., 1996; Hilaiet et al., 2012; Merkel & Hilaiet, 2015), not only Singh et al. (1998).

We understand that the previous studies listed by the reviewer are important works. Unfortunately, however, the total number of references is limited up to 70 in *Nat. Commun.* Please understand that we already cited the maximum number of the papers in the manuscript. Singh et al. (1998 *J. App. Phys.*) is a pioneering work on the stress calculation from the diffraction of high-energy X-rays. Thus, we selected this paper as a representative work.

They should properly describe the analysis tools they use: the Multifit/Polydefix framework (Merkel & Hilaiet, 2015) or any other protocol.

We used software developed by Dr. Seto for the analysis. In the revised manuscript, Seto (2010 *Rev. High Press. Sci. Technol.*) is cited (lines 541-544):

“Peak positions of four diffraction peaks of olivine (hkl = 021, 101, 130, and 131: Supplementary Fig. 5c) in each subdivided sector were semi-automatically determined in pressure and stress measurements using software (IPAnalyzer and PDIndexer)⁶¹.”

Finally, note that the provided definition of the AE magnitude is used by previous studies on similar lithologies (e.g. Schubnel et al., 2013; Ferrand et al., 2017; Kita & Ferrand, 2018).

We agree, and we cited Lei and Sato (2007 *Tectonophys.*) for the definition of the AE magnitude (please see lines 597-599).

“...that is, $M_{ae} = \log_{10} V_{max}$. Note that the values of M_{ae} obtained in this way ought to be considered only as first-order approximations of the relative values⁶⁶.”

Regarding the research topic, some key studies of the recent literature are ignored. Thus, the discussion on mechanisms is incomplete (major comment #3).

Please see our response#2-3 for details. We carefully selected very important studies for citation. Please understand that we needed to select them due to the limitation in the number of citations.

The overinterpretation favoring partial melting also ignore studies reporting partial crystallization (major comment #5).

As mentioned above, we do not exclude the possibility of partial crystallization (after total melting). Please see our response#2-5 for details.

[In addition, this study looks like a high-pressure equivalent of the TROPICO project on $Mg_2GeO_4 + 8\% MgGeO_3$, which I proposed to the JSPS in 2017 and 2018 and was rejected 3 times in a row.]

We are sorry to say that this is the first time to hear about the “TROPICO project”. In this study, we deal silicate olivine $(Mg,Fe)_2SiO_4$, not germanate olivine. The former one requires at least pressures of 13 GPa for phase transition. On the other hand, 2 GPa is enough for the phase transition of germanate olivine. We consider that our experimental conditions/techniques are quite different from those for the “TROPICO project”.

On top of that, I have noticed inappropriate sentences, which artificially reduce the importance of previous works, although the present study is the continuation of these works. The authors state that the analogs used by previous studies may not be adequate but minimize the legitimate reasons why these analogs have been used. In spite of differences, mentioned by numerous studies (e.g. Green & Burnley, 1989; Green et al., 1990; Schubnel et al., 2013; Ferrand & Deldicque, 2021), both Mg_2GeO_4 and $(Mg,Fe)_2SiO_4$ transform to a spinel or spineloid structure with increasing pressure. I highly recommend that the authors read the paper of Riggs & Green (2005) reporting the exact same microstructures in Mg_2GeO_4 as Officer & Secco (2020) in Fe_2SiO_4 , i.e. coeval development of spinelfilled anticracks and dislocation stacks within olivine.

We do appreciate the earlier works using analogue materials but the point here is that Mg_2GeO_4 olivine transforms directly to the spinel structure, as is the case for Fe_2SiO_4 olivine suggested by the reviewer, unlike mantle olivine. It is good idea to use analogue materials when the high-pressure technology is premature and does not reach the actual pressure and temperature conditions of the deep mantle, but we believe efforts should be done to develop the technology to reproduce more realistic situations to minimize the ambiguity using analogue materials, as demonstrated in our present paper. This is clearly seen in the history of high-pressure geoscience; e.g. the transformation of olivine to modified spinel was discovered only in Mg_2SiO_4 and $(Mg,Fe)_2SiO_4$ compositions close to those of mantle olivine, while this transformation does not exist in the analogue compositions. Nevertheless, we have modified the sentence to avoid such misunderstanding and unnecessary displeasure, as follows (lines 53-55):

“However, Mg_2GeO_4 olivine transforms directly to the spinel phase without passing through the modified spinel phase⁸, implying that this might not be a perfect analogue of mantle olivine⁸.”

In addition, the authors cite Green et al. (1990) at lines 52-53 in a way that could trigger doubt, and here I recall that this seminal study reported anticracks in natural minerals and not only in analogs. For the sake of fairness and to promote a mutual respect

atmosphere, I request that the authors rephrase these sentences (see line-by-line comments).

We definitely value the importance of the anticrack model and highly respect a series of related works. However, because of the limitations in text length and number of references, we need to avoid redundant expressions and just cited Green et al. (1990 Nature) as a pioneering work on silicate olivine (Mg,Fe)₂SiO₄. Please see lines 55-58: “A pioneering work on (Mg,Fe)₂SiO₄ olivine⁸ pointed out the possibility of faulting at the onset of the olivine-wadsleyite transition at 15 GPa and 1550–1650 K based on the microstructural observations of recovered samples.”

#2-3. Mix between triggering mechanisms and rupture mechanisms

I note that the authors and most reviewers continue to mix the triggering, nucleation and dynamic propagation mechanisms altogether, which is one of the major causes of the current (apparent) mess on the topic. Therefore, hereinafter I explain it all again, with the hope that everyone understand. Recent studies (and my own PhD thesis) have clearly explained that the triggering mechanisms of initial mechanical instabilities and the rupture (nucleation and propagation) mechanisms should be distinguished (e.g. Ferrand, 2017; Ferrand et al., 2021; Mao et al., 2022). A rupture is a rupture, i.e. a dynamic process with more or less efficient lubrication that depends on the overall conditions of the tearing material (Ferrand et al., 2021). “Transformational faulting” is a vague concept that we need to consider as a transitional tool towards a better understanding of deep earthquakes. Numerical modeling of thermal runaway helps understand the dynamics of rupture propagation (Thielmann et al., 2015; Thielmann, 2018a) but it is not sufficient to explain the entire rupture process, especially the nucleation stage (Thielmann, 2018b; Ferrand et al., 2021). We have published a review of evidences of rupture-induced melting in various lithologies from the lab and from the field, with numerical modeling of the energy balance at the advancing melting front (Ferrand et al., 2021). Ferrand et al. (2021) study the dynamic rupture mechanism (which should be distinguished from the nucleation mechanism). Rupture-induced melting (due to shear heating) is also proposed as the most likely mechanism for deep seismic ruptures by recent seismological studies both around and within the metastable olivine wedge by Zhan (2017) and Mao et al. (2022), respectively. Actually, our recent study (Mao et al., 2022) highlights very high b-values of deep seismicity for $M_w < 3.8$ (JMA catalog, i.e. with the smallest completeness magnitude), most likely related to the hydration state (OH content) of cold subducting slab segments. So not only temperature, although the arguments about temperature are important as well. Alternative/complementary mechanisms, such as thermal pressurization (e.g. Wibberley & Shimamoto, 2005; Viesca & Garagash, 2015) and pulverization (e.g. Reches & Dewers, 2005; Aben et al., 2016; Incel et al. 2019) could also be mentioned. In addition, the triggering mechanism (local transformation) and

rupture mechanisms (e.g. transient melting due to shear heating of either olivine peridotite or wadsleyite peridotite) are discussed by Mao et al. (2022). The triggering mechanism of deep earthquakes describes how the initial critical distortion of the stress field is generated. They would mostly be related to olivine phase transitions (e.g. Green & Burnley, 1989; Green et al., 1990; Schubnel et al., 2013) but also possibly to the destabilization of other minerals such as pyroxene (Shi et al., 2017; Xu et al., 2018). The triggering mechanism describes neither the rupture nucleation nor the rupture propagation. It describes only the origin of the critical stress distortion upon local transformation, as detailed for intermediate depths (Ferrand, 2019a). Regarding the initiation of the shear crack instability (nucleation mechanism), it would be fair to mention the reaction-induced grain size reduction (Incel et al., 2017; Thielmann, 2018b) and the grainboundary disordering mechanism (Ferrand & Deldicque, 2021).

First, we appreciate the reviewer's long and thoughtful comments on the triggering and rupture mechanisms. The rupture-induced melting model recently developed by the reviewer's group is indeed quite interesting, which is also cited here. However, we don't think all of the references cited by the reviewer are related to the mechanism of the deep-focus earthquakes. Besides, we have some strict limitation in the number of the references in *Nat. Commun.*, which is very different from the discussion in other specialized journals, and are forced to some selected works really related to the present study, as we emphasized repeatedly in the above. It is true that "transformational faulting" is a vague concept, as pointed out by the reviewer, but it is also true that we need to develop this kind of concept as a transitional tool towards a better understanding of deep-focus earthquakes, as firstly shown experimentally here under the pressure and temperature conditions equivalent to those of the MTZ with realistic mantle olivine compositions.

The reviewer #1 also asked for a better consideration of alternative mechanisms, including GB processes. Please note the grain-boundary instability reported in Mg₂GeO₄ in a narrow temperature window, which can constitute an efficient viscosity reduction mechanism before reaching the melting conditions.

In the 1st paragraph, we already reviewed grain-size-sensitive creep (i.e., the grain-boundary instability) induced by phase transition. The importance of grain-size-sensitive creep has been widely accepted by many researchers, and thus three previous studies (Green & Burnley, 1989 *Nature*; Schubnel et al., 2013 *Science*; Green et al., 2015 *Nat. Geosci.*) are cited in the manuscript. Please see lines 50-53.

"The anticrack is orientated normal to the direction of the maximum compressive stress and is filled with a nanocrystalline aggregate of the high-pressure phases, that behaves as a low-viscosity layer due to grain-size-sensitive (GSS) creep^{4,5,7}."

Regarding the “narrow temperature window” of the grain-boundary instability, we showed that grain boundary nucleation of wadsleyite in the semi-brittle regime is effective at a narrow temperature window of 1100-1160 K. Please see Fig. 8 and the 1st paragraph of Discussion (especially, lines 270-274).

“As microcracking would be ineffective at temperatures above 1200 K, indicated by the very low AE activities (Fig. 2b), the maximum transition rate on GBs in the semi-brittle regime should be realized at temperatures of 1100–1160 K under these pressures, consistent with the occurrence of throughgoing faulting only in the runs around this temperature.”

I think that the authors should recall that transformations (e.g. phase transitions, grain-boundary disordering, amorphization, melting...) can be important deformation mechanisms (Wheeler, 2020; Green et al., 2015).

As shown in this study, phase transitions of olivine and melting play important roles in the occurrence of deep-focus earthquakes. Please see Abstract and Fig. 9. Other mechanisms (grain-boundary disordering and amorphization) may contribute to the occurrence of intermediate-depth earthquakes, but not for deep-focus earthquakes.

The mentioned Peierls creep is a low-temperature high-stress mechanism related to the Peierls stress, i.e. stress required to trigger dislocation glide in crystals. Several other mechanisms could be discussed in the paper. See a review of mechanisms by Ferrand & Deldicque (2021).

We discussed not only Peierls creep but also many other creep mechanisms (dislocation creep, dislocation-accommodated grain boundary sliding, superplasticity, and diffusion creep) in Figs. 2 and 9. In Table S1, we listed the flow-laws considered in this study. Please see Table S1.

I think that the authors should recall the different kinds of grain-boundary sliding/rotation mechanisms, including the transformation-assisted grain-boundary sliding proposed to explain the viscosity reduction in Mg₂GeO₄ in a narrow temperature window (1000-1150°C). The “transformation” of the transformation-assisted GBS can be any kind of grain-boundary destabilization, including phase transition (Green et al., 2015), grain-boundary premelting/disordering (Takei et al., 2019; Yamauchi & Takei, 2020; Ferrand et al., 2021), melting (present study; Kloe et al., 2000) or amorphization (Samae et al., 2021).

As mentioned above, we showed that grain boundary nucleation of wadsleyite in the semi-brittle regime is effective at a “narrow temperature window” of 1100-1160 K. Please see Fig. 8 and the 1st paragraph of Discussion. We consider that a diffusion-

accommodated grain-boundary sliding (i.e., superplasticity) of olivine could be the process controlling the strength of fault. Please see lines 292-295:

“Thus, the adiabatic instability can be initiated by the formation of a nanocrystalline olivine layer (due to linking-up of the lenticular packets: Fig. 9b) followed by shear localization accelerated by diffusion creep⁴³ or superplasticity of olivine⁴⁴ (Fig. 9a: see Methods for the details of calculations).”

Please note that we already reviewed grain-size-sensitive creep (including grain-boundary sliding) in the 1st paragraph. Regarding the transformation-assisted grain boundary sliding, we cited Green et al. (2015 Nat. Geosci.) as a representative study. Please see lines 50-53.

“The anticrack is orientated normal to the direction of the maximum compressive stress and is filled with a nanocrystalline aggregate of the high-pressure phases, that behaves as a low-viscosity layer due to grain-size-sensitive (GSS) creep^{4,5,7}.”

As explained in major comment #6, the present study sheds new light on the nucleation mechanism of deep ruptures in the MTZ, and confirms the melt-assisted propagation mechanism.

We appreciate the reviewer’s positive comment.

#2-4. Confusion between seismic “faulting” and coalescence “faulting”

Throughout the paper, there is a confusion between single-event “faulting” due to seismic or aseismic rupture processes (e.g. Aubry et al., 2018; Ferrand et al., 2017;2018) and sample-scale “faulting” due to the (final) coalescence of individual faults (e.g. Shen et al., 1995; French & Zhu, 2017; Gasc et al., 2017). In such experiments, the coalescence is controlled by the single-event faults that form first (associated with the reported acoustic activity), but also by the sample dimensions and rheological contrasts between the sample and the other materials within the deformation assembly. The word “faulting” is problematic; thus, a proper definition is required.

Based on microstructural observations, we conclude that “Linking-up of the lenticular packets with nanocrystalline olivine and wadsleyite should be the origin of weak fault gouge layers...” (lines 236-237). Our model (Fig. 9b) should be distinguished from the usual faulting mechanisms including rupture process and/or coalescence of individual faults, and we agree that the term “faulting” might be a vague phrase. To avoid confusions, “faulting” was replaced by “throughgoing faulting” in the revised manuscript.

As I have explained in details (Ferrand et al., 2021), micropseudotachylytes (microfaults) that are confined inside the sample (i.e. fault length smaller than the sample size) are

more representative of natural seismic faults than sample-scale faults (affecting the deformation assembly). I recall that the largest rheology contrasts are not within the sample but between the different parts of the assembly. It would have been interesting to see the single-event faults corresponding to the reported labquakes (before coalescence), because the coalescence usually reworks/connects only some of them during the formation of the sample-scale fault zone, and such confined faults (before coalescence) are not contaminated by the other part of the experimental assembly (see major comment #5) and can give important insights on rupture mechanics (e.g. Ferrand et al., 2017).

Yes, it would be interesting if single-event faulting is detected in experiments. Unfortunately, such observations are quite difficult at this time, and the experimental technique needs to be improved to test this hypothesis. We believe this kind of experiment is a future important target in our community.

#2-5. Overinterpretation regarding the transient melting stage

The authors claim that partial melting occurs during their experiments, but they do not provide convincing evidence to conclude that the melting would actually be “partial”. The interpretation in terms of incongruent melting is not reasonable and seems contradictory with the reported results. The composition of the melt could be explained by either partial melting or partial crystallization.

Presence of melt-like phase in the sample (Figs. 4-6) and its solidus temperature provide us the information about the lower-limit value of the peak temperature. We presume that “partial crystallization” means partial crystallization after total melting above liquidus temperatures.

We agree that our results cannot exclude the possibility of total melting of olivine during throughgoing faulting. Thus, the term “partial melting” was replaced by “melting” in the revised manuscript.

Regarding run M2676, we found that peak temperature reached 2200-2450 K (i.e., melting of platinum-iron alloy) during the throughgoing faulting. This temperature range is overlapped with the liquidus temperature of dry harzburgite (~2270 K: Fig. 1a), meaning that the possibility of total melting cannot be excluded. Thus, we conclude that the observed amorphous phase is a quenched product after the total melting or partial melting of the sample. To clarify these points, following sentences were added (lines 204-209):

“Formation of platinum-iron alloy blobs in the M2676 sample (Fig. 5c) shows that the temperature crossed the solidus line for dry harzburgite (2140 K at 13 GPa)²⁶ and the liquidus curve for the platinum-iron alloy (~2200–2450 K at 13 GPa)^{22,23} during throughgoing faulting. Partial and total melting of the gouge are expected in the cases that the peak temperatures are below and above the liquidus curve for dry harzburgite (~2270 K at 13 GPa)²⁶, respectively (Fig. 1a).”

Partial melting is a well-known process affecting the mantle in fixed P-T conditions, responsible for the significant chemical difference between mantle and crustal rocks. In contrast, pseudotachylytes (either natural or experimental), report local but total melting (see Ferrand et al., 2021 for a review). The fossilization of a multi-phase system after the seismic event would indicate partial crystallization of the melt during the quenching stage (deceleration), i.e. late coseismic creep (Ferrand et al., 2018). Rupture-induced total melting is observed in both experimental (e.g. Nielsen et al., 2010; Di Toro et al., 2011; Hayward & Cox, 2017; Lockner et al., 2017; Aubry et al., 2018) and in natural rocks (e.g. Andersen et al., 2014; Ferrand et al., 2018), with subsequent crystallization documented during cooling soon after mantle ruptures (Ferrand et al., 2017; 2018).

We have read Ferrand et al. (2021 JGR), which was published right after we prepared our manuscript. The studies listed by the reviewer are about granitic rocks with low solidus temperatures (< 1000 K) and at pressures below 0.5 GPa, corresponding to the conditions for shallower-depth earthquakes (at depths < 20km). On the other hand, our target is the mechanism of deep-focus earthquakes occurring at much higher pressures above 13 GPa (depths > 400 km). The material used in our study is olivine, while it is more siliceous granitic rock in these studies. As the melting behaviors of these materials are quite different and the nature of quench products varies with pressure, we are sorry to say that the interpretations for the results on the different material at much lower pressures are not directly applicable to the phenomena relevant to deep-focus earthquakes.

In the HP pseudotachylyte of the Balmuccia peridotite, fractionation is seen along “injection” veins (i.e. tension cracks; Ferrand et al., 2018).

The Balmuccia peridotite came from the depths ~40 km, which may not be related to the mechanism of deep-focus earthquakes as mentioned above.

I do not see in the present paper sufficient evidence to conclude in favor of partial melting. If the authors want to stick to the partial melting interpretation, they should clearly detail the alternative possibilities and properly explain why their hypothesis is favored. Importantly, considering competent rocks under pressure, the temperature reached right at the tip of the shear crack is high enough to melt anything on the rupture path (especially for fresh peridotites, for which melting points of all minerals are relatively close to each other).

We are not sticking to the partial melting interpretation. As mentioned above (our response#2-5), the term “partial melting” was replaced by “melting” in the revised manuscript. Also, we found that total melting followed by partial crystallization may

occurred in run M2676.

In illustrations, the authors should show not only the solidus but also the liquidus (partial vs total melting).

We agree, and the liquidus curves for dry lherzolite and forsterite were added to Fig. 1a.

If the authors are interested in understanding the recent achievements about seismic rupture dynamics, I would be happy to explain it in details. They can read the recent comparative study by Ferrand et al. (2021), considering various works from either field geology or laboratory experiments and proposing modeling of the advancing melting front as a Stefan problem. → See major comment #3.

We appreciate the reviewer's kind suggestion. We found that Ferrand et al. (2021 JGR) is a very nice piece of work, and we cited it in the section of Introduction. Please see lines 60-61:

"... that the development of substantial fault slip and the propagation of the rupture requires an additional mechanism such as adiabatic instability¹⁰⁻¹²."

The authors claim that they observe "iron-rich nanoparticles (FeO/MgO ~0.4–0.9)" associated with "silicon rich patches (SiO₂ = 62-84 wt.%)" that would be "distributed in a gouge layer developed in the OL100 sample of M3100 (Figs. 5f-h and 6b)." They further claim that they observe SiO₂ particles, interpret it as stishovite and propose that it would originate from incongruent melting of wadsleyite. All of this is overinterpretation. First, the Fe-rich phase is documented in samples M2676 and M3425, whereas the "Si-rich path" is only reported in sample M3100, so these two phases are not "associated".

As shown in Fig. 1a, incongruent melting of olivine polymorphs to SiO₂ stishovite + melt is plausible at pressures > 14 GPa according to the melting relations in the Fe₂SiO₄ system (Ohtani, 1979 J. Phys. Earth). On the other hand, incongruent melting never occurs in the olivine+opx system (M2676 run) at ~13 GPa, which is also consistent with the melting relations. As the reviewer commented, SiO₂ particles should be present in M3425 sample, but we could not find any SiO₂ particles in the sample probably because SiO₂ particles are very tiny and the amount is very limited. We believe our interpretation is valid. Nevertheless, we agree that there is a slight chance of misleading, and our statement was toned down (line 191-194), as suggested by the reviewer:

"Formation of both the iron-rich and silicon-rich phases may be explained by incongruent melting of olivine polymorphs to an iron-rich liquid and stishovite at a peak temperature of >2500 K (at 15.5 GPa: Fig. 1a) if we adopt a melting relation of Fe₂SiO₄ (ref.²⁴)."

However, Ohtani and Kumazawa (1981 PEPI) reported an incongruent melting of

Mg₂SiO₄ forsterite to MgO periclase + melt at pressures of 8-20 GPa. If this is applicable to the (Mg,Fe)₂SiO₄ olivine system, the observed SiO₂ particles cannot be explained by the incongruent melting of olivine polymorphs. The origin of the SiO₂ particles can be attributed to a non-equilibrium process. To clarify this point, following sentences were added (lines 194-197):

“Taking an incongruent melting of forsterite (i.e., α -phase of Mg₂SiO₄) to a liquid and periclase (i.e., MgO) at a temperature above 2800 K (at 15.5 GPa)²⁵ into account, the possibility that the observed silicon-rich phase is a metastable phase or a quench product cannot be excluded.”

Second, on Fig.5 (sample M3100), no SiO₂ is documented.

In Fig. 5g, the SiO₂ phase is indicated as “Si-rich patch”.

Third, on Fig.6b the reported compositional trend could well be explained by the resolution limit of EDS measurements, because the local patches are small and close to minerals of other compositions. The authors should show where the data are collected

The gouge layers such as Fig. 5g were analyzed, and the results are summarized in Fig. 6b. To clarify this point, the 2nd sentence of the caption of Fig. 6 was modified as follows:

“(b) Chemical compositions of olivine polymorphs (open triangle), majorite (solid triangle), silicon-rich patches (Si-rich: square), and iron-rich phase (Fe-rich: circle) in the gouge layer developed in M3100 sample (e.g., Fig. 5g).”

and check whether or not the observed trend could be due to contamination of the EDS signal by neighboring phases.

The variation in the chemical compositions of the silicon-rich patches is due to the contamination of other phases in the EDS analyses. We admit that our explanation was insufficient. To clarify this point, following sentences were added (lines 185-190):

“The mixing line along olivine polymorphs and the silicon-rich patches (Fig. 6b) implies that the main constituent of the silicon-rich patches is a SiO₂ phase (i.e., stishovite under the experimental conditions). The variation in the chemical compositions of silicon-rich patches (Fig. 6b) would be due to contamination of the energy dispersive X-ray (EDX) spectra from the neighboring grains of olivine polymorphs (or majorite) in the analyses of the SiO₂ phase.”

The EDS method is semi-quantitative and one should be very careful with absolute values. The authors should provide a justification for the uncertainty range, especially for the ultra-fine grain materials and the Fe-rich grain-boundary films.

We agree, and following sentences were added to Methods to clarify the points (lines 650-653, 656-658):

“The cores of each crystal having a diameter larger than 100nm (except for cases of ultrafine grains/patches) were selected for quantitative analysis with a probe size of ~2 nm to avoid the contamination of spectra from the neighboring grains.”

“Chemical compositions are recalculated as 100 wt.% anhydrous. The uncertainty of the calculated chemical composition reflects the error in the counts of characteristic X-rays.”

One could hypothesize that the documented Si-rich patch would consist of nanograins of stishovite lost in the middle of some amorphous material, but this interpretation seems really exaggerated. The presence of stishovite is not really compatible with the rheology. Sample M3100 is dunite, and stishovite is not expected in dunite. In such a rock, the Si-rich phase can only come from the crystallization of the rupture-induced melt or be a pre-existing phase (which is likely the case on Fig.5g). Stishovite can only exist in pyroxene-rich compositions at $P > 16$ GPa (Fig.R1).

Stishovite can exist in the wadsleyite/ringwoodite system as incongruent melting occurs in olivine composition (Ohtani, 1979 J. Phys. Earth), meaning that the presence of pyroxene-like compositions, as mentioned by the reviewer, is not mandatory for the precipitation of stishovite. Incongruent melting means dissociation of wadsleyite/ringwoodite to melt + stishovite. Ohtani reported the incongruent melting of fayalite (i.e., Fe_2SiO_4 olivine) at pressures above ~ 13 GPa.

However, Ohtani and Kumazawa (1981 PEPI) reported an incongruent melting of Mg_2SiO_4 forsterite to MgO periclase + melt at pressures of 8-20 GPa. If this is applicable to the $(\text{Mg,Fe})_2\text{SiO}_4$ olivine system, the observed SiO_2 particles cannot be explained by the incongruent melting of olivine polymorphs. In this case, the SiO_2 particles were formed via a non-equilibrium process. Please see our response shown above for details.

In addition, the only “Si-rich patch” showed by the authors (Fig.5g) does not seem to originate from melting. It is clearly a pre-existing phase, cut and shifted by a branch of the fault zone that I highlight in the reinterpretation of Fig.R2. This phase could be minor enstatite, frequent with San Carlos olivine, but the authors could propose some other mineral richer in Si.

We appreciate the reviewer’s discussion with his figure R2. Fig. 6b shows that the Si-rich patches contain very high SiO_2 compositions of 62–84 wt.%. As shown in Fig. 6b, major minerals are limited to wadsleyite/ringwoodite ($\text{SiO}_2 \sim 42$ wt.%) or majorite (i.e., a pyroxene-like phase with $\text{SiO}_2 \sim 48$ wt.%) in our experiments. As discussed in the manuscript, we believe that precipitation of stishovite nanocrystals is the origin of the Si-rich patches with $\text{SiO}_2 = 62\text{--}84$ wt.%.

In any case, I request that the authors show the acquisition location of the data presented in Fig.6b.

As already mentioned above, the gouge layers such as Fig. 5g were analyzed, and the results are summarized in Fig. 6b. To clarify this point, the 2nd sentence of the caption

of Fig. 6 was modified as follows:

“Chemical compositions of olivine polymorphs (open triangles), majorite (solid triangles), silicon-rich patches (Si-rich: squares), and the iron-rich phase (circles) in the gouge layer developed in the M3100 sample (e.g., Fig. 5g).”

I would suggest to summarize the experimental series using a P-T diagram, to help the reader (and maybe the authors) better understand the significance of this work. See an example for sample M3100 in Fig.R3. At 15.6 GPa (M3100), a temperature of 1160 K ($\approx 890^{\circ}\text{C}$) is the condition for the transition from wadsleyite to ringwoodite. This transition may influence the relationship between deformation and transformation during the experiment (major comment #3).

We appreciate the reviewer’s kind suggestion. We agree, and the figure (please see Fig. 1) was modified based on the reviewer’s suggestion, as illustrated in his figure R3. We now believe that readability of the figure has been significantly improved.

In addition, shear heating would stabilize wadsleyite and even favor a transformation back to olivine around 2170 K ($\approx 1900^{\circ}\text{C}$; Fig.R3). This means that, in case of transient rupture-induced melting, olivine will crystallize first, followed by wadsleyite. This could well explain the olivine+wadsleyite aggregates in fault veins (but not in the oblique lenticular aggregates (see major comment #6).

As mentioned above, we agree that our results cannot exclude the possibility of total melting of olivine during throughgoing faulting. To clarify the possibility of partial crystallization, we added following this sentence (lines 209-211):

“The olivine grains in the gouge in M2676 sample could be the product of back-transformation of wadsleyite or partial crystallization from the melt in the olivine stability field (1900–2100 K at 13 GPa: Fig. 1a).”

On Fig.4e and Fig.5c, the “Pt” and “Ni” patches give the same elemental signal as what the authors refer to as “Fe-rich phase”. The melting point of “Pt” is 1768°C (i.e. 2041 K), which is significantly less than the estimated peak temperature of 2500 K.

We constrained the peak temperature during the shear heating using the melting points of Pt and Ni. Please note the melting point of “Pt” as 1768°C (i.e. 2041 K), mentioned by the reviewer, is the value at the ambient pressure. It is well known that melting temperatures of most materials significantly increase with increasing pressure, and the melting point of pure platinum reaches ~ 2500 K at 15.5 GPa (Wang et al., 2001 Physica B). Considering that the platinum blobs contain iron, the liquidus temperature (i.e., melting point) could decrease down to ~ 2200 K. This means that the peak temperature reached at least ~ 2200 K. To clarify this point, following sentence was added (lines 174-176):

“The liquidus curve for the platinum-iron alloy constrains the peak temperature during the throughgoing faulting to the range of ~ 2200 – 2500 K (at 15.5 GPa)^{22,23} or higher.”

Consequently, both the metal spherulites and grainboundary phase could contain metals from the assembly, and a clarification is requested.

We agree that origin of the metal patches needs to be discussed. As already mentioned in the manuscript, the origin of platinum blobs is the platinum strain marker (lines 171-172).

“..., showing melting of the platinum strain marker and olivine during the throughgoing faulting.”

The origin of nickel blobs is the metal capsule surrounding the sample. To clarify this point, following sentence was added (lines 215-218):

“..., nickel-iron alloy blobs (10–200 nm), due to melting of the nickel capsule used in M2676 sample (Fig. 5c), also support the generation of instantaneous high temperature exceeding the melting point of pure nickel (~2100 K at 13 GPa)²².”

#2-6. Olivine transitions: anticracks vs oblique lenticular aggregates

The authors report oblique lenticular aggregates of mixed olivine polymorphs, which is, in my opinion, the key finding of the study. But there is a serious problem in the interpretation (e.g. lines 188-191). On one hand, indeed, the oblique lenticular aggregates seem to provide evidence for the process of rupture nucleation at the onset of transformational faulting (extremely important!!) BUT on the other hand, the microstructures reported by the authors indicate that these features are fundamentally different from anticracks (also visible on Fig. 5g), contrary to what the authors write at line 184. Consequently, the paragraph of lines 174-191 should be cut into two paragraphs: one paragraph for the oblique lenticular features (possibly the rupture nucleation stage) and one paragraph for anticracks (with no clear evidence for a link with local instability). In Fig.5g, we can see the anticracks (consisting of wadsleyite and/or ringwoodite) described by Green & Burnley (1989) and Green et al. (1990), and reproduced by further studies (Riggs & Green, 2005; Officer & Secco, 2020). The authors annotate the anticrack-like features with “Wad/Rin” but do not describe or discuss these features. It is unfortunate that the FIB section does not cross the anticracks (Fig. 5g). Anticracks form normal to the maximum compressive stress (+ some variability in case of rupture due to the stress distortion at the rupture front), and won't be aligned with the fault. In addition, anticracks seem cut by faults (see Fig.R2 or Fig.5g), suggesting that the transformation and the rupture do not occur within one single process (main hypothesis of the unclear “transformational faulting” model). The authors could explain that, while the oblique lenticular aggregates seem indeed related to rupture nucleation, the observed anticracks do not seem related to the fault initiation.

We presume that the sentence “*The morphology of the observed microstructures is quite*

similar to that of ‘anticracks’ in Mg₂GeO₄ (ref.^{2,23}) and Fe₂SiO₄ (ref.⁷).” caused this confusion. The lenticular packets are aligned with high angle (~30°) to the compression direction, but anticracks should be preferentially aligned to the direction normal to the compression direction. To clarify this point, the sentence was modified as follows (lines 233-236):

“The lenticular packets are aligned with high angles (~30°) to the compression direction and have a width of 2–3 μm. The morphology of the observed microstructures (other than the shape-preferred orientation) is quite similar to that of the ‘anticracks’ in Mg₂GeO₄ (ref.^{4,29}) and Fe₂SiO₄ (ref.⁹).”

We appreciate the reviewer’s important suggestion. However, because the observed microstructures were overprinted by many cracks formed during the decompression process (i.e., after the deformation experiment), we do not deal the cracks/anticracks in the recovered samples to avoid misinterpretation. We focus on the topic of fault nucleation in the paragraph (subsection “*Lenticular packets filled with nanocrystalline olivine and wadsleyite*”).

We agree that the transformation and the rupture do not occur within one single process, as pointed by the reviewer. To clarify the role of lenticular packets on fault nucleation, following sentences were added (lines 236-239):

“Linking-up of the lenticular packets with nanocrystalline olivine and wadsleyite should be the origin of weak fault gouge layers observed in the faulted samples (Figs. 4 and 5), consistent with the hypothesis of transformational faulting based on the anticrack model⁴ or the nano-shear band model⁵.”

Note, however, that the left side of Fig.5g shows many anticracks, while the right side does not show much of it. These anticracks might be a consequence of previous seismic ruptures (several events recorded). Considering a rupture nucleating around the center of the sample and propagating to the area shown in Fig.5g, the left side, full of anticracks (mostly normal to the main compressive stress or to the fault) would correspond to the compressive quadrant. In my opinion, here the authors have a good opportunity to discuss the rupture mechanisms thanks to these key microstructural imaging (Fig.7d-e).

As mentioned above, we consider that interpretation of cracks/anticracks observed in the recovered samples is risky because the microstructures were overprinted by many cracks formed during the decompression process.

Alternatively, we reported the occurrence of rupture during the faulting based on our AE data. Fig. 3b clearly show the process of rupture (i.e., a rupture nucleating around the center of the sample and propagating to outside the sample). Please see lines 122-127:

“In these three runs, one or two large AEs (maximum amplitude > 3V) occurred inside the sample, followed by relatively large AEs outside the sample and aligned with the fault plane crossing the sample (Fig. 3 and Supplementary Fig. 1). This suggests that the occurrence of a throughgoing fault slip caused by the rupture was

followed by rupture propagation outside the sample.”

Finally, in this paper, there is no mention of poirierite (the ω -olivine), recently recognized as the fourth olivine polymorph (Tomioka & Okuchi, 2017; Tomioka et al., 2021). Although this is relatively new and we do not know if poirierite has a proper P-T stability field (Ferrand & Deldicque, 2021), it should at least be mentioned in the introduction and an explanation for its absence (or absence of observation) in the reported microstructures should be provided in the discussion.

As suggested by the reviewer, we cited Tomioka et al. (2021 Commu. Earth & Environment) in Introduction. Following sentence about poirierite was added (lines 40-42):

“... with increasing pressure, although the transitions may proceed via an intermediate spinelloid structure¹.”

#2-7a. Line-by-line comments (Title)

The present study documents significant achievements + replications of features already documented in the literature. In my opinion, the title does not highlight the new insights of the paper, i.e. the evidence of the nucleation stage of the mysterious “transformational faulting” “mechanism”. → I would suggest the following rephrasing and clarification:

“In-situ X-ray and acoustic monitoring during olivine phase transitions reveal the rupture nucleation stage of transformational faulting” or “Simultaneous in-situ X-ray and acoustic monitoring during olivine high-pressure transitions reveal the rupture nucleation stage of deep earthquakes”

We really appreciate the kind offer of the reviewer on the title of our paper. As the reviewer commented, one of the most important points of this study is that the nucleation stage of the mysterious transformational faulting has been clarified (Fig. 9b). However, not only the nucleation stage but also the rupture propagation stage (Fig. 9b) is also important. The term “seismic faulting” includes both the nucleation and the rupture propagation stages. Thus, we believe that the current title “*In situ X-ray and acoustic observations of deep seismic faulting upon phase transitions in mantle olivine*” well fits to this study. On the other hand, it might be good idea to emphasize the rupture nucleation process, as proposed by the reviewer, but it could be too strict and difficult to understand for broader readers.

#2-7b. Line-by-line comments (Abstract)

Lines 21: “abruptly decreases to zero at 680 km” → Deep earthquakes can occur deeper (e.g. Kuge, 2017; Kiser, 2021). Some large events were recorded down to \approx 750 km depth, due to cold slabs portions reaching below the transition zone (Kiser et al., 2021). → clarification needed.

We agree that deep earthquakes can occur at depths greater than 680 km. However, their activity is low. Due to the length of abstract is limited, following sentence was added to the 1st paragraph of Introduction (lines 43-46):

“An abrupt decrease in the seismicity from a peak at ~600 km to almost zero at 680 km (ref.²), with a very limited number of earthquakes at deeper regions down to 750 km (ref.³), also implies an essential role of the phase transitions of olivine on occurrence of deep-focus earthquakes.”

In addition, we modified the discussion part of the text according to the reviewer’s comments (lines 337-345, see also our response#1-6):

“In fact, the observed deep-focus earthquakes are reported to be located along an isotherm of ~1000 K in deep slabs⁵⁵ and can propagate outside the MOW³⁸. Numerical studies based on experimental kinetic data show that metastable olivine in cold slabs probably persists to depths of ~630 km, below which ringwoodite is the major phase down to ~700 km (ref.⁵⁶), where it breaks down to bridgmanite and ferropericase. The shear zones filled with nanocrystalline olivine required for throughgoing faulting would not be preserved in subducted slabs at depths beyond this kinetic boundary of ~630 km, consistent with the abrupt decrease in seismicity at depths below ~600 km (ref.²).”

Lines 21-22: “because brittle failure is unlikely to occur under the corresponding pressures of 13–24 GPa.” → This is a classic statement that all the community has been writing for decades because of the misunderstandings we have had regarding deep rupture processes. They are NOT unlikely, otherwise they would not occur so frequently. So, this sentence is not really informative and sounds outdated and misleading. → see major comment #1.

We replaced the term “brittle failure” by “conventional brittle failure” to meet the reviewer’s request. Please see our response#2-1 for details.

Line 23: “suggested” → At this stage, this is not a “suggestion” anymore. → rephrasing needed.

We believe this issue has not been completely verified and “suggested” is an appropriate phrase. Nevertheless, following this suggestion, “It has been suggested” was replaced to “It becomes increasingly clear” (lines 22-23).

“It becomes increasingly clear that pressure-induced phase transitions of olivine are responsible for the occurrence of the earthquakes...”

Line 24: “deep earthquakes” → This term is generally used to described both intermediate-depth earthquakes and deep-focus earthquakes. Writing “deep-focus earthquakes” would help avoid confusion. In any case, in should be properly defined in the introduction.

We agree, and the term “deep earthquakes” was replaced by “deep-focus earthquakes”

in this manuscript.

Lines 25-27: → unnecessary attack against previous seminal works. Science progresses every day, and this is not a reason to minimize the importance of previous works. The present experimental work exists thanks to previous works and can be presented as a necessary continuity of previous achievements, instead of illegitimate criticism → could be easily rephrased.

It is not our intention to attack the earlier studies, and we rephrased the relevant sentences in Abstract as follows: (lines 22-26):

“It becomes increasingly clear that pressure-induced phase transitions of olivine are responsible for the occurrence of the earthquakes, based on deformation experiments under pressure. However, many such experiments were made using analogue materials and those on mantle olivine are required to verify the hypotheses developed by these studies.”

Line 30: “colder” → “cold”

Line 30: “mantle transition region” → “mantle transition zone” NB: Even if both terminologies are found in the literature, it is better to keep using the dedicated vocabulary.

We appreciate the reviewer’s kind suggestions. Fixed.

Lines 32-32: “immediately before the rupture” → WARNING: vocabulary issue + misconception. The acoustic activity located inside the sample is necessarily the consequence of dynamic ruptures (e.g. double-couple cracks). It cannot be before! Thus, the term “rupture” at line 33 is incorrect. Instead, the authors probably refer to the “sample-scale faulting”, due to coalescence (major comment #4).

We agree that the expression “immediately before the rupture” may be misleading, and rephrased this sentence as (lines 29-31):

“We find that throughgoing faulting occurs only at very limited temperatures of 1100–1160 K, accompanied by intense acoustic emissions at the onset of rupture.”

Lines 36-37 → This is an important finding, which definitely legitimates publication in Nature Communications!

Lines 37-39 → This is correct, and consistent with most recent studies on deep-focus earthquakes.

We appreciate the reviewer’s positive comments.

#2-7c. Line-by-line comments (Introduction)

Line 40: “(Mg,Fe)₂SiO₄ olivine …” → “Polymorphs of (Mg,Fe)₂SiO₄ …” or

“Natural olivine polymorphs, (Mg,Fe)₂SiO₄, ...”

In the following sentence (*...it shows successive phase transitions to wadsleyite...: lines 38-42*), we are introducing the pressure-induced phase transitions of olivine. Thus, the subject of this sentence should be “(Mg,Fe)₂SiO₄ olivine”.

Lines 45-46 → WARNING: anticracks do not form along the maximum compressive stress but normal to it... → see major comment #6. → clarification needed.

We agree, and “along the direction of the maximum shear stress” was removed from the sentence. Similarly, another sentence was modified as follows (lines 50-51):

“The anticrack is orientated normal to the direction of the maximum compressive stress and...”

Lines 49-50: “which behaves as a low-viscosity fluid due to...” → Incorrect → suggested rephrasing: “which would be characterized by low viscosity due to...” (see major comment #1).

As mentioned above, “fluid” was replaced by “layer”. Please see lines 52-53.

“..., that behaves as a low-viscosity layer due to grain-size-sensitive (GSS) creep^{4,5,7}.”

Lines 52-53: “which may not be an adequate analogue” → Inappropriate. In spite of differences, always mentioned by the cited studies, both Mg₂GeO₄ and (Mg,Fe)₂SiO₄ transform to a spinel or spineloid structure when cold slabs travel in the mantle transition zone. In addition, citing Green et al. (1990) to trigger doubt regarding their own work does not seem honest. Indeed, Green et al. (1990) have shown that anticracks form in natural minerals and not only analog materials. → attack not needed → could be easily written in a less “aggressive” way (see major comment #2).

Please see our response#2-2 for details. We understand the reviewer’s suggestion. The sentence was modified as follows (lines 53-55):

“However, Mg₂GeO₄ olivine transforms directly to the spinel phase without passing through the modified spinel phase⁸, implying that this might not be a perfect analogue of mantle olivine⁸.”

Lines 58-60 → The reference to Officer & Secco (2020) is relevant, and I want to draw your attention about additional publications that could help understand/arguing. Recent studies clearly explain that the triggering mechanisms of initial mechanical instabilities and the rupture mechanisms should be distinguished (e.g. Ferrand et al., 2021; Mao et al., 2022). → (see major comment #3).

Line 59: “suggested that development of the fault slip requires ...” → “suggested that the development of substantial fault slip requires ...”

We agree, and the sentence was modified as follows (lines 59-61). Ferrand et al. (2021 JGR) is cited in this sentence.

“...but suggested that the development of substantial fault slip and the propagation of the rupture requires an

additional mechanism such as adiabatic instability^{10-12.}”

Line 60 → *The reference to Aben et al. (2015) is not appropriate here.*

We agree, and this reference was removed.

Line 60: “*olivine/fayalite*” → “*olivine (forsterite or fayalite)*” or “*olivine (San Carlos or fayalite)*” Remark: *Fayalite is olivine too, and Mg₂GeO₄ is the analogue of forsterite. We need to remain very careful using the terms, especially considering that the targeted journal can bring readers from various fields, who may get lost at some point.*

We agree, and the sentence was modified as follows (lines 61-62):

“Both of these studies on silicate olivine (San Carlos or fayalite), however, were...”

Lines 60-62 → *This sentence appears incorrect or excessive. And the formulation is not clear. The experiments of Officer & Secco (2020) were performed using another type of multi-anvil apparatus, but several studies (Wang et al., 2004; Gasc et al., 2011; Schubnel et al., 2013; Incel et al., 2017; Ferrand et al. (2017) have already used the D-DIA apparatus with both high-energy X rays and acoustic recording, notably leading to substantial improvements of the understanding of intermediate-depth earthquakes. I have contributed to several of these studies and I can confirm that the evolution of stress and strain is precisely monitored and that strain at the scale of the deformation assembly can be linear using reasonable conditions. → see major comment #2.*

We believe that there is no incorrect/excessive information, contrary to the comment. As mentioned in this sentence “*Both of these studies on silicate olivine (San Carlos or fayalite), however, were performed with the ‘stress-relaxation test’¹³, where the timing and degree of sample deformation cannot be controlled.*”(lines 61-64), two previous studies on silicate olivine (Green et al., 1990 Nature; Officer & Secco, 2020 PEPI) used Kawai-type (or Walker-type) multianvil apparatuses. The cell assembly designed for the stress-relaxation test is needed for deformation experiments in a Kawai-type apparatus (Karato and Rubie, 1997 JGR). Please note that high-energy X-ray was not used in Green et al. (1990 Nature) and Officer & Secco (2020 PEPI).

#2-7d. Line-by-line comments (Results)

Line 69: “*synchrotron in situ deformation experiments on sintered body ...*” → *several problems: first, what is “in-situ” is the X-ray diffraction and radiography, not the experiment;*

We agree, and the sentence was modified as follows (lines 68-70):

“In this work, we performed high-pressure, high-temperature deformation experiments using in situ synchrotron X-ray diffraction and radiography combined with acoustic recording on a sintered body of a natural olivine powder...”

second, the high-energy X-ray produced by the synchrotron technology is important to mention, because they are a key of the study, which requires clarification in the formulation;

We agree, and the use of synchrotron X-ray is mentioned in the modified sentence (lines 68-69).

“In this work, we performed high-pressure, high-temperature deformation experiments using in situ synchrotron X-ray diffraction and radiography combined with...”

On the other hand, we should mention that the use of “high-energy X-ray produced by the synchrotron technology” is quite popular in high-pressure community over the last 20 years since the third-generation synchrotron facilities (APS, ESRF, and SPring-8) were constructed in late 1990s. Please note that the energy value of the synchrotron X-ray (60 keV) is shown in Method (lines 535-536). The value of 60 keV indicates that a high-energy synchrotron X-ray was used in this study.

“Two-dimensional radial diffraction patterns of monochromatic X-rays (energy 60 keV) were taken by...”

third, other key aspects of the used technology, such a high pressure and acoustic recording, should be mentioned at the same time. → I recommend for example the following rephrasing: “high-pressure, high-temperature deformation experiments with in-situ synchrotron radiation and acoustic recording on sintered samples of ...”

We thank the reviewer’s kind suggestion. The sentence was modified as suggested.

Line 77 → change “:” for “.”

Lines 92-93: “does not account” → “accounts” (double negation with “Neither ... nor...”).

We fixed them, as suggested.

Line 94: “Peierls creep” → This is fully contradictory with the previous sentence. → clarification needed. + Why highlighting the Peierls creep and not other mechanisms?

We found that a recent experimental study on the Peierls creep of wet olivine (Ohuchi, 2022 PEPI) is partly consistent with our experimental results. Taking this into account, the sentence was modified as follows (lines 101-104). Also, Fig. 2 was updated.

“While the highest strength at each temperature is close to the value calculated from the Peierls creep of wet olivine¹⁷, neither dislocation creep¹⁸, nor dislocation-accommodated grain boundary sliding of wet olivine¹⁹ account for the observed strengths.”

Especially, I think the authors recall that transformations (e.g. phase transitions, grain-

boundary disordering, melting, amorphization...) are among the deformation mechanisms (Wheeler, 2020; Green et al., 2015; Yamauchi & Takei, 2020; Ferrand & Deldicque, 2021; Kloe et al., 2000; Samae et al., 2021).

Such transformations may contribute to the observed softening of olivine. Unfortunately, flow laws of olivine at high pressures (>11 GPa) have not been well investigated because of the difficulty of the experiments. The previous experimental studies listed by the reviewer (Wheeler, 2020; Green et al., 2015...) were conducted at low pressures (< 2 GPa) and thus their results cannot be applicable to this study. In contrast, Ohuchi (2022 PEPI) reported that a grain-size-sensitive creep of olivine could be induced by the olivine-wadsleyite transition. To clarify this point, following sentence was added (lines 106-108).

“Partial contribution of GSS creep of olivine/wadsleyite induced by the nucleation of wadsleyite¹⁷ to the bulk strength may cause softening of the samples, resulting in further dispersion of strength values.”

The authors could consider the transformation-assisted grain-boundary sliding proposed by Ferrand & Deldicque (2021). → see major comment #3..

Please see our response#2-3. We already considered grain boundary sliding in our discussion (Fig. 9a). We used flow-law parameters determined by previous high-pressure experiments (Table S1). Please note that Ferrand & Deldicque (2021 Minerals) is an experimental study on Mg₂GeO₄ and thus their results cannot be applicable to this study (i.e., silicate olivine).

Line 96: “AE” should be defined into parentheses earlier, because some readers may not know that this refers to “acoustic emissions”.

Line 96: “which” → “but” or “and”

Line 97: “suggesting” → “indicating”

We fixed them, as suggested.

98 → Wouldn't “ductile” be the right word here?

The word “plastic flow” was replaced by “plastic deformation” (lines 110-111).

“..., indicating the dominance of microcracking-free plastic deformation above 1160 K (Fig. 2b).”

Line 106: “the semi-brittle flow was terminated by faulting” → If I understand well what the authors report in the previous subsections, this semi-brittle flow is the source of the acoustic emission, and “faulting” refers to the final sample-scale faulting, due to the coalescence of the seismic and/or aseismic faults forming during the semi-brittle flow. → Substantial rephrasing and details required (see major comment #4).

As mentioned in our response#2-4, the term “faulting” might be vague. In most cases of

this study, “faulting” means “throughgoing faulting” (corresponding to sample-scale faulting). In the revised manuscript, “faulting” was replaced by “throughgoing faulting”.

Line 106: “followed by a sudden large pressure drop (blow-out)” → In the high-pressure community, a “blow-out” refers to an unexpected pressure drop and end of the experiment due to a mechanical instability at the scale of the pressure chamber. In all likelihood, the very large deformation associated with the coalescence is responsible for the stress transfer at the assembly scale triggering blow-out (see major comment #4).

As mentioned in our response#1-2, we added the result of an earlier run (M2472), where sample deformation was terminated by a sudden blow-out, but formation of any fault was not observed in the recovered sample (Fig. S4b). This supports our consideration that a catastrophic adjustment of the pressure medium/gasket induced by the faulting results in a blow-out (lines 132-134).

“The faulting is likely to induce a catastrophic adjustment of the pressure medium/gasket which inevitably leads to a blow-out...”

The result of run M2472 is consistent with the reviewer’s comment. To strengthen this point, following sentence was added (lines 137-143):

“Note that a blow-out is not the cause of faulting in our cell assembly. As demonstrated in M2472, stress abnormally decreased down to -2 GPa (i.e., a strong tensile stress) accompanied by AEs before the occurrence of a blow-out (Supplementary Fig. 4a). Such strong tensile stress, which would be a characteristic phenomenon for blow-outs, was not detected in other runs. Even though sudden sample shortening was observed at the time of a blow-out, development of any faults was not observed in the recovered sample (Supplementary Fig. 4b).”

Line 108: “the former three runs” → “these three runs”

Line 110: “outside the sample along the fault plane crossing the sample” → “outside the sample and aligned with the fault plane” NB: this most likely correspond to ruptures in MgO and/or Al₂O₃, as indicated by fractures in Fig. 5a and Fig.5f.

We fixed them, as suggested.

Line 111: “suggesting the occurrence of a fault-slip associated rupture” → Extremely confusing for the reader. I recall that, considering a single event, the fault slip is the consequence of the rupture (see major comment #4).

We agree, and the sentence was modified as follows (lines 125-127):

“This suggests that the occurrence of a throughgoing fault slip caused by the rupture was followed by rupture propagation outside the sample.”

Line 115: “samples” → “sample”

Line 116: “subsequent active AEs” → confusing: “active acoustic emissions” does not make sense, so I propose the following rephrasing: “subsequent substantial/intense acoustic activity”.

Line 117: “upon faulting” → confusing → “upon coalescence” or “upon sample-scale faulting” (see major comment #4).

We fixed them, as suggested. Please note that the word “sample-scale” has the same meaning as “throughgoing”. Since we already used “throughgoing” in the manuscript, “throughgoing” was used, instead of “sample-scale”.

Line 117: “catastrophic adjustment of the pressure” → This is the stress transfer I refer to in my comment to line 106. → It is not a pressure adjustment but a stress adjustment, and more precisely, a “catastrophic perturbation of the stress field around the sample”. Then, the blow-out is associated with a pressure drop, which is a consequence of the latter.

In this sentence, “catastrophic adjustment of the pressure medium/gasket” is considered. Please note that “pressure” is not considered.

Line 115: “before faulting” → WARNING: very problematic confusion (see major comment #4).

We agree, and the word “faulting” was replaced by “the occurrence of throughgoing faulting” (lines 145-146).

“...before the occurrence of throughgoing faulting in M3425 with high time-resolution stress/strain measurements (Fig. 3a).”

Line 124 → The reference to Okazaki & Hirth (2016), i.e. study of dehydration embrittlement of lawsonite at intermediate-depth conditions, is totally out of topic. → find adequate reference or remove sentence.

The sentence was removed, as suggested.

Line 128 → This argument comes from Chernak & Hirth (2011), and has been referred to in studies on mantle rheologies (e.g. Ferrand et al., 2017). → change reference for Chernak & Hirth (2011).

We agree, and Okazaki & Hirth (2016 Nature) was replaced by Chernak & Hirth (2011 Geology). Please see lines 524-526.

“The ratios of temperature-ramping rate and strain rate in the path#3 runs ($T/\dot{\epsilon} = 1.0\text{--}1.4 \times 10^4$ K) are comparable to those expected values for natural subduction zones⁶⁰.”

Lines 128-129: No... First, as noted above, “faulting” should be properly defined

(rupture vs coalescence). Second, the use of the term “precursor” is totally incorrect, because splitting is the consequence of coalescence. → see major comment #4.

We agree, and “precursor of faulting” was replaced by “the consequence of throughgoing faulting” (lines 150-151).

“Splitting of the middle strain marker in the X-ray radiograph is expected as a consequence of throughgoing faulting.”

Lines 132-133: “commencement of faulting” → seismic and/or co-seismic brittle events (depending on acoustic activity) → see major comment #4.

Please see our response#2-4.

Lines 133-136 → Long sentence that could be shortened, and the end is unclear.

We agree, and the sentence was divided into two sentences as follows (lines 155-159):

“Because the X-ray radiographic imaging was interrupted by the acquisition of each diffraction pattern with a long exposure time (70–540 seconds), it was hard to determine the whole faulting process solely by the in-situ measurements. Thus, microstructures of the recovered samples were also carefully examined to obtain complementary evidence for the process after the deformation experiments.”

Line 138 + throughout the manuscript: “partial melting” → Are the authors sure about the evidence of partial melting? To what extent are they sure that it could not be fractionation of the melt due to crystallization during fault slip? → see major comment #5.

Please see our response#2-5 for details. In this section, we are discussing about the lower limit of the peak temperature (Figs. 4-6). We do not exclude the possibility of total melting. In fact, the term “partial melting” was replaced by “melting” in the revised manuscript.

Line 140: “sample of M3425” → “sample of M3425”

We presume that the corrected phrase “sample of M3425” should be replaced by “sample M3425”. Fixed.

Line 140: “...; where the displacement of the fault is 280 μm .” → “...; with a cumulative displacement of 280 μm .” → see major comment #4.

Line 147: “... of gouge layer.” → “... of the gouge layer.”

We fixed them, as suggested.

Line 148: QUESTION: Is there any other source of Fe in the assembly?

As shown in Fig. S4a, there is no other iron source in the assembly.

Line 149: “partial” → see major comment #5.

Line 150: “Evidence of partial melting” → I cannot see that. It could be either partial melting (unlikely) or partial crystallization! → see major comment #5.

Please see our response#2-5 for details. We do not exclude the possibility of total melting. In fact, the term “partial melting” was replaced by “melting” in the revised manuscript. Please note that we found a possibility of total melting followed by partial crystallization in run M2676.

Lines 150-152 → If the authors are interested in understanding the recent achievements about seismic rupture dynamics, I would be happy to explain it in details. They can read the recent comparative study by Ferrand et al. (2021), considering various studies from either field geology or laboratory experiments and proposing simple modeling of the advancing melt front. → see major comment #5.

Lines 152-153: “instantaneous temperature increase” → I would advise that the authors support their work with some adequate references about adiabatic/diabatic dynamic melting during dynamic rupture (e.g. Ferrand et al., 2021).

We appreciate the reviewer’s kind suggestion. We found that Ferrand et al. (2021 JGR) is a very nice work, and we cited it in Introduction (lines 59-61):

“...but suggested that the development of substantial fault slip and the propagation of the rupture requires an additional mechanism such as adiabatic instability¹⁰⁻¹².”

Line 156: “a throughgoing faults” → “a throughgoing fault” or “a throughgoing fault zone”

Again, we appreciate the reviewer’s kind suggestion. This is a typo. Fixed.

Line 156: “in the OL92 sample of M2676.” → “in run M2676 (sample OL92).”

Line 156: “displacement” → “cumulative displacement”

Lines 160-161: “melt-like phase” → Term not adequate → Suggested rephrasing: “amorphous material” or “grain-boundary amorphous phase” or “grain-boundary glass” → correction needed.

We fixed them, as suggested.

Line 161: “No micrometric wadsleyite” → Why do the authors specify “micrometric” here? Could there be nanometric wadsleyite? → clarification needed.

We consider that the word “micrometric” is confusing. It was removed from the sentence.

Line 162: “The melt-like phase is amorphous (Figs. 5d-e), which is enriched...” → “The phase wetting grain boundaries (Figs. 5d-e) is amorphous and enriched ...”

We fixed this sentence, as suggested.

Lines 163-171: WARNING: unreasonable interpretation, which seems contradictory with the reported results. → see major comments #5 and #6.

Please see our response#2-5 and #2-6 for details. Please understand that this study is the 1st report on the shear heating in mantle olivine under the pressure/temperature conditions of the mantle transition zone. As previous studies were conducted at much lower pressures/temperatures, the interpretation needs not be common between previous studies and this study.

Line 174: “without faulting ” → WARNING: incorrect. The authors have not demonstrated that there would be no faulting in the sample. In addition, this would be contradictory with the recording of acoustic activity from inside the sample. → rephrasing required (see major comment #4).

To avoid confusion, “without faulting” was replaced by “without throughgoing faulting”.

Lines 177-178: “Both of these two and other smaller AEs are all confined in the sample region ” → It would have been interesting to see the single-event faults corresponding to these labquakes, because such confined faults are not contaminated by the other part of the experimental assembly and can give important insights on rupture mechanics (e.g. Ferrand et al., 2017).

Please see our response#2-4 for details. It would be interesting if single-event faulting can be detected in experiments. Unfortunately, that is technically impossible. We consider that such kinds of experiments are future works in our community.

Lines 181-183 → Speculative and useless.

This sentence was supported by the following sentence (lines 231-233):

“In fact, the growth of numerous nanocrystalline olivine and wadsleyite grains from old olivine grains containing many dislocations is suggested (Fig. 5h).”

Line 183: “pulverization ” → WARNING: pulverization is a fully different process, associated with rupture mechanics and related to loss of cohesion and further grain size reduction during rupture propagation (e.g. Reches & Dewers, 2005; Aben et al., 2016; Incel et al. 2019). + Note that there is NO mention of pulverization in the cited paper (Liu et al., 1998)!!

We agree, and “pulverization of” was replaced by “reduction in”.

Line 184: “growth of” → “the growth of”

Line 184: “is observed” is doubled → remove.

We fixed them, as suggested.

Lines 184-185 → This is not a convincing argument. Coincidence is not proof of causality.

We consider that Fig. 5h strongly support our interpretation that the growth of numerous nanocrystalline olivine/wadsleyite grains from old olivine grains containing many dislocations occurred. But taking the reviewer’s suggestion into account, our statement was toned down (i.e., “observed” was replaced by “suggested”). Please see lines 231-233.

“In fact, the growth of numerous nanocrystalline olivine and wadsleyite grains from old olivine grains containing many dislocations is suggested (Fig. 5h).”

Line 187: “The morphology of the observed microstructures is quite similar to that of ‘anticracks’” →WARNING: No, it is not. The previous sentence (lines 185-186) deals with the oblique lenticular packets (Fig.7d-e), very different from anticracks and with different orientation! → see major comment #6.

Please see our response#2-6 for details. We agree, and the sentence was modified as follows (lines 234-236):

“The morphology of the observed microstructures (other than the shape-preferred orientation) is quite similar to that of the ‘anticracks’ in Mg₂GeO₄ (ref.^{4,29}) and Fe₂SiO₄ (ref.⁹).

Lines 188-191 → Yes and no: on one hand, indeed, the lenticular packets (key observation of the study) seem to provide evidence for the process of rupture nucleation at the onset of transformational faulting (extremely important!!) BUT on the other hand, the microstructures reported by the authors indicate that these features are fundamentally different from anticracks, also visible on Fig. 5g.

As commented in our response#2-6, we consider that interpretation of cracks/anticracks observed in the recovered samples (such as Fig. 5g) is risky because the microstructures are overprinted by many cracks formed during the decompression process. Formation of such cracks is unavoidable in high-pressure experiments. Thus, we consider that Fig. 5g does not affect our interpretation written in the sentence.

Lines 174-191 → This paragraph should be cut in two paragraphs: one paragraph for the oblique lenticular features (possibly the rupture nucleation stage) and one paragraphs for anticracks (with no clear evidence for a link with local instability).

If this paragraph is cut in two paragraphs, the second one will become very short. Also, the topics of the rupture nucleation stage and local instability are strongly connected to

each other. Thus, we do not cut this paragraph in two paragraphs.

#2-7e. Line-by-line comments (Discussion)

Line 196: “proceeds on GBs” → “proceeds at GBs”

Lines 196-197: “To estimate the rate of olivine-wadsleyite phase transition on” → “To estimate K at”

Line 200: “follows” → “follow”

We fixed them, as suggested.

Line 203 → the units of a_0 and b_0 should be specified.

We agree, and the actual values, units and the references were added to the sentence (please see line 252):

“where $a_0 (=5.3 \times 10^{42} \text{ m}^{-2} \text{ s}^{-1} \text{ K}^{-1})^{31}$ and $b_0 (=4.5 \times 10^4 \text{ ms}^{-1} \text{ K}^{-1})^{33}$ are constants,…”

Line 204: “of reaction” → “of the reaction”

Line 211: “the volume fraction transformed” → “the transformed volume fraction”

Line 212: “the rate constant during the early stages of transition K is given as” → “the rate constant K during the early stages of transition is given as follows”

We fixed them, as suggested.

Line 213 → Choice required between “K” and “K”. Usually, italic font style is used for variables and regular font style for constants.

We choose italic font style for “K” and other parameters, following Rubie et al. (1990 JGR).

Remark: the definition of K (eq.3) should be given first, followed by the detailed description of all variables in the equation, including additional equations (eq.1 and eq.2).

We agree, and the orders of the Eqs. (1) to (3) were switched. Please see lines 247-252.

“The K during the early stages of transition is given as a function of the nucleation rate \dot{N} and growth rate \dot{x}

(ref^{31,32}):

$$K = \frac{\pi}{3} \dot{N} \dot{x}^3 \quad (1)$$

$$\dot{N} = a_0 T \exp\left(-\frac{\phi \Delta G_{hom}^*}{kT}\right) \exp\left(-\frac{Q}{RT}\right) \quad (2)$$

$$\dot{x} = b_0 T \exp\left(-\frac{Q}{RT}\right) \left[1 - \exp\left(-\frac{\Delta G_r}{RT}\right)\right] \quad (3)$$

where…“

Line 214: “Formation of” → “The formation of”

Line 215 → For sake of clarity, it should be defined in the first sentence of the Discussion section, that this parameter is a constant in fixed P-T conditions but may

significantly vary during the experiment.

Line 215: “of the K” → “of K”

We fixed them, as suggested.

Line 219: “very low” is not nothing...

The AE activity is almost ignorable but not nothing above 1200 K. Please see Fig. 2b.

Line 221: “faulting” → clarification needed. → see major comment #4.

The word “faulting” was replaced by “throughgoing faulting”

Line 223: “Elastic energy stored in ...” → “The elastic energy stored in ...”

We fixed it, as suggested.

Line 224: “high-temperature” is obvious and redundant with “thermal runaway”

We agree, and thus “high-temperature” was removed from the sentence.

Lines 224-226 → WARNING: serious lack of consideration for the literature (major comment #2).

We cited Kelemen and Hirth (2007 Nature) (Kelemen and Hirth, 2007) (Kelemen and Hirth, 2007) (Kelemen and Hirth, 2007) (Kelemen and Hirth, 2007) because this is the most important study among the previous ones on self-localizing thermal runaway. We know that other studies are also important, but we could not cite them due to the limitation in citation total number (i.e., up to 70). Please understand that we already cited 70 of previous studies in the manuscript.

Line 235: “Instantaneous strain rate” → “The dynamic strain rate”

Line 238: “by formation” → “by the formation”

We fixed them, as suggested.

Lines 238-239 → Regarding the initiation of the shear crack instability, it would be fair to mention the reaction-induced grain size reduction (Incel et al., 2017; Thielmann, 2018b) and the grain-boundary disordering mechanism (Ferrand & Deldicque, 2021). Please note that the grain-boundary instability is reported in Mg₂GeO₄ in a narrow temperature window, which can constitute an efficient viscosity reduction mechanism before reaching the melting conditions. The authors might find useful to mention premelting (e.g. Takei et al., 2019; Yamauchi & Takei, 2020) → see major comment #3.

Please see our response#2-3 for details. In the 1st paragraph of Introduction, we already reviewed grain-size-sensitive creep (i.e., the grain-boundary instability) induced by

phase transition. However, we consider that other mechanisms (grain size reduction, grain-boundary disordering and amorphization) have been proposed for the cause of intermediate-depth earthquakes, not for deep-focus earthquakes. Such mechanisms are beyond the scope of this study.

Line 240: “On the other hand” → should come after “On one hand” in order to highlight two things, before concluding about the level of consistency or discrepancy between the two things.

“On one hand” corresponds to “diffusion creep/superplasticity of olivine” written in the previous sentence (please see line 294-296).

“... followed by shear localization accelerated by diffusion creep⁴³ or superplasticity of olivine⁴⁴ (Fig. 9a: see Methods for the details of calculations). On the other hand, diffusion creep of...”

Lines 240-242 → WARNING: Such a strong statement would require a proper demonstration!!! It does not sound reasonable at all.

Following this advice, our statement was toned down. The word “cannot” was replaced by “does not” (lines 295-298).

“On the other hand, diffusion creep of nanocrystalline wadsleyite⁴⁵ with a grain size of 20 nm does not account for the initiation of adiabatic instability due to slower diffusivities of silicon in high-pressure polymorphs of olivine⁴⁶.”

Lines 242-245 → WARNING: This is incorrect. The authors completely ignore grain-boundary processes and the increased importance of grain boundaries in case of grain-boundary instability. This is not acceptable. In addition, I want to stress that there is not only one rate-limiting process for superplasticity. I recommend that the authors read more papers of material sciences, especially ceramics and metallurgy. They could have a look at the trial of interdisciplinary integration by Ferrand & Deldicque (2021). → see major comment #3.

Following this advice, our statement was toned down. The word “would” was replaced by “might”. Please understand that the reported flow laws for silicate olivine applicable to high pressures are very limited (all listed in Table S1), because deformation experiments at elevated pressures are difficult. Ferrand & Deldicque (2021 Minerals) is an experimental study on Mg₂GeO₄ analogue at low pressures and thus their results cannot be applicable to silicate olivine. Please see our response#2-3 for details.

Line 264: “due to accumulation” → “due to the accumulation”

We fixed it, as suggested.

Line 265 → Again, to what extent are we sure that deep earthquakes are triggered by

an adiabatic instability? → Are they “triggered” by this process or do they “nucleate” via this process? This is a key distinction that we all need to understand and/or discuss. → clarification needed (I would suggest to replace “triggered” by “nucleated”). → see major comment #3.

We consider that nucleation of deep-focus earthquakes is triggered by formation of lenticular packets filled with nanocrystalline olivine/wadsleyite. Adiabatic instability and fault slip occur at the onset of rupture, namely, these two processes are the results of earthquake nucleation. To clarify these points, our discussion was improved (lines 331-338):

“The present study suggests that the deep-focus earthquakes nucleate when the temperature of the MOW reaches those close to 1100–1160 K at pressures of the mantle transition region. Here the nanocrystallization of olivine occurs upon the phase transition (i.e., formation and linking-up of the lenticular packets filled with nanocrystalline olivine/wadsleyite) followed by throughgoing faulting due to shear instability (Fig. 9b), although this critical temperature may be lower on the geological time scale (Fig. 8). In fact, the observed deep-focus earthquakes are reported to be located along an isotherm of ~1000 K in deep slabs⁵⁵...”

Line 268: “superheating” → What does that mean?! + grammatical issue ... → clarification needed.

Superheating is a technical term often used in the community of high-pressure mineralogy. Superheating means a sudden temperature increase due to latent heat release via a phase transition. Please see Bina (1998 Earth Planets Space), for example.

Lines 272-277 → This is very important and it should be discussed in light of recent machine-learning based seismological findings, which suggest that deep seismicity within the mantle transition zone would occur in the unstable rim around the metastable wedge (Mao et al., 2022). The unstable rim is most likely corresponding to a thermal limit at the balance between displacive and diffusive processes (Ferrand & Deldicque, 2021).

Yes, we know that Mao et al. (2022 Commu. Earth & Environment) is a nice work, but we already cited a pioneering work by Zhan (2017 EPSL). Not only Mao et al. but Zhan (2017 EPSL) also reported deep seismicity in the unstable rim around the metastable wedge. Please understand that we already cited 70 of previous studies in the manuscript. We need to select them.

Line 280 → Please pay attention to the most-recent b-value analysis supporting this sentence (Mao et al., 2022).

Lines 280-284 → Actually, our recent study (Mao et al., 2022) suggests that it is controlled by temperature and OH content. → see major comment #3.

Following the reviewer#1’s advice, the sentences about the b-value were removed from

the revised manuscript. Please see our response#1-6 for details.

Lines 285-287 → If this is correct and fully applicable to all subducting slabs, then deep earthquakes between 630 and 750 km depth would be due to some other transformation. But what?

We consider that the rapid decrease in seismicity from ~600 km to ~680 km is due to the kinetic boundary of ~630 km for metastable olivine (estimated by Kubo et al., 2009 EPSL). This is well explained by our findings. Please see lines 342-345.

“The shear zones filled with nanocrystalline olivine required for throughgoing faulting would not be preserved in subducted slabs at depths beyond this kinetic boundary of ~630 km, consistent with the abrupt decrease in seismicity at depths below ~600 km (ref.²).”

Regarding limited seismicity at depths 680-750 km, other transformation (e.g., ringwoodite to bridgemanite+magnesiowustite) may cause the seismicity. However, this is beyond the scope of our study.

Lines 287-288: “shear zones filled with nanocrystalline olivine required for faulting would not be preserved” → But their souvenir would remain! (i.e. nanocrystalline wadsleyite and/or ringwoodite).

Yes, shear zones filled with nanocrystalline wadsleyite and/or ringwoodite may exist at depths greater than ~630 km. However, our calculations in Fig. 9a suggests that nanocrystalline wadsleyite does not trigger faulting.

Line 289 → See major comment #1.

Please see our response#2-1 for details. We understand that deep seismicity has been reported at depths ~750 km, but the important point is the “abrupt” decrease in seismicity from ~600 km to ~680 km.

#2-7f. Line-by-line comments (Figures and captions)

Lines 292-293: “a, Faulting occurred. b, No faulting occurred” → INCORRECT (major comment #4).

The word “faulting” was replaced by “throughgoing faulting”.

Lines 295-304 → the crosses should be described in the caption. A clear distinction should be written between the latter (corresponding to rupture-induced melting, i.e. local and transient) and the other symbols (corresponding to the average bulk conditions). → clarification needed.

Following sentence was added to the caption for Fig. 1:

“Crosses show the lower limit of the peak temperature during the throughgoing faulting (estimated from the

microstructures: see text for details).”

Caption of Fig.2 → For colors and symbols, it should be written hat the legend is in Fig.1.

The sentence “Large symbols represent the runs with throughgoing faulting (M2676, M3100 and M3425).” was moved to the caption for Fig. 1.

Fig.3a → I suggest to adjust the colors of the strain and amplitude axes to be consistent with the rest and to help the reader not to get lost.

The color for the strain axis was changed to gray, as suggested. Please understand that the color for the amplitude axis should be black because two colors (i.e., red and green) are assigned to the AE amplitude.

Fig.3c → For $\varepsilon = 0.85$, an explanation for the top pink arrow (throughgoing faulting) could be useful for the reader.

Following sentence was added to the caption for Fig. 3:

“The pink arrow shows a discontinuous increase in strain due to throughgoing faulting.”

Fig.3 → I suggest to move b and c next to a., which I believe would better match the journal format.

We modified the figure and caption based on the journal format.

Lines 323-324: “No diffraction peak of wadsleyite 240 was detected throughout the run, as shown in a dotted area.” → It looks a bit odd. It is good that it is written in the caption, but the location of this dotted area in the plot seems misleading. → remove. In addition, I would rephrase for a more understandable sentence as follows: “The diffraction peak 240 of wadsleyite was not detected in the run.”

Because the reviewer#1 pointed out that the “no Wad” purple boxes are not informative, the boxes were removed from the figures (please see our response #1-3 for details).

Line 327: “shows typical errors for the location of hypocenter.” → “shows the uncertainty range for hypocenter locations.”

Line 328: “at each strain ε ” → “for three values of strain ε representative of the entire experiment.”

We appreciate the reviewer’s kind suggestions. We fixed them, as suggested.

Lines 331-332: “A backscattered electron image” → “Backscattered electron image”

Line 333: “A STEM image of a fault associating the gouge layer (the region “b” in

a).” → “Scanning transmission electron microscope (STEM) image of a 4- μ m-thick fault (located in a).” Remark: It is important to define every acronym when it occurs for the first time in the manuscript. + most readers know SEM and TEM, but not necessarily STEM.

Line 334: “A magnified view” → “Magnified view”

Line 335: “Typical grain size” → “The grain size”

Line 336: “A TEM image of wadsleyite grains in the gouge layer. A diffraction pattern is also shown.”

→ “TEM image of a wadsleyite nanocrystal in the olivine-wadsleyite gouge aggregate, with associated diffraction pattern.”

Line 337: “An enlarged STEM image and an element map” → “Enlarged STEM image and Mg/Fe element map”

Line 338 → “In element maps, grayscale corresponds to concentration of elements”

→ “In the element map, the intensity of colors accounts for element concentrations”

Caption of Fig.4: all the occurrences “(the region “y” in x)” can be shorter → “(located in x)” .

Again, we appreciate kind suggestions. We fixed them, as suggested.

Fig.4d + Fig.5e → WARNING: such diffraction patterns should not be truncated and they should be accompanied with a scale (in nm⁻¹)!!! → See major comment #1.

Please see our response#2-1 for details. Miller’s index (i.e., hkl) is shown in the figure, and thus a scale bar is not added to the indexed diffraction pattern (please see TEM studies, for example Miyajima et al., 2006 GRL). Please note that the pattern shown in Fig. 4d is not truncated.

Fig.5g → WARNING: overinterpretation. → See major comment #5

Please note that no interpretation is included in Fig. 5g. “Si-rich patch” was identified using STEM/EDS, as demonstrated in Fig. 6b.

Caption of Fig.5 → Same remarks as for the caption of Fig.4. → corrections needed. + “Mj” definition is missing.

We fixed it, as suggested.

Fig.6 → The composition estimates from EDS measurements do not seem convincing. → see major comment #5.

Please see our response#2-5 for details. Uncertainties in the measurements are shown in Fig. 6. Following sentences were added to Methods to show that our analyses were properly conducted (lines 650-653, 656-658):

“The cores of each crystal having a diameter larger than 100nm (except for cases of ultrafine grains/patches) were

selected for quantitative analysis with a probe size of ~2 nm to avoid the contamination of spectra from the neighboring grains.”

“Chemical compositions are recalculated as 100 wt.% anhydrous. The uncertainty of the calculated chemical composition reflects the error in the counts of characteristic X-rays.”

Caption of Fig.6 → *again, majorite is not defined.*

Please note that majorite (Mj) is defined in the revised caption for Fig. 5.

Caption of Fig.7: “without occurrence of faulting” (line 371) → *WARNING* → *see major comment #4.*

The word “faulting” was replaced by “throughgoing faulting”

Caption of Fig.8: “The temperature dependence of the olivine-wadsleyite transition *K* is calculated ...” → “Temperature dependence of the rate constant *K* for the olivine-wadsleyite transition calculated...”

Caption of Fig.9: “mantle transition region” → “mantle transition zone”

We fixed them, as suggested.

Extended Data Figure 4 → *The large yellow dashed circle is hiding the arrow showing the tungsten ring. → clarification possible.*

Line thickness for the large yellow dashed circle was decreased. The arrow showing the tungsten ring is visible in the modified figure (please see Fig. S5a).

#2-7g. Line-by-line comments (Methods)

Line 419: “8wt.%” → *add space*

We fixed it, as suggested.

Lines 417-429: → *The authors do not describe the mixture procedure of olivine and enstatite, and they do not tell about the homogeneity of the enstatite distribution.*

To explain the details of the mixed-powder preparation, following sentences were added (lines 460-466):

“Gem-quality crystals of olivine (from San Carlos, USA) and orthoenstatite (from Kilosa, Tanzania) having no inclusions were finely ground for 2 hours in an agate mortar. The fine-grained powders of olivine and orthoenstatite were weighed in the desired proportion (i.e., 92 wt.% olivine and 8 wt.% orthoenstatite) and then mechanically mixed/ground for 3 hours in an agate mortar. The mechanical mixing process is essential to obtain OL92 samples with homogeneously dispersed olivine and orthoenstatite grains.”

Lines 424-426: → *Why not. However, again, the authors should discuss why they*

consider the solidus and not the liquidus, while field observations (e.g. Ferrand et al., 2018) on mantle rocks show local but total melting on the rupture path.

Please see our response #2-5 for details. To avoid confusion, “Partial” was removed from the sentence (lines 471-473):

“Melting of the OL100 and OL92 samples is used for estimation of a peak temperature during faulting (e.g., OL100: >2500 K at 15.5 GPa; OL92: ~2170 K at 13 GPa)^{24,26}.”

Line 426: “Mechanical behavior” → “The mechanical behavior” + several other similar grammatical issues in throughout the section (many “The” are missing). → language verification needed.

We fixed it, as suggested. The final version of the manuscript was checked by two native English speakers/scientists.

Line 429: “independent on these two samples” → “independent of this compositional different.”

Line 453: “events of small AEs (the maximum amplitude <IV)” → “small events (AE maximum amplitude <IV)”

Line 464: “equilibrium” → “equilibria”

Line 488: “later” → “the last”

Line 497: “The uncertainty of” → “The uncertainty on”

Line 532 → remove “on the signals”

Line 540: “thorough” → “through”

Line 543 → remove “high-pressure” NB: It would be useful to remove all redundant information.

We appreciate kind suggestions. We fixed them, as suggested.

Line 546 → Note that this definition of the acoustic emissions magnitude is used by previous studies on similar lithologies (e.g. Schubnel et al., 2013; Ferrand et al., 2017; Kita & Ferrand, 2018).

Please see our response#1-1 for details. We cited Lei and Sato (2007 Tectonophys.) for the definition.

References

- Bina, C.R., 1998. A note on latent heat release from disequilibrium phase transformations and deep seismogenesis. *Earth Planet Space* 50, 1029-1034.
- Chernak, L.J., Hirth, G., 2011. Syndeformational antigorite dehydration produces stable fault slip. *Geology* 39, 847-850.
- Ferrand, T., Deldicque, D., 2021. Reduced viscosity of Mg₂GeO₄ with minor MgGeO₃ between 1000 and

1150 °C suggests solid-state lubrication at the lithosphere-asthenosphere boundary. *Minerals* 11.

Gasc, J., Schubnel, A., Brunet, F., Guillon, S., Mueller, H.-J., Lathe, C., 2011. Simultaneous acoustic emissions monitoring and synchrotron X-ray diffraction at high pressure and temperature: Calibration and application to serpentinite dehydration. *Phys Earth Planet Inter* 189, 121-133.

Gleason, G.C., Green II, H.W., 2009. A general test of the hypothesis that transformation-induced faulting cannot occur in the lower mantle. *Phys Earth Planet Inter* 172, 91-103.

Green II, H.W., Burnley, P.C., 1989. A new self-organizing mechanism for deep-focus earthquakes. *Nature* 341, 733-737.

Green II, H.W., Shi, F., Bozhilov, K., Xia, G., Reches, Z., 2015. Phase transformation and nanometric flow cause extreme weakening during fault slip. *Nat Geosci* 8, 484-490.

Irifune, T., Ringwood, A.E., 1987. Phase transformations in a harzburgite composition to 26 GPa: implications for dynamical behavior of the subducting slab. *Earth Planet Sci Lett* 86, 365-376.

John, T., Medvedev, S., Rüpke, L.H., Anderson, T.B., Podladchikov, Y.Y., Austrheim, H., 2009. Generation of intermediate-depth earthquakes by self-localizing thermal runaway. *Nat Geosci* 2, 137-140.

Karato, S., Rubie, D.C., 1997. Toward an experimental study of deep mantle rheology: A new multianvil sample assembly for deformation studies under high pressures and temperatures. *J Geophys Res* 102, 20111-20122.

Kelemen, P.B., Hirth, G., 2007. A periodic shear-heating mechanism for intermediate-depth earthquakes in the mantle. *Nature* 446, 787-790.

Kubo, T., Kaneshima, S., Torii, Y., Yoshioka, S., 2009. Seismological and experimental constraints on metastable phase transformations and rheology of the Mariana slab. *Earth Planet Sci Lett* 287, 12-23.

Lei, X., Satoh, T., 2007. Indicators of critical point behavior prior to rock failure inferred from pre-failure damage. *Tectonophysics* 431, 97-111.

Liu, M., Kerschhofei, L., Mosenfelder, J.L., Rubie, D.C., 1998. The effect of strain energy on growth rates during the olivine-spinel transformation and implications for olivine metastability in subducting slabs. *J Geophys Res* 103, 23897-23909.

Mao, G., Ferrand, T., Li, J., Zhu, B., Xi, Z., Chen, M., 2022. Unsupervised machine learning reveals slab hydration variations from deep earthquake distributions beneath the northwest Pacific. *Communications Earth and Environment* 3.

Miyajima, N., Ohgushi, K., Ichihara, M., Yagi, T., 2006. Crystal morphology and dislocation microstructures of CaIrO₃: A TEM study of an analogue of the MgSiO₃ post-perovskite phase. *Geophys Res Lett* 33, L12302.

Officer, T., Secco, R.A., 2020. Detection of high P, T transformational faulting in Fe₂SiO₄ via in-situ acoustic emission: Relevance to deep-focus earthquakes. *Phys Earth Planet Inter* 300, 106429.

Ohtani, E., 1979. Melting relation of Fe₂SiO₄ up to about 200 kbar. *J Phys Earth* 27, 189-208.

Ohtani, E., Kumazawa, M., 1981. Melting of forsterite up to 15 GPa. *Phys Earth Planet Inter* 27, 32-38.

Ohuchi, T., 2022. Grain-size-sensitive creep of olivine induced by oxidation of olivine in the Earth's deep

upper mantle: Implications for weakening of the subduction interface. *Phys Earth Planet Inter* 326.

Okazaki, K., Hirth, G., 2016. Dehydration of lawsonite could directly trigger earthquakes in subducting oceanic crust. *Nature* 530, 81-83.

Perrillat, J.P., Chollet, M., Durand, S., van de Moortéle, B., Chambat, F., Mezouar, M., Daniel, I., 2016. Kinetics of the olivine-ringwoodite transformation and seismic attenuation in the Earth's mantle transition zone. *Earth Planet Sci Lett* 433, 360-369.

Perrillat, J.P., Daniel, I., Bolfan-Casanova, N., Chollet, M., Morard, G., Mezouar, M., 2013. Mechanism and kinetics of the α - β transition in SanCarlos olivine $Mg_{1.8}Fe_{0.2}SiO_4$. *J Geophys Res* 118, 110-119.

Rubie, D.C., Tsuchida, Y., Yagi, T., Utsumi, W., Kikegawa, T., Shimomura, O., Brearley, A.J., 1990. An in situ X ray diffraction study of the kinetics of the Ni_2SiO_4 olivine-spinel transformation. *J Geophys Res* 95, 15829-15844.

Schubnel, A., Brunet, F., Hilairet, N., Gasc, J., Wang, Y., Green II, H.W., 2013. Deep-focus earthquake analogs recorded at high pressure and temperature in the laboratory. *Science* 341, 1377-1380.

Seto, Y., Nishio-Hamane, D., Nagai, T., Sata, N., 2010. Development of a software suite on X-ray diffraction experiments. *The Review of High Pressure Science and Technology* 20, 269-276.

Singh, A.K., Balasingh, C., Mao, H.-K., Hemley, R.J., Shu, J., 1998. Analysis of lattice strains measured under nonhydrostatic pressure. *J. Appl. Phys.* 83, 7567-7575.

Tomioka, N., Bindi, L., Okuchi, T., Miyahara, M., Iitaka, T., Li, Z., Kawatsu, T., Xie, X., Purevjav, N., Tani, R., Kodama, Y., 2021. Poirierite, a dense metastable polymorph of magnesium iron silicate in shocked meteorites. *Communications Earth and Environment* 2.

Wang, Y., Durham, W.B., Getting, I.C., Weidner, D., 2003. The deformation-DIA: A new apparatus for high temperature triaxial deformation to pressures up to 15 GPa. *Rev Sci Instrum* 74, 3002-3011.

Zhan, Z., 2017. Gutenberg-Richter law for deep earthquakes revisited: A dual-mechanism hypothesis. *Earth Planet Sci Lett* 461, 1-7.

REVIEWERS' COMMENTS

Reviewer #1 (Remarks to the Author):

The authors have answered clearly to all my suggestions and comments. the manuscript rips can be published as is.